



# Resistance and resilience of stream metabolism to high flow disturbances

Brynn O'Donnell[1] and Erin R. Hotchkiss[1]

[1]Department of Biological Sciences, Virginia Polytechnic Institute and State University, Blacksburg, Virginia

**Correspondence:** Erin R. Hotchkiss (ehotchkiss@vt.edu)

**Abstract.** Streams are ecosystems organized by disturbance. One of the most frequent and variable disturbances in running waters is elevated flow. Yet, we still have few estimates of how ecosystem processes, such as stream metabolism (gross primary production and ecosystem respiration; GPP and ER), respond to high flow events. Furthermore, we lack a predictive framework for understanding controls on within-site metabolic responses to flow disturbances. Using five years of high-frequency dissolved oxygen data from an urban- and agriculturally-influenced stream, we estimated daily GPP and ER and analyzed metabolic changes across 15 isolated high flow events. Metabolism was variable from day to day, even during lower flows. Thus, we calculated metabolic resistance as the magnitude of departure from the dynamic equilibrium during antecedent lower flows and quantified resilience from the days until GPP and ER returned to the range of antecedent dynamic equilibrium. We evaluated correlations between metabolic resistance and resilience with characteristics of each high flow event, antecedent conditions, and time since last flow disturbance. ER was more resistant and resilient than GPP. GPP was typically suppressed following flow disturbances, regardless of disturbance intensity. In contrast, the ER magnitude of departure increased with disturbance intensity. Additionally, GPP was less resilient and took longer to recover (0 to >9 days, mean = 2.2) than ER (0 to 2 days, mean = 0.6). Given the flashy nature of streams draining human-altered landscapes and the variable consequences of flow for GPP and ER, testing how ecosystem processes respond to flow disturbances is essential to an integrative understanding of ecosystem function.

## 1 Introduction

Disturbances can alter stream ecosystem function by changing flow while influencing carbon and nutrient inputs, transformations, and exports (Stanley et al., 2010). Stream biogeochemical cycles are altered by long-term 'press' disturbances, such as land use change (e.g., Plont et al. 2020), and by episodic 'pulse' disturbances, such as transitory changes in allochthonous inputs (e.g., Bender et al. 1984; Dodds et al. 2004; Seybold and McGlynn 2018). Here, we use the definition of disturbance from White and Pickett (1985): "any relatively discrete event in time that disrupts the ecosystem... and changes resources, substrate availability, or the physical environment". Frequent disturbances generate oscillations that form a pulsing steady state



(*sensu* Odum et al. 1995) that includes ambient variability in processes (Resh et al., 1988; Stanley et al., 2010). Stream distur-
bances come in many forms, including: rapid increases in the volume and velocity of water, drought, substrate movement, and
anthropogenic alterations of channel morphology, flow, or solute chemistry (Resh et al., 1988).

Elevated flow is one of the most pervasive, frequent disturbances to streams. Flow disturbances can scour the benthos,
increase turbidity, and reduce light – all of which can change stream function (Hall et al., 2015; Blaszczak et al., 2019).
However, flow is an inherent characteristic of streams and may influence stream function along a "subsidy-stress" gradient
(*sensu* Odum et al. 1979; Figure 1). Extreme high flows can stress stream biota and induce conditions unfavorable for biotic
processes, whereas more 'normal', frequent high flows can stimulate internal biogeochemical transformations by bringing in
limiting nutrients or organic matter subsidies (Lamberti and Steinman, 1997; Roley et al., 2014; Demars, 2019). How changes
in flow subsidize or stress stream functions will depend on a variety of factors, including the ecosystem process of interest.

Stream metabolism is an integrative whole-ecosystem estimate of the carbon fixed and respired by autotrophs and het-
erotrophs. Metabolism is most commonly estimated via diel changes in dissolved oxygen (Hall and Hotchkiss, 2017): au-
totrophs produce oxygen during gross primary production (GPP); autotrophs and heterotrophs consume oxygen during ecosys-
tem respiration (ER). Together, ER and GPP can elucidate whether a stream is a net producer (autotrophic; GPP > ER) or
consumer (heterotrophic; ER > GPP) of carbon. Ecosystem metabolism is coupled with other ecosystem processes (e.g., nitro-
gen uptake, Hall and Tank 2003) and is used to monitor the stream health (Young et al., 2008) as well as ecosystem responses
to disturbance and restoration (e.g., Arroita et al. 2019; Blersch et al. 2019; Palmer and Ruhi 2019).

Metabolism on any given day is influenced by current and past environmental factors. GPP can increase with light (Mulhol-
land et al., 2001; Roberts and Mulholland, 2007), nutrients (Grimm and Fisher, 1986; Mulholland et al., 2001), temperature
(Acuña et al., 2004), and transient storage (Mulholland et al., 2001). ER is controlled by organic carbon availability (e.g.,
Demars 2019), as well as the same physicochemical conditions as GPP, and consequently often mirrors GPP (e.g., Roberts
et al. 2007; Griffiths et al. 2013; Roley et al. 2014). Antecedent conditions may also play a role in the variability of ecosystem
responses to flow (McMillan et al., 2018; Uehlinger and Naegeli, 1998). GPP and ER respond differently to flow distur-
bances (O'Donnell and Hotchkiss, 2019), likely influenced by where the microbes contributing to GPP and ER reside within
the heterogeneous stream benthos (e.g., Uehlinger 2000, 2006). Autotroph reliance on light for energy creates a stream bed
commonly dominated by photoautotrophic algal communities and associated heterotrophs. Many heterotrophs, on the other
hand, are established within the substrata and hyporheic zone, which can increase resistance and resilience of ER relative to
GPP (Uehlinger, 2000; Qasem et al., 2019). Environmental drivers of metabolism fluctuate in response to disturbances (e.g.,
Uehlinger 2000) but also vary sub-daily to seasonally, thus inducing temporal variation in GPP and ER during base flows that
are best characterized as a pulsing steady state or dynamic equilibrium (e.g., Roberts et al. 2007).

The subsidy-stress relationship between flow and ecosystem function likely induces a range of metabolic responses to and
recovery from flow changes (Figure 1). Both GPP and ER may decline due to disturbance during high flows (Uehlinger,
2006; Roley et al., 2014; Reisinger et al., 2017); however, flow changes can also stimulate metabolism (Roberts et al., 2007;
Demars, 2019). Ultimately, resistance is reflected in the capacity of microbial assemblages to withstand a flow disturbance, with
metabolic processes not reduced or stimulated outside of a pulsing steady state. Resistance captures the instantaneous response





of ecosystem metabolism to a flow disturbance. We can also quantify post-disturbance ecosystem responses by estimating
resilience: the time it takes for a process returns to equilibrium following a disturbance (Carpenter et al., 1992). The resilience
of ER and GPP following a flow disturbance may take anywhere from days to weeks (e.g., Uehlinger and Naegeli 1998; Smith
and Kaushal 2015; Reisinger et al. 2017), and likely varies with season and the magnitude of disturbance (Uehlinger, 2006;
Roberts et al., 2007). A flow event of lesser magnitude may yield higher resistance and resilience for both GPP and ER, by
supplying subsidizing, limiting nutrients and organic matter from the terrestrial landscape without inducing extreme scour.
Stream metabolism appears to have low resistance to disturbance but high resilience (Uehlinger and Naegeli, 1998; Reisinger
et al., 2017). Understanding how different attributes of flow events (e.g., magnitude, timing) control resistance and recovery
trajectories is a critical next step in characterizing metabolic responses to flow changes within and among ecosystems.

We quantified ecosystem resistance and resilience over several years of isolated, high flow events to examine controls on and
patterns of stream metabolic responses to disturbance. We had four hypotheses (Figure 1): (H1) ER will be more resistant than
GPP to flow disturbances, given the protection of many heterotrophs within the streambed; (H2) there will be a stimulation of
GPP and ER at intermediate flow disturbances due to an influx of limiting carbon and nutrients; (H3) metabolic resistance and
resilience will change with the size of the event, with larger flow disturbances inducing more stress due to enhanced scour;
and (H0) some flow events will not push GPP and ER outside of their pulsing equilibrium. We analyzed response and recovery
dynamics (i.e., resistance and resilience) relative to a pulsing equilibrium for 15 isolated flow events across 5 years in a flashy
urban- and agriculturally-influenced stream. We assessed which antecedent conditions or disturbance characteristics influence
within-site variability in GPP and ER resistance and resilience.

## 2  METHODS

### 2.1  Study site

Stroubles Creek is a third-order, urban- and agriculturally-influenced stream draining a 15 km$^2$ sub-watershed of the New
River in Southwest Virginia in the United States (Figure A1, O'Donnell and Hotchkiss 2019). The mean annual precipita-
tion of Stroubles Creek's catchment is 100.6 cm, with more than half (54%) of that precipitation falling from May-October
(PRISM Climate Group, 2013). Annual mean air temperature is 11.3°C (0.4-22.0°C monthly mean minimum and maximum;
PRISM Climate Group 2013). The catchment draining into Stroubles Creek at our study location is 85.5% developed, 11.6%
agriculture (pasture and crops), and 2.9% forested (Homer et al., 2015). Our study site on Stroubles is a part of the Stream
Research, Education, and Management Lab (StREAM Lab, www.bse.vt.edu/research/facilities/StREAM_Lab.html), and has
been monitored by Virginia Tech researchers for over 10 years.

### 2.2  Sensor data collection

High temporal resolution sensor data were collected from 2013-01-08 through 2018-04-14. Dissolved oxygen (DO) (mg L$^{-1}$),
turbidity (nephelometric turbidity unit, NTU), conductivity (ms cm$^{-1}$), pH, and temperature (°C) data were logged at 15-



minute intervals by an in situ YSI 6920V2 sonde (Hession et al., 2020; O'Donnell and Hotchkiss, 2019). Because a freeze
event impaired DO measurements from the YSI sonde, we gap-filled with calibration-checked and comparable data from an
adjacent PME MiniDOT from 2017-09-01 to 2018-04-14 (Figure A2; O'Donnell and Hotchkiss 2019). We obtained light
data and barometric pressure measurements from a nearby weather station. A Campbell Scientific CS451 pressure transducer
recorded stage measurements every 10 minutes. Velocity and width measurements were taken over multiple years to create

site-specific relationships between stage, velocity, wetted width, and discharge. A stage-discharge relationship was created in
2013 and updated in 2018. Sensors were calibrated every 2-4 weeks.

   To remove lower-quality sensor data due to sensor error or periods of low flow, we used data cleaning and quality checks as in
O'Donnell and Hotchkiss (2019). Briefly, we excluded values below the 1% and above the 99% quantile for physicochemical
parameters that were heavily skewed (i.e., turbidity and conductivity). We removed physicochemical values we knew to be

unreasonable (e.g., turbidity was cut off at zero). We calculated daily medians of physicochemical parameters for all days that
had at least 80% of measurements over the course of the day after confirming the 80% cutoff as one that would not bias daily
medians from dates without gaps in sensor measurements. Data from lower flow periods when individual sensors may have
been out of water (Hession et al., 2020) were excluded when values were out of range of grab sample calibration checks.

### 2.3    Ecosystem metabolism

We estimated GPP, ER, and K (air-water gas exchange) from diel $O_2$ (DO), light, and temperature sensor data using the same
inverse modeling approach and data as O'Donnell and Hotchkiss (2019). We selected the *streamMetabolizer* R package for our
analyses (Appling et al., 2018a), which uses Bayesian parameter estimation and a hierarchical state space modeling framework
to generate daily estimates of GPP, ER, and K that create the best fit between modeled and observed DO data (Appling et al.
2018b; Equation 1; Table 1).

$$mDO_i = mDO_{i-\Delta t} + \frac{GPP \times PAR_t}{z \times \Sigma PAR} + \frac{ER}{z}\Delta t + K_O(DO_{sat(i-\Delta t)} - mDO_{i-\Delta t})\Delta t \qquad (1)$$

   We used most of the default model specs for *streamMetabolizer*. Model convergence was visualized via *traceplot* in the *rstan*
package (Stan Development Team, 2019) to identify the proper number of burn-in steps (500); we saved 2000 Markov chain
Monte Carlo (mcmc) steps from four chains after burn-in. We modeled GPP, ER, and K with both observation error and process
error. We calculated mean daily standard error derived from the mcmc-derived distributions of GPP and ER. Additionally, to

decrease the chances of equifinality between GPP, ER, and K estimates (Appling et al., 2018b), we constrained day-to-day
variability in K by binning the range of possible K estimates according to discharge (O'Donnell and Hotchkiss, 2019). We
divided yearly discharge into six bins, which the hierarchical modeling framework of *streamMetabolizer* then used to create
K-Q relationships to constrain model K estimates (O'Donnell and Hotchkiss, 2019). We used nighttime linear regression of
DO as another way to estimate the range in K in Stroubles (Hall and Hotchkiss, 2017) and used these estimates of K to quality

check values of modeled K from *streamMetabolizer* (e.g., Figure A3).

   We quality-check metabolism model output as in O'Donnell and Hotchkiss 2019. We removed all metabolism estimates that
were biologically impossible, such as negative GPP or positive ER (ER is modeled as a negative flux of $O_2$ consumption).





Next, we used diagnostics from *fit()* in *stan* to remove values resulting from a poor model fit or lack of chain convergence (Stan Development Team, 2019). We removed dates with poor model convergence when *Rhat* exceeded 1.1 and poor model

fit when *N_eff* (effective sample size) ended at or exceeded the product of the number of chains (4) and the number of saved mcmc steps (2000) specified for our model. Additionally, to avoid using biased estimates of metabolism, we removed K values below the 1% (< 3.38 d$^{-1}$) and above the 99% (> 27.21 d$^{-1}$) quantile of model estimates. 246 days of metabolism estimates were ultimately removed due to these QA/QC criteria, resulting in 1375 days (of 1621 total from 2013-01-08 to 2018-04-14) quality-checked GPP and ER for further analyses.

## 2.4   Selection of isolated flow events

To identify flow events for our analyses of metabolic resistance and resilience, we calculated the percent change in cumulative daily discharge (Q) relative to the day prior (Equation 2).

$$\%\Delta Q = \frac{Q_i - Q_{i-1}}{Q_{i-1}} \times 100 \tag{2}$$

We selected isolated flow events that had a greater than 50% Q change relative to the antecedent cumulative daily Q. We defined

isolation as a period of three days before and three days after a high flow event where no other flow events exceeding 10% Q change occurred. In total, there were 15 isolated flow events across all 5 years that met our criteria for isolated flow events and had quality-checked metabolism estimates (Figure 2). Hydrograph and metabolism time series for each isolated flow event are available in Appendix Figures 4 - 18.

## 2.5   Characterizing metabolic resistance and resilience

To acknowledge the pulsing, day-to-day variability of GPP and ER, we used metabolism estimates from three days prior to define a range of antecedent metabolism for each isolated flow event. We quantified metabolic responses to flow disturbances by comparing the pre-event metabolic range with event and post-event metabolism rates. To assess resistance, we estimated metabolic magnitude of departure (M) during events to quantify the resistance of GPP and ER to disturbance. We calculated M per isolated flow event by comparing the difference between GPP and ER to the nearest value of the antecedent range (Equation

3; Figure 3),

$$M = 1 - \frac{X_{event}}{X_{prior}} \tag{3}$$

where $X_{event}$ is either GPP or ER (g O$_2$ m$^{-2}$ d$^{-1}$) on the day of the isolated flow event. $X_{prior}$ is the maximum or minimum value of GPP or ER from the antecedent range, depending on if the isolated flow event resulted in a stimulated (increased) or repressed (reduced) metabolic response. For instance, if GPP was repressed during a flow event, M was calculated as the

difference between GPP for the isolated flow event and the minimum value from the antecedent 3-day range (Figure 3). If the estimate of GPP or ER on the event day did not fall outside of the antecedent range, the magnitude of departure was zero, thus indicating high resistance. A negative magnitude of departure (M) represents a suppression, and a positive M a stimulation, of GPP or ER relative to the antecedent equilibrium.





To quantify the resilience of GPP and ER, we estimated recovery intervals (RI) by counting the number of days until
metabolic rates returned to within the range of pre-event values, signifying a return to antecedent dynamic equilibrium (Fig-
ure 3). If metabolism from the isolated flow event did not fall outside of the antecedent range and M is zero, there was no RI
(metabolism cannot recover if it never shifts outside the range of normality). To ensure additional flow events did not obscure
the recovery interval of GPP or ER, we stopped counting RI the day before the next event (i.e., if another flow event happened
four days later, we stopped counting RI at 3 days), and have noted this in our results as days+ and used different symbols in
data figures. To test for statistically significant differences between ER and GPP recovery intervals ($RI_{ER}$ and $RI_{GPP}$) and ER
and GPP magnitude of departure ( $M_{ER}$ and $M_{GPP}$), we ran Welch's t-tests in R (R Core Team, 2018).

### 2.5.1   Testing controls on metabolic resistance and resilience

Quantifying how different antecedent conditions induce variable responses from GPP and ER is critical to furthering our
understanding of stream ecosystem responses to flow disturbances. We assessed three categories of potential predictors of
metabolic resistance and resilience: antecedent conditions, characteristics of the isolated flow event, and characteristics of the
most recent prior flow event. Antecedent conditions included median GPP and ER (g $O_2$ m$^{-2}$ d$^{-1}$), turbidity (NTU), water
temperature (°C), and PAR ($\mu$mol m$^{-2}$ s$^{-1}$). Antecedent medians for turbidity were estimated from seven days prior due to
missing sensor data. For all other variables, we estimated median values from three days prior to the flow event for correlations
between metabolism M and RI. Flow event characteristics included flow magnitude (% change of cumulative daily discharge;
m$^3$ d$^{-1}$; Equation 2), time (HH:MM) of peak discharge (m$^3$ s$^{-1}$), and environmental conditions (e.g., light, temperature, turbidity,
season) on the event day. Characteristics of the most recent flow event included the magnitude of and days since the last flow
event. We visually identified the most recent flow event (% change in cumulative daily discharge > 50) prior to each isolated
flow event. We ran bivariate correlation analyses to quantify the strength and directions of linear relationships between predictor
variables and metabolic resistance and resilience using the R *cor.test* function (R Core Team, 2018). All modeling and analyses
were conducted in R (R Core Team, 2018).

## 3   Results

### 3.1   Flow and metabolism

Stroubles Creek is a hydrologically flashy stream, with frequent high flow events (Figure 2). Cumulative daily discharge for
days with quality-checked metabolism estimates ranged from 66 to 114,408 m$^3$ d$^{-1}$, with a median of 6,230 m$^3$ d$^{-1}$. The 15
isolated flow events selected for analyses were within the mid-high range of all cumulative daily discharge values, and were
of magnitudes that occurred multiple times a year (Table 2, Figure 2). We identified isolated flow events of interest based on
percent changes in flow, so changes in cumulative daily discharge are proportional across seasons. GPP ranged from 0.002
to 17.3 g $O_2$ m$^{-2}$ d$^{-1}$ (median = 4.0); ER ranged from -0.1 to -27.2 g $O_2$ m$^{-2}$ d$^{-1}$ (median = -9.6) (Figure 4). Stroubles was





heterotrophic (|ER| > GPP), except for 38 days (3%) where GPP > ER, all of which occurred in spring except for one day in
the fall.

## 3.2    Metabolic resistance and resilience

GPP most often declined following an isolated flow event, whereas ER rarely deviated from the antecedent equilibrium during
higher flows. The magnitude of departure for GPP ($M_{GPP}$) ranged from -0.92 to 0.09, with a median of -0.14 (Table 3; Figure
5). GPP was inhibited during 11 and stimulated during 2 of 15 isolated flow events. The magnitude of departure for ER ($M_{ER}$)
ranged from -0.59 to 0.22, with a median of 0. Three of 15 flow events stimulated ER, 5 repressed ER, and ER did not deviate
from the antecedent equilibrium for 7 events (i.e., $M_{ER}$ was 0). Overall, the direction of ER response to elevated flow was more
variable than that of GPP.

Although GPP exhibited stronger responses across isolated flow events than ER, $M_{GPP}$ and $M_{ER}$ were positively correlated
($R^2$ = 0.39, p = 0.007, Figure 5) and not significantly different (p = 0.06, $\alpha$ = 0.05). $M_{GPP}$ was less than $M_{ER}$ for nearly all flow
events, except for one in which $M_{GPP}$ and $M_{ER}$ were both zero and two where $M_{GPP}$ and $M_{ER}$ were both small (Figure 5, Figure
A19). The isolated flow event that induced the greatest stimulation of GPP ($M_{GPP}$ = 0.09) also stimulated ER ($M_{ER}$ = 0.05).
Of the other two events that stimulated ER, one had no GPP response ($M_{GPP}$ = 0) and the other had a minor GPP reduction
($M_{GPP}$ = -0.06). Similarly, the only other event that stimulated GPP ($M_{GPP}$ = 0.03) had no ER response, suggesting many flow
disturbances may decouple GPP and ER.

Both GPP and ER typically recovered from flow-related stimulation or reduction in less than three days (Table 3). There
were many isolated flow events where GPP took multiple days to recover but ER never departed from the antecedent dynamic
equilibrium (i.e., M = 0; Figure 5). When $M_{GPP}$ and $M_{ER}$ were both greater than zero, ER almost always recovered faster than
GPP. $RI_{GPP}$ ranged from 1-9+ d, with an average of 2.5 d (Table 3). $RI_{ER}$ ranged from 1-2 d, with an average of 1.1 d. There was
only one isolated flow event where GPP recovered before ER. While ER always recovered before another flow event occurred,
GPP did not recover before another flow event for 4 of 15 analyzed events. Excluding the 4 of 15 events when GPP did not
recover, recovery intervals for GPP and ER were not correlated (Figure 5) or significantly different (p = 0.12, $\alpha$ = 0.05).

## 3.3    Controls on metabolic resistance and resilience after a flow disturbance

Although GPP and ER are linked processes, the variables that were moderate or strong predictors of resistance or resilience
(r > 0.5) differed between ER and GPP (Table 4; Figure 6). There were no predictors with moderate or stronger relationships
for both $M_{GPP}$ and $RI_{GPP}$. Because the median $RI_{ER}$ was zero, bivariate correlations could not be used to determine potential
predictors of ER resilience. Peak discharge of the flow event was negatively correlated with $M_{ER}$ (r = -0.59). The magnitude
of each disturbance, characterized by the % change in cumulative daily discharge, was negatively correlated with $M_{GPP}$ (r =
-0.46, p = 0.08) and $M_{ER}$ (r = 0.66, p = 0.01, $\alpha$ = 0.05) (Figure 7), but was not significantly correlated with $RI_{GPP}$ (p = 0.54; $\alpha$
= 0.05). Overall, there were multiple environmental controls on metabolic resistance or resilience that were strongly correlated
with either GPP or ER, but no significant drivers of both GPP and ER resistance and resilience.





## 4   Discussion

### 4.1   Metabolic resistance and resilience

GPP and ER responded differently to isolated flow events in a heterotrophic stream draining a heterogeneous urban-agricultural landscape. Notably, ER was more resistant than GPP (Figure 1). Of the fifteen high flow events analyzed here, three events
stimulated ER and there were two instances of minor GPP stimulation (Table 3). The flow disturbance of heterotrophic activity was likely balanced by increased inputs of organic carbon from terrestrial sources that stimulated respiration (Roberts et al., 2007; Demars, 2019). We found similar differences in how GPP and ER change with discharge across all days with metabolism estimates from Stroubles Creek (instead of event-specific analyses presented in this paper); GPP decreased but ER was relatively constant on days with higher than median flow (O'Donnell and Hotchkiss, 2019). How often and when GPP and
ER respond similarly to flow disturbances may differ among ecosystems as a function of their metabolic balance (GPP:ER), nutrient limitation status, and history of flow disturbance. Ultimately, flow-induced changes disproportionately disturbed GPP relative to ER.

ER was more resilient than GPP. Differences in ER and GPP resilience were likely a result of flow-induced changes to physicochemical parameters (e.g., turbidity) that can also enhance the physical disturbance of flow on GPP (O'Donnell and
230 Hotchkiss, 2019). For instance, sustained periods of high turbidity following a flow disturbance can prolong the recovery of GPP by inhibiting light attenuation (Blaszczak et al., 2019). In contrast, higher resilience of ER is likely a function of greater resistance of ER to disturbances (i.e., smaller M; Table 3) as well as flow-induced ER stimulation. The correlation of $M_{GPP}$ and $M_{ER}$, but a lack of correlation between $RI_{GPP}$ and $RI_{ER}$ (Figure 5), suggests GPP and ER were temporarily decoupled while recovering, despite similar initial responses of GPP and ER to flow disturbances.

The dynamic nature of stream metabolism, even during low flow periods, must guide how we quantify metabolic responses to disturbance. When we quantified resistance as a deviation from an antecedent average (e.g., Reisinger et al. 2017; Roley et al. 2014), few flow events yielded 100% resistance and every metabolism estimate was classified as a reduction or stimulation. Without acknowledging the "pulsing steady state" of stream metabolism, we may overestimate disturbances in ecosystem function. In assessing metabolic responses and recovery from smaller flow events relative to the dynamic equilibrium of metabolism
at baseflow, we found some of the shortest metabolic recovery intervals recorded in the literature (Figure 8; Table A1). Incorporating the pulsing equilibrium of metabolism and standardizing calculations of metabolic recovery dynamics will enable more robust, cross-site comparisons of complex ecosystem response to changes in flow.

### 4.2   Controls on metabolic resistance and resilience after a flow disturbance

While GPP responded similarly to flows regardless of magnitude, ER was more resistant to smaller magnitude isolated flow
events. Our hypothesis that isolated flow events of greater magnitudes (i.e., larger % change in cumulative daily discharge) would result in less resistance and higher $M_{GPP}$ and $M_{ER}$, due to increased scouring, was supported only for $M_{ER}$. GPP appears to have low resistance to flow disturbances, regardless of magnitude (Table 4, Figure 8; Reisinger et al. 2017; Roley et al. 2014). The different responses of GPP and ER to variable flow may be attributed to differences in energy sources and locations





of autotrophs and heterotrophs (Uehlinger, 2000, 2006). Primary producers reside in exposed areas on the streambed to access
light required for photosynthesis, and are thus more vulnerable to scour than heterotrophic biofilms tucked within, and protected
by, substrates in the streambed, sediments, and hyporheic zone (Uehlinger, 2000). At some threshold of higher flows that disturb
more protected areas within and below streambeds, we expect ER will decline as flow-induced stress exceeds flow-induced
carbon and nutrient subsidies. Future analyses that include event duration may also provide new insights into flow-metabolism
dynamics: Do sustained, higher flows change GPP and ER in the same way as a more instantaneous, intense flow event? As is
255 common of long-term characterizations of metabolism in streams, many high flow days had metabolism model outputs that did
not hold up to quality checks and thus were not included in our analyses. Overcoming the logistic and computational challenges
of estimating metabolism during extreme flows that disturb deeper substrates will allow us to better test predictions relating
flow magnitude with ecosystem functions.

Contrary to our predictions, the size of the most recent antecedent flow disturbance had a positive relationship with $M_{GPP}$ and
260 $M_{ER}$ (Figure A19). $M_{GPP}$ was smaller and GPP was more resistant when the most recent flow events were larger. Similarly, the
% change in cumulative daily discharge from the last event was positively correlated with $M_{ER}$. Stream biota still recovering
and regenerating biomass lost from scour might respond differently to flow events depending on successional stage (Peterson
and Stevenson, 1992). Furthermore, biomass growth initially stimulated by a preceding event may have been limited by one
or more nutrients later supplied by the isolated flow event. We ultimately do not know what caused the unexpected negative
relationships between the magnitude of the most recent event and $M_{GPP}$ as well as $M_{ER}$ in Stroubles Creek; quantifying the
interactions between successional recovery and changes in nutrient limitation across multiple flow events may elucidate this
interesting relationship.

Environmental conditions on the day of isolated flow events that promote biomass growth, such as high light and temperature,
were not significant predictors of ER or GPP recovery intervals. Metabolic recovery trajectories often increase with temperature
and PAR (Uehlinger and Naegeli, 1998; Uehlinger, 2000), and consequently may also display a seasonal component, with faster
recoveries in spring and slower recoveries in winter (Uehlinger, 2000, 2006). We found a negative relationship between $RI_{GPP}$
and both temperature and season (Figure A20), suggesting that under colder conditions and in winter months, GPP takes longer
to recover. While we did not find strong predictors of $RI_{ER}$ among the environmental variables in our dataset, changes in the
source, magnitude, and biological lability of organic matter inputs may alter $RI_{ER}$ (Roberts et al., 2007). How do changes
in nutrients and organic matter govern the resistance and resilience of GPP and ER? Combining high-frequency nutrient and
organic matter quality measurements with metabolic resistance and resilience estimates will offer an improved understanding
of how changing resources mediate metabolic responses to flow changes.

We did not find evidence to support a single subsidy-stress response of metabolic changes during and after high flow events
(Figure 1). ER was more resistant than GPP to most flow disturbances (H1). At small-intermediate sized flow disturbances,
the response of metabolism was variable (H2,H3), with the greatest range of metabolic stimulation or reduction (i.e., subsidy
or stress) observed at smaller flow changes (Figure 7). ER and GPP did not increase or decrease relative to their antecedent
"pulsing equilibrium" during several high flow events (H0). Light, temperature, or turbidity may control the variability in



metabolic response to smaller flow disturbances. With increasing intensity of flow disturbance, stress and replacement may indeed scale with intensity (H3).

## 285  5   Conclusions

Metabolic regimes are punctuated by high flow events that create frequent pulses of stimulated or repressed GPP or ER (e.g., Uehlinger 2006; Beaulieu et al. 2013; Bernhardt et al. 2018). As such, changes in flow play an influential role in the trends and variability in daily metabolism. While geomorphology and disturbance regimes may control metabolic resistance across sites (Uehlinger, 2000; Blaszczak et al., 2019), within-site variability of M and RI may be controlled by the characteristics of each

flow event. Differences between ER and GPP response and recovery to flow disturbances at our study site were controlled by higher resistance and resilience of ER relative to GPP. Within this study, our prediction that ER would be more resistant than GPP to flow disturbances was supported, as ER frequently did not even deviate from the antecedent equilibrium. However, ER had less resistance to events of greater magnitude: $M_{ER}$ was negatively correlated with the % change in discharge of flow event, whereas $M_{GPP}$ was not, suggesting that GPP responded similarly to changes in flow regardless of flow magnitude. Metabolic

response to small and intermediate flow disturbances was variable: GPP and ER were both stimulated and suppressed. We suggest there may be a resistance threshold to flow disturbances, where controls other than flow magnitude (e.g., season, light, turbidity) might regulate metabolic responses to lower flow changes. Using segmented process-discharge relationships to quantify a resistance threshold of processes to flow disturbances (O'Donnell and Hotchkiss, 2019) may support a more predictive understanding of metabolic response to flow disturbances.

Ultimately, we are entering an era of metabolic data opportunity (e.g., Bernhardt et al. 2018). As time series of metabolism lengthen and modeling tools improve, we envision exciting opportunities to better assess the consequences of isolated flow events as well as the impacts of multiple, sequential high flow disturbances that did not meet our criteria for analyzing isolated flow events in this paper. While the short time periods between high flow events in many streams and rivers make isolating and quantifying functional resistance and resilience an ongoing challenge, including dynamic flow in our assessment of metabolic

regimes is a critical next step toward a more holistic understanding of frequently disturbed ecosystems.

*Code and data availability.*  All data and results are included in the Appendices of this paper. Supplemental data files and metadata (not embedded in this document) can be downloaded at https://tinyurl.com/RMetab

*Author contributions.*  BO developed the ideas for this manuscript with ERH. BO led the data analyses and development of resistance/resilience indices with ERH. BO and ERH wrote and edited the manuscript.

*Competing interests.*  No competing interests





*Acknowledgements.* We thank W.C. Hession for sharing StREAM Lab sensor data and his knowledge of Stroubles Creek with us and L. Lehmann for assistance with database access and questions. S. Rahman assisted with fieldwork and preliminary data analyses. We acknowledge D. McLaughlin for helpful conversations and edits throughout this project. This work was supported by Virginia Tech's Department of Biological Sciences and an Endowment Award to BO from the Society for Freshwater Science.





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





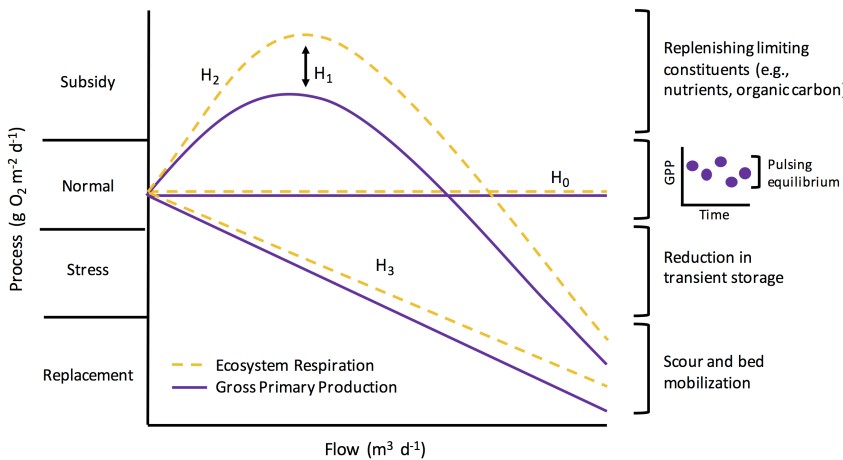

**Figure 1.** Potential metabolic responses along a subsidy-stress gradient of stream flow (adopted from Odum et al. 1979). Flow is on the x-axis. The y-axis represents ecosystem metabolism (i.e., gross primary production and ecosystem respiration; GPP and ER), scaled to the same "normal" starting values for comparison, and is broken into four categories as proposed by Odum et al. (1979): (1) subsidy (when subsidization dominates and metabolism increases), (2) normal (periods of metabolic pulsing equilibrium under ambient flow), (3) stress (when ecosystem processes are repressed by disturbance), and (4) replacement (when there is a severe reduction in metabolism and communities are scoured or replaced). The inset graph next to the 'normal' bracket depicts how ambient process rates are best represented by a pulsing equilibrium rather than a fixed point of stability (*sensu* Odum et al. 1995).





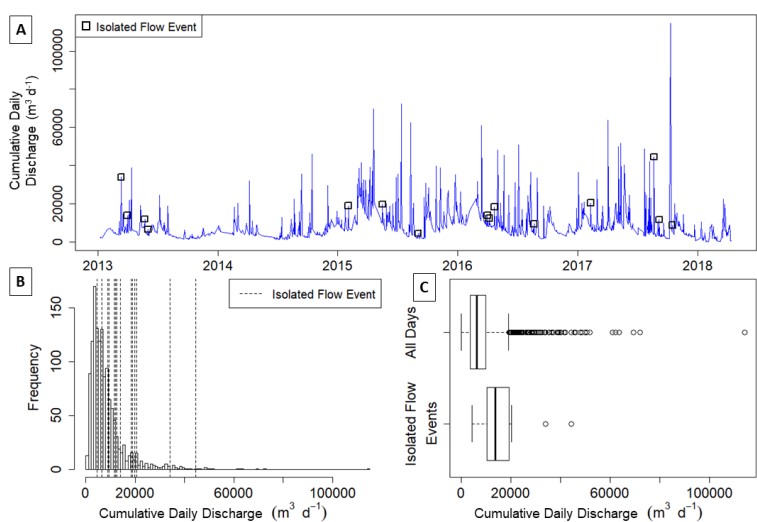

**Figure 2.** (A) Time series of cumulative daily discharge ($m^3$ $d^{-1}$) on all days with quality-checked metabolism estimates from 2013-01-08 to 2018-04-14. The 15 isolated flow events analyzed for metabolic responses to higher flow are represented by open squares. (B) Frequency distribution of cumulative daily discharge for days with quality-checked metabolism estimates. Vertical dashed lines denote the cumulative daily discharge values of the 15 different isolated flow events. (C) Box plots of cumulative daily discharge (($m^3$ $d^{-1}$)) for all days with metabolism estimates versus from isolated flow event days that fit our criteria for analyzing metabolic resistance and resilience.





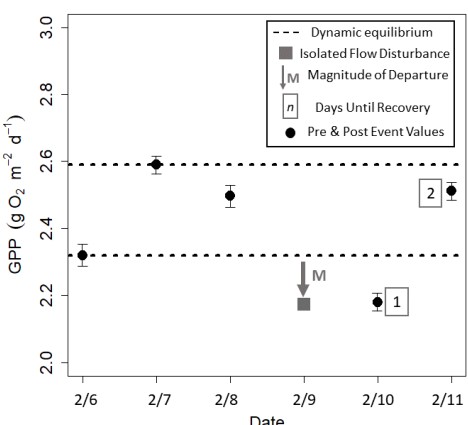

**Figure 3.** Example calculations of metabolic resistance (M) and resilience (RI). Daily gross primary production (GPP) was estimated for the three days before, one day during (grey square), and two days following an isolated flow event that occurred on 2017-02-09. Dashed horizontal lines represent the maximum and minimum GPP estimates from three days prior to the flow event. In this case, GPP was repressed, and the magnitude of departure (M with grey arrow) is the difference between minimum GPP estimate from the antecedent range (bottom dashed line) and GPP during the event. After this flow event, GPP recovered to antecedent range in 2 days; recovery interval days (RI) are represented by the grey boxed numbers next to GPP estimates.



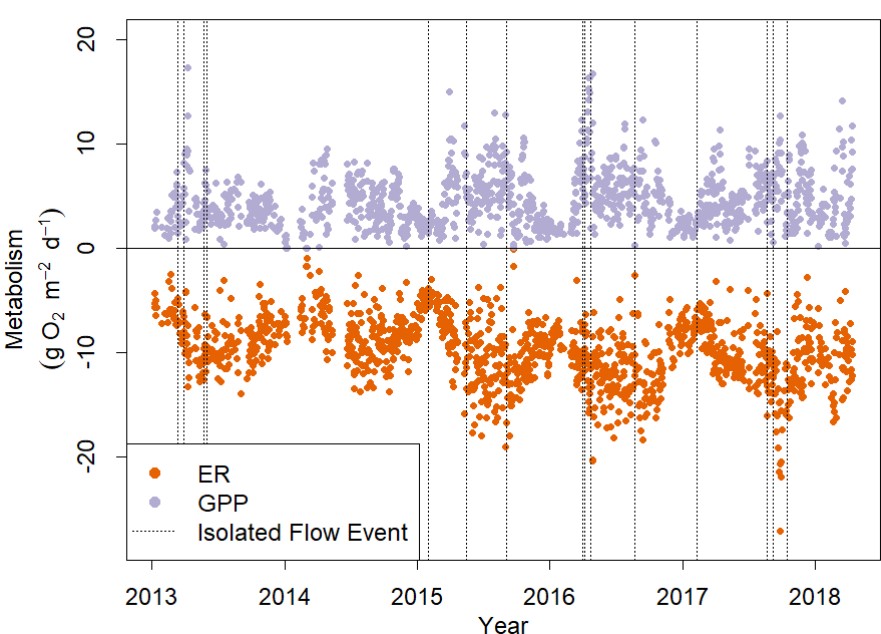

**Figure 4.** Gross primary production (GPP, top) and ecosystem respiration (ER, bottom) in Stroubles Creek, VA from 2013-01-08 to 2018-04-14. ER is represented here as a negative rate because it is the consumption of oxygen. Dashed vertical lines mark the isolated flow events that fit our criteria for analyzing metabolic responses to flow change (Figure 2).

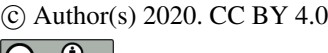


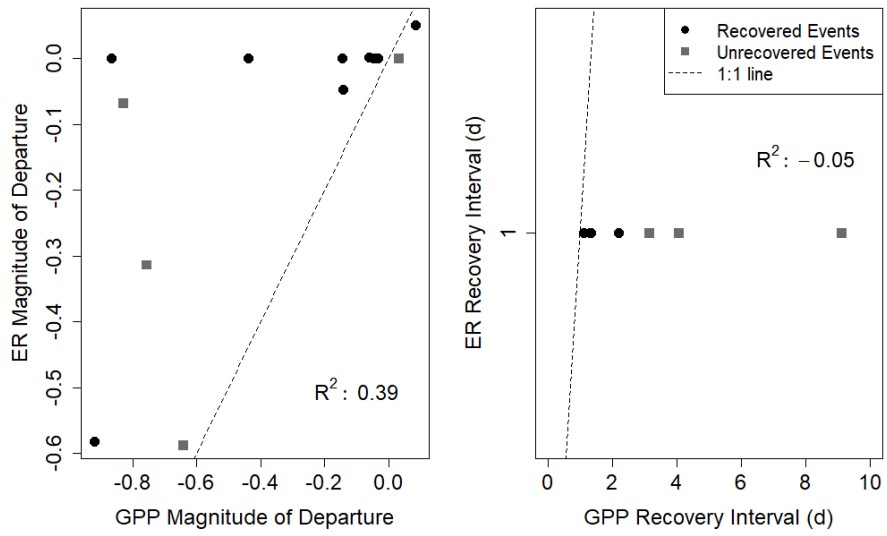

**Figure 5.** (A) Resistance (i.e., magnitude of departure; left) and (B) resilience (i.e., recovery interval; right) of gross primary production (GPP) versus ecosystem respiration (ER) in Stroubles Creek, VA. Dashed lines are 1:1 lines. Black circles are isolated flow events that did recover to antecedent equilibrium, or were unresponsive (i.e., M = 0). Grey squares are the isolated flow events that did not recover to antecedent equilibrium before another flow event occurred. RI values are jittered along the x-axis (the points are slightly scattered around their value to visualize overlapping values); each recovery interval was calculated at the daily scale.





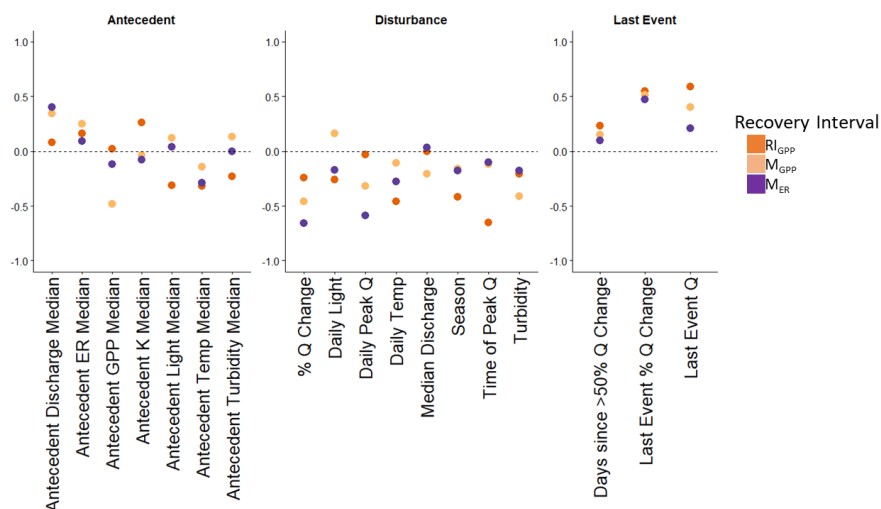

**Figure 6.** Pearson correlation coefficients for tested controls on ecosystem respiration magnitude of departure ($M_{ER}$, gross primary production magnitude of departure ($M_{GPP}$), and recovery interval of GPP ($RI_{GPP}$). Variables are divided into three groups (left-right): median antecedent conditions, disturbance characteristics, and characteristics of the most recent flow event. For units of variables, see Table 4.





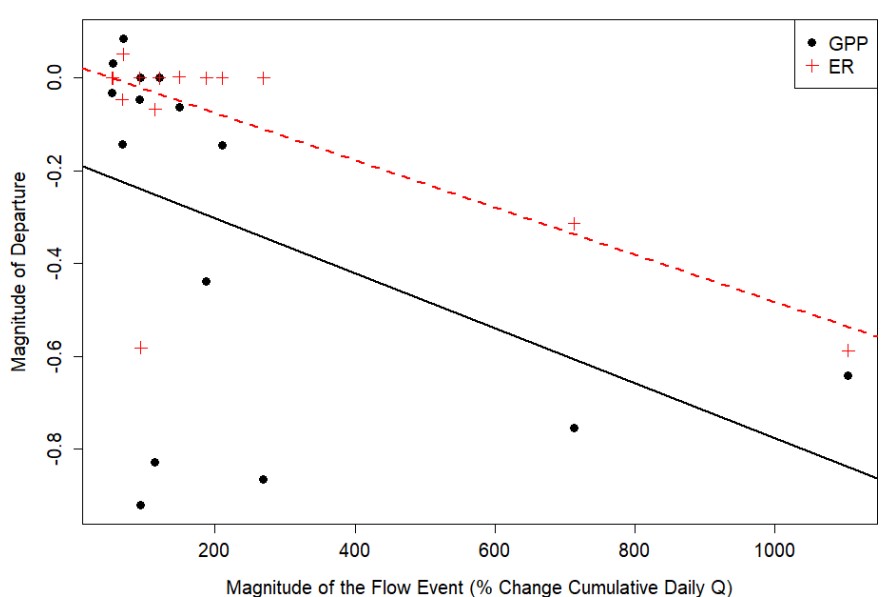

**Figure 7.** Flow event magnitude (% change in cumulative daily discharge relative to the day prior) was negatively correlated with magnitude of departure (M) for gross primary production (GPP; $R^2$ = 0.15, r = -0.46, p = 0.08) and ecosystem respiration (ER; $R^2$ = 0.39, r = -0.66, p = 0.01, $\alpha$ = 0.05). The solid black line is the regression for the relationship between $M_{GPP}$ and % change in discharge, while the dashed red line is the regression for the relationship between $M_{ER}$ and % change in discharge. % change in cumulative daily discharge was significantly negatively correlated with $M_{ER}$, but not significantly correlated with $M_{GPP}$.





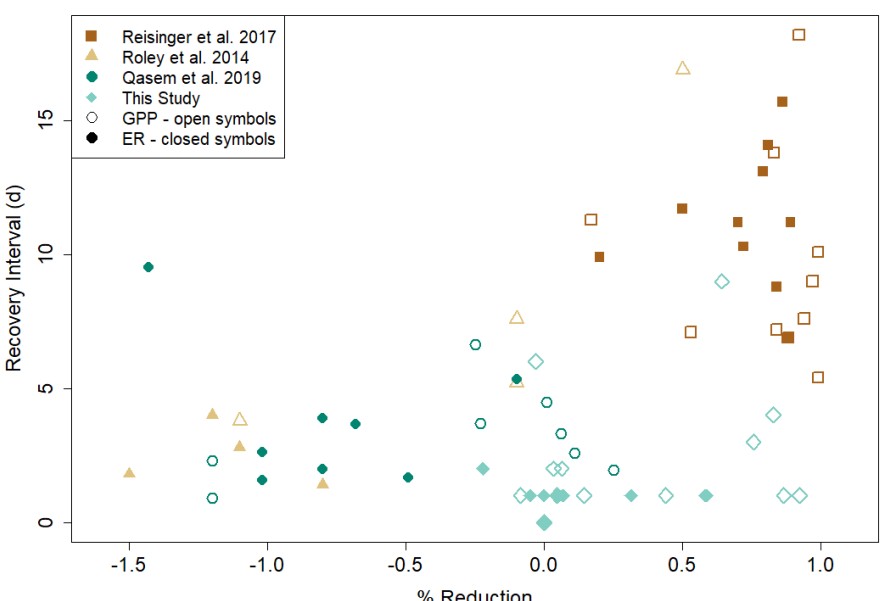

**Figure 8.** A synthesis of metabolic recovery intervals (days) and % reduction of gross primary production (GPP) and ecosystem respiration (ER) in response to flow disturbances. Open symbols represent GPP response, and closed symbols signify ER response. A negative % reduction is a stimulation. Included in Supplemental Table 1 are additional studies that have reported either recovery intervals, or % metabolic reduction in response to flow disturbances, but not both, and consequently could not be included here.





**Table 1.** Parameter symbols, descriptions, and units used in Equation 1

| Parameter symbol | Parameter description (units) |
| --- | --- |
| $mDO$ | Modeled $O_2$ (g $O_2$ m$^{-3}$) |
| $\Delta t$ | Measurement interval (d) |
| $GPP$ | Gross primary production (g $O_2$ m$^{-2}$ d$^{-1}$) |
| $ER$ | Ecosystem respiration (g $O_2$ m$^{-2}$ d$^{-1}$) |
| $z$ | Mean stream channel depth (m) |
| $K_O$ | Air-water gas exchange of $O_2$ (d$^{-1}$) |
| $O_{sat}$ | DO at saturation (g $O_2$ m$^{-3}$) |
| $PAR$ | Photsynthetically active radiation ($\mu$mol m$^{-2}$ s$^{-1}$) |





**Table 2.** Cumulative daily discharge (CDQ), % change in CDQ relative to the prior day, metabolism (gross primary production (GPP), ecosystem respiration (ER)), and air-water gas exchange (K) of isolated flow events analyzed for metabolic recovery

| Date | CDQ ($m^3$ $d^{-1}$) | %Δ CDQ | GPP (g $O_2$ $m^{-2}$ $d^{-1}$) | ER (g $O_2$ $m^{-2}$ $d^{-1}$) | K ($d^{-1}$) |
|---|---|---|---|---|---|
| 2013-03-12 | 33,970 | 713 | 1.5 | -4.8 | 9.0 |
| 2013-03-31 | 13,849 | 188 | 2.4 | -8.0 | 13.0 |
| 2013-05-23 | 11,923 | 69 | 3.7 | -12.6 | 15.2 |
| 2013-06-02 | 6,545 | 93 | 3.2 | -10.6 | 13.1 |
| 2015-02-02 | 18,842 | 210 | 1.4 | -4.7 | 20.9 |
| 2015-05-17 | 19,683 | 94 | 7.2 | -13.5 | 15.9 |
| 2015-09-03 | 4,447 | 120 | 6.5 | -11.2 | 12.8 |
| 2016-04-01 | 13,869 | 67 | 4.8 | -7.6 | 13.9 |
| 2016-04-07 | 12,478 | 53 | 5.0 | -9.7 | 19.2 |
| 2016-04-22 | 18,340 | 114 | 1.9 | -10.4 | 13.0 |
| 2016-08-21 | 9,418 | 94 | 0.3 | -2.6 | 4.8 |
| 2017-02-09 | 20,383 | 149 | 2.2 | -7.4 | 17.6 |
| 2017-08-21 | 44,543 | 1,105 | 2.5 | -4.3 | 4.1 |
| 2017-09-06 | 11,600 | 269 | 0.6 | -12.1 | 17.3 |
| 2017-10-16 | 8,761 | 54 | 3.4 | -11.4 | 17.8 |





**Table 3.** Magnitude of departure (M, unitless) and recovery intervals (RI, days) of gross primary production (GPP) and ecosystem respiration (ER) during and after fifteen isolated flow events between 2013-01-08 and 2018-04-14. A negative M represents a suppression, and a positive M a stimulation, where GPP or ER increase relative to the antecedent equilibrium. If GPP did not recover before the next flow event, RI is noted with a "*" and those events are not included in recovery interval correlation analyses.

| Date | $M_{GPP}$ | $RI_{GPP}$ (d) | $M_{ER}$ | $RI_{ER}$ (d) |
|---|---|---|---|---|
| 2013-03-12 | -0.76 | 3+* | -0.31 | 1* |
| 2013-03-31 | -0.44 | 1 | 0.00 | n/a |
| 2013-05-23 | 0.09 | 1 | 0.05 | 1 |
| 2013-06-02 | -0.05 | 1 | 0.00 | n/a |
| 2015-02-02 | -0.14 | 1 | 0.00 | n/a |
| 2015-05-17 | 0.00 | n/a | 0.22 | 2 |
| 2015-09-03 | 0.00 | n/a | 0.00 | n/a |
| 2016-04-01 | -0.14 | 1 | -0.05 | 1 |
| 2016-04-07 | -0.03 | 2 | 0.00 | n/a |
| 2016-04-22 | -0.83 | 4+* | -0.07 | 1* |
| 2016-08-21 | -0.92 | 1 | -0.58 | 1 |
| 2017-02-09 | -0.06 | 2 | 0.002 | 1 |
| 2017-08-21 | -0.64 | 9+* | -0.59 | 1* |
| 2017-09-06 | -0.87 | 1 | 0.00 | n/a |
| 2017-10-16 | 0.03 | 6+* | 0.00 | n/a* |





**Table 4.** Pearson correlations (r) between predicted drivers of gross primary production (GPP) and ecosystem respiration (ER) magnitudes of departure (M) and recovery intervals (RI) of isolated flow events. Predictor variables with moderate or stronger relationships (r > 0.5) are bolded.

| Predictor variable (units) | r, $RI_{GPP}$ | r, $M_{GPP}$ | r, $RI_{ER}$ | r, $M_{ER}$ |
|---|---|---|---|---|
| **Isolated flow event of interest** | | | | |
| Daily median photosynthetic active radiation ($\mu$mol m$^{-2}$ s$^{-1}$) | -0.26 | 0.16 | n/a | -0.17 |
| Daily peak discharge (m$^{-3}$ s$^{-1}$) | -0.03 | -0.32 | n/a | **-0.59** |
| Daily median temperature (°C) | -0.46 | -0.11 | n/a | -0.28 |
| Event median discharge (m$^{-3}$ s$^{-1}$) | 0.00 | -0.21 | n/a | 0.03 |
| Change in discharge during event (%) | -0.24 | -0.46 | n/a | **-0.66** |
| Season | -0.42 | -0.16 | n/a | -0.18 |
| Time of peak discharge (HH:MM) | **-0.65** | -0.12 | n/a | -0.10 |
| Turbidity (nephelometric turbidity unit, NTU) | -0.21 | -0.41 | n/a | -0.18 |
| **Most recent flow event** | | | | |
| Days since last event (d) | 0.23 | 0.15 | n/a | 0.10 |
| Last event cumulative daily discharge (m$^{-3}$ s$^{-1}$) | **0.59** | 0.40 | n/a | 0.21 |
| Change in discharge during last event (%) | **0.55** | **0.51** | n/a | 0.47 |
| **Antecedent conditions** | | | | |
| Antecedent median gas exchange (d$^{-1}$) | 0.26 | -0.04 | n/a | -0.08 |
| Antecedent gross primary production (g O$_2$ m$^{-2}$ d$^{-1}$) | 0.02 | -0.48 | n/a | -0.12 |
| Antecedent ecosystem respiration (g O$_2$ m$^{-2}$ d$^{-1}$) | 0.16 | 0.25 | n/a | 0.09 |
| Antecedent median photosynthetic active radiation ($\mu$mol m$^{-2}$ s$^{-1}$) | -0.31 | 0.12 | n/a | 0.04 |
| Antecedent median discharge (m$^{-3}$ s$^{-1}$) | 0.08 | 0.34 | n/a | 0.40 |
| Antecedent median water temperature (°C) | -0.32 | -0.14 | n/a | -0.29 |
| Antecedent median turbidity (nephelometric turbidity unit, NTU) | -0.23 | 0.13 | n/a | 0.00 |

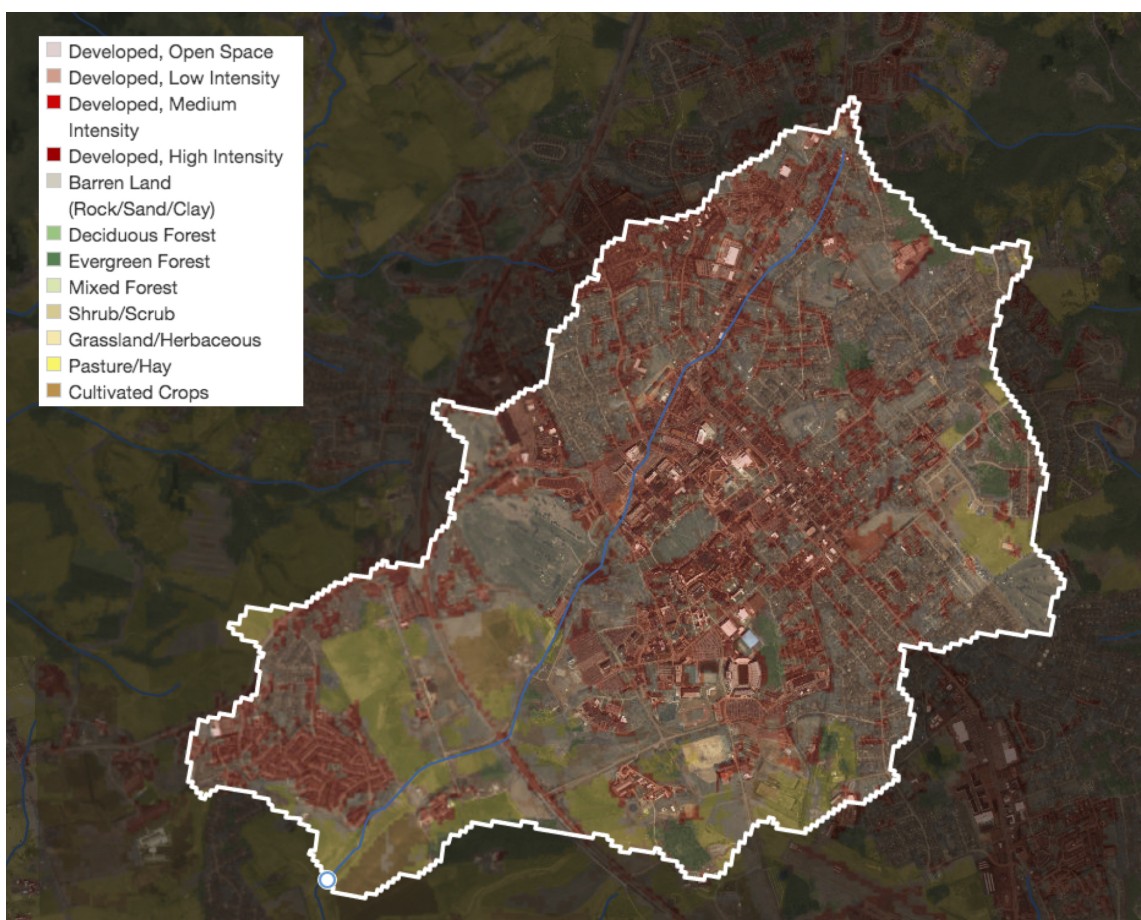

**Figure A1.** Stroubles Creek watershed and land cover types in the area that drains to the StREAM Lab monitoring site at Bridge 1. Blacksburg, VA, U.S.A. We created this map using ArcGIS, NHDplus version 2.1, and the U.S. Geological Survey's 2011 National Land Cover Database.





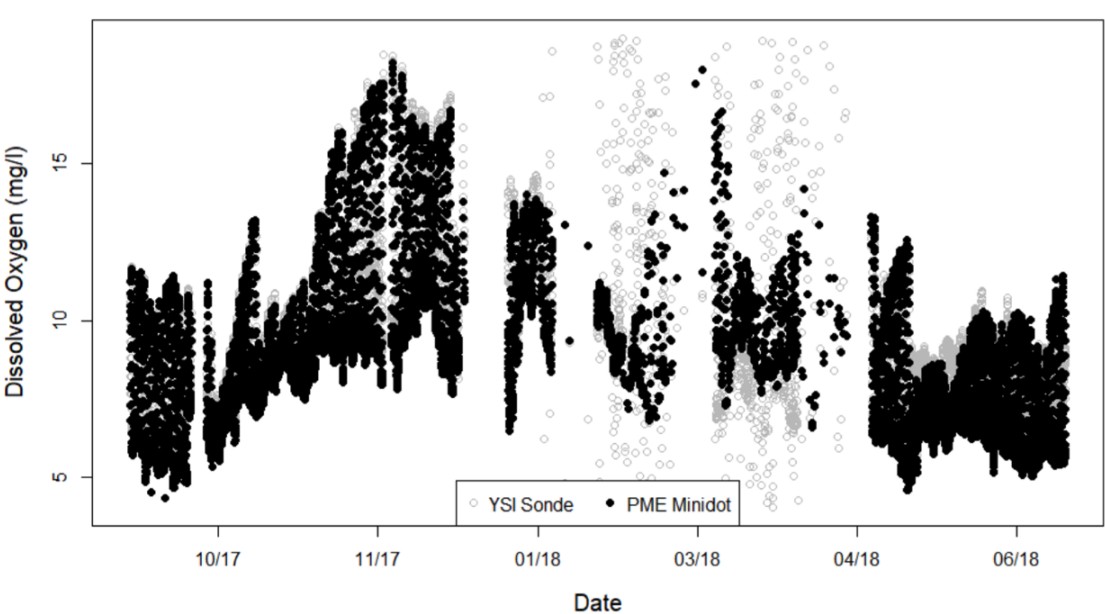

**Figure A2.** Dissolved oxygen measurements from the two sensors – YSI Sonde and PME Minidot - at Bridge 1 on Stroubles Creek. The spread of YSI Sonde values spanning from the end of January to mid-April was likely a result of a freeze event. We used PME data during the period of record when YSI data did not pass our quality assurance checks.

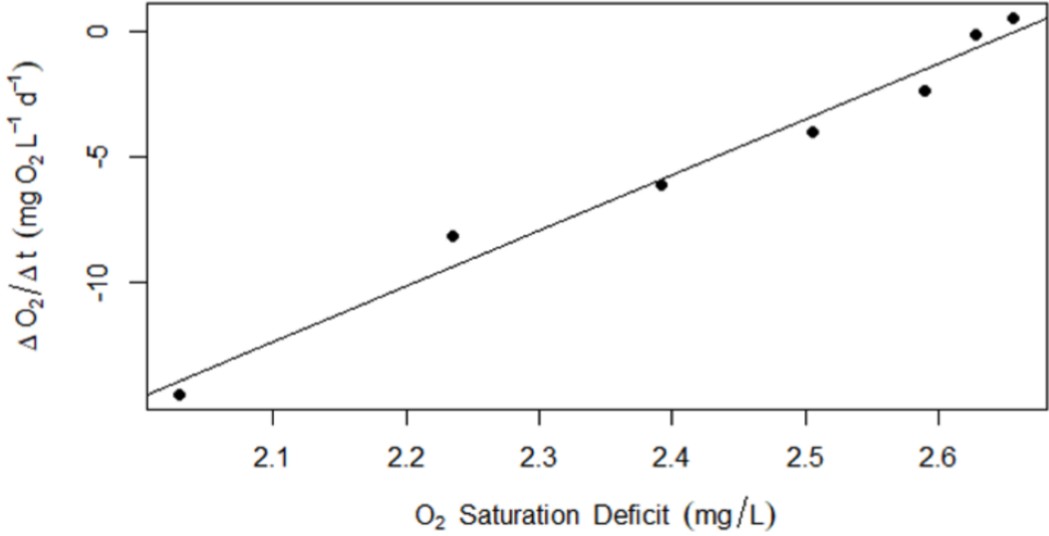

**Figure A3.** Example of data used to confirm modeled $K_{600}$ ($d^{-1}$) using a regression of the nighttime dissolved oxygen saturation deficit versus changes in saturation (as in Hall and Hotchkiss 2017). These data are from 2017-09-04, when the estimated value for $K_{600}$ was 22 $d^{-1}$.





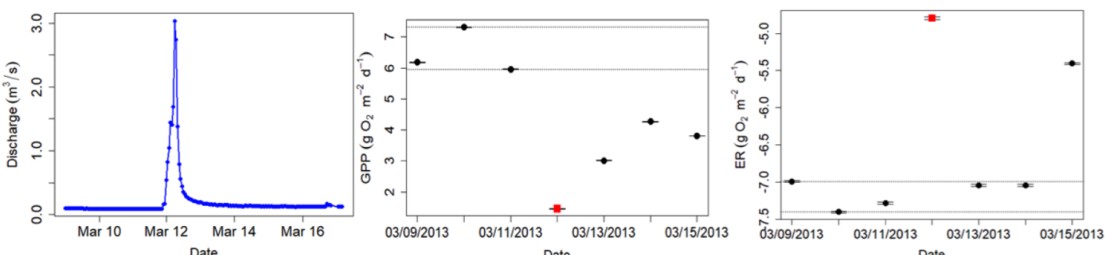

**Figure A4.** Hydrograph and metabolism (gross primary production; GPP and ecosystem respiration; ER) time series for the Stroubles Creek flow event on 2013-03-12.

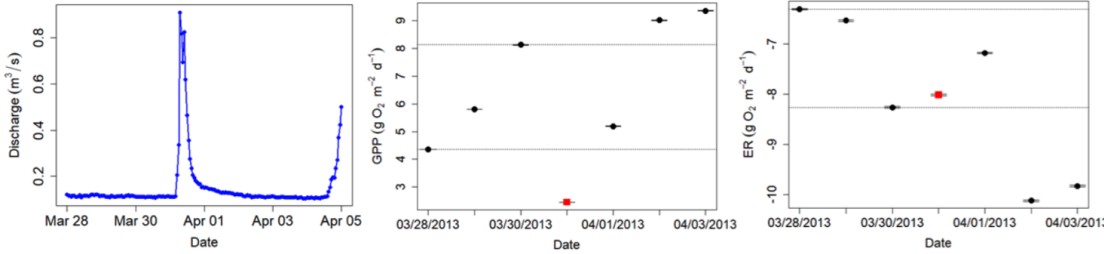

**Figure A5.** Hydrograph and metabolism (gross primary production; GPP and ecosystem respiration; ER) time series for the Stroubles Creek flow event on 2013-03-31.

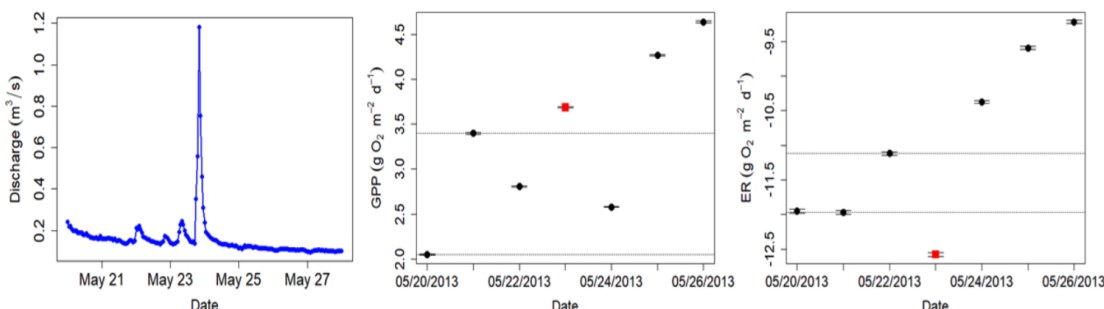

**Figure A6.** Hydrograph and metabolism (gross primary production; GPP and ecosystem respiration; ER) time series for the Stroubles Creek flow event on 2013-05-23.



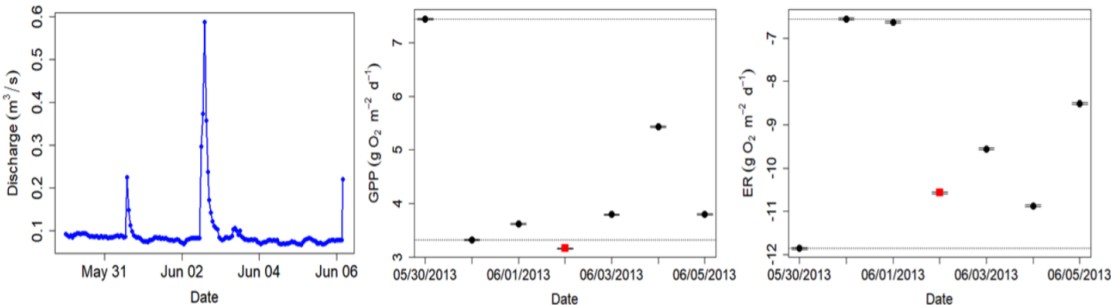

**Figure A7.** Hydrograph and metabolism (gross primary production; GPP and ecosystem respiration; ER) time series for the Stroubles Creek flow event on 2013-06-02.

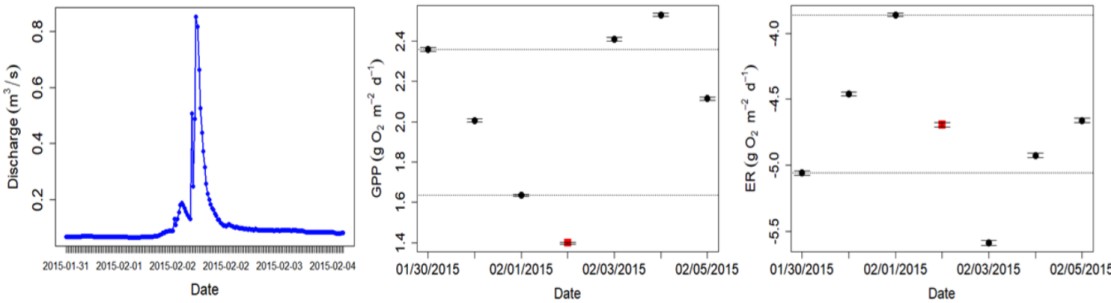

**Figure A8.** Hydrograph and metabolism (gross primary production; GPP and ecosystem respiration; ER) time series for the Stroubles Creek flow event on 2015-02-02.

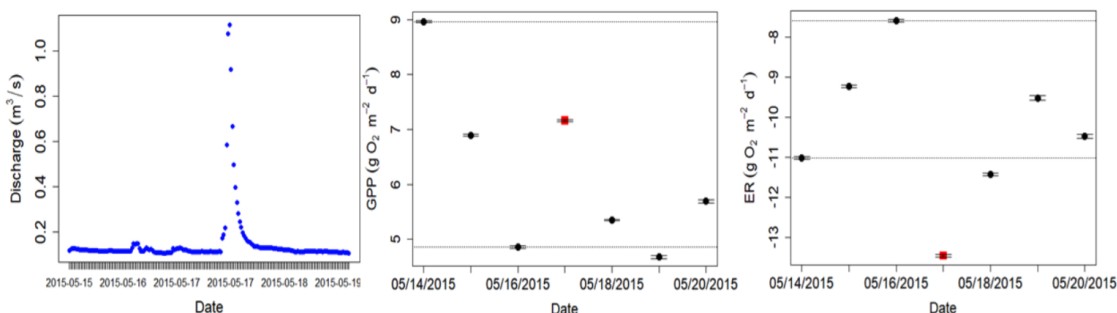

**Figure A9.** Hydrograph and metabolism (gross primary production; GPP and ecosystem respiration; ER) time series for the Stroubles Creek flow event on 2015-05-17.





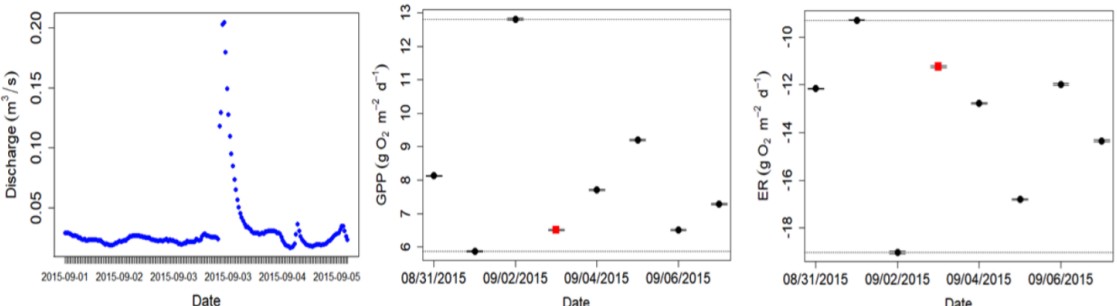

**Figure A10.** Hydrograph and metabolism (gross primary production; GPP and ecosystem respiration; ER) time series for the Stroubles Creek flow event on 2015-09-03.

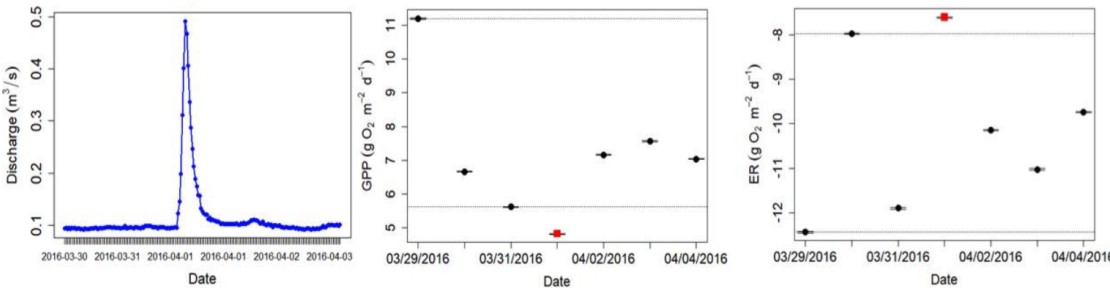

**Figure A11.** Hydrograph and metabolism (gross primary production; GPP and ecosystem respiration; ER) time series for the Stroubles Creek flow event on 2016-04-01.

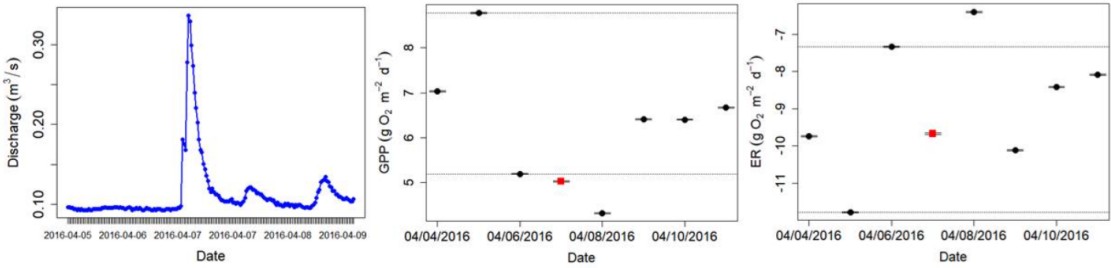

**Figure A12.** Hydrograph and metabolism (gross primary production; GPP and ecosystem respiration; ER) time series for the Stroubles Creek flow event on 2016-04-07.





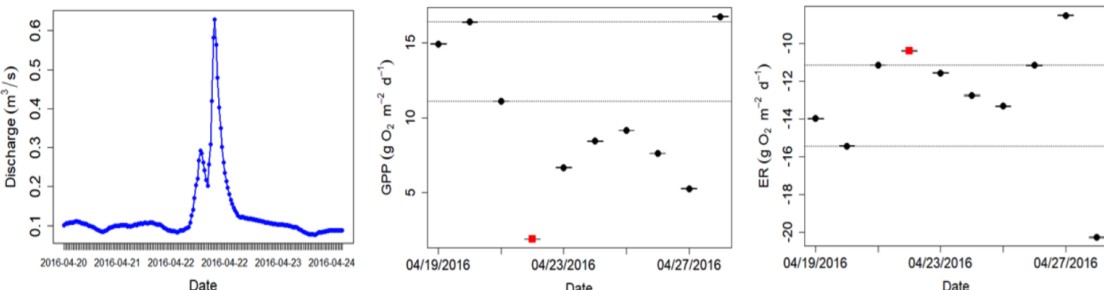

**Figure A13.** Hydrograph and metabolism (gross primary production; GPP and ecosystem respiration; ER) time series for the Stroubles Creek flow event on 2016-04-22.

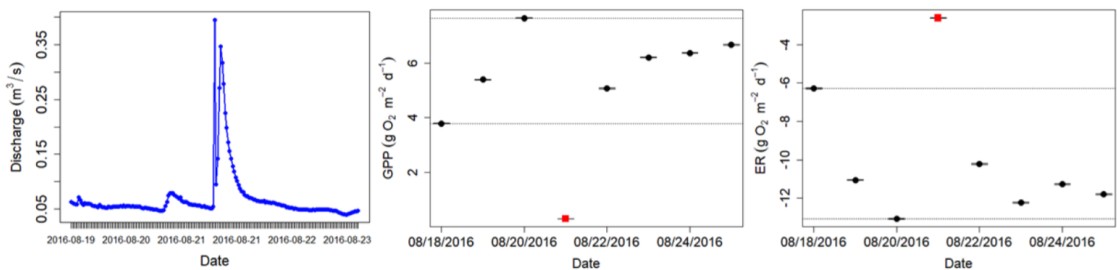

**Figure A14.** Hydrograph and metabolism (gross primary production; GPP and ecosystem respiration; ER) time series for the Stroubles Creek flow event on 2016-08-21.

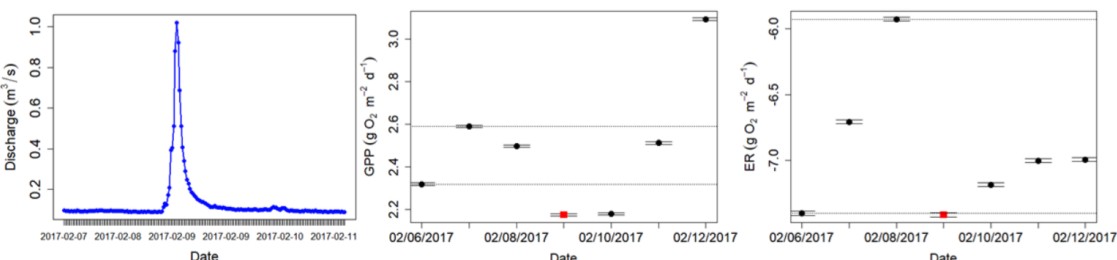

**Figure A15.** Hydrograph and metabolism (gross primary production; GPP and ecosystem respiration; ER) time series for the Stroubles Creek flow event on 2017-02-09.



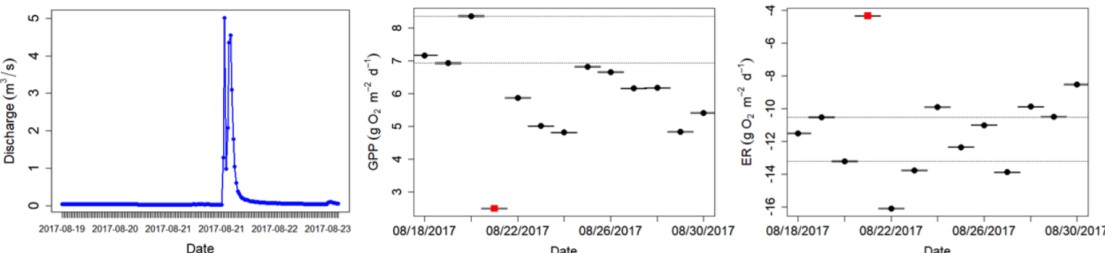

**Figure A16.** Hydrograph and metabolism (gross primary production; GPP and ecosystem respiration; ER) time series for the Stroubles Creek flow event on 2017-08-21.

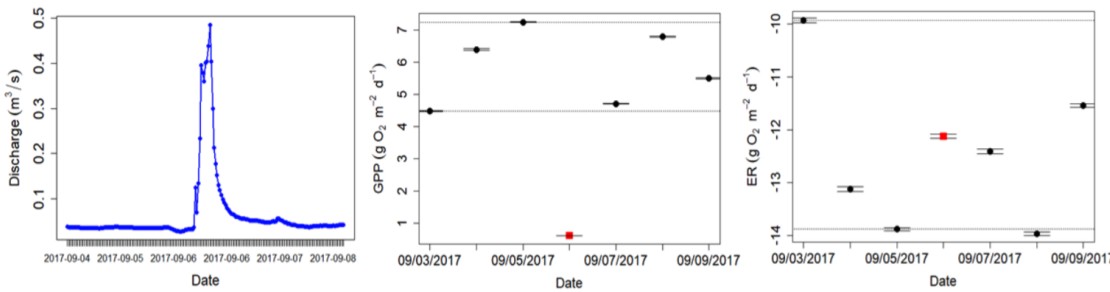

**Figure A17.** Hydrograph and metabolism (gross primary production; GPP and ecosystem respiration; ER) time series for the Stroubles Creek flow event on 2017-09-06.

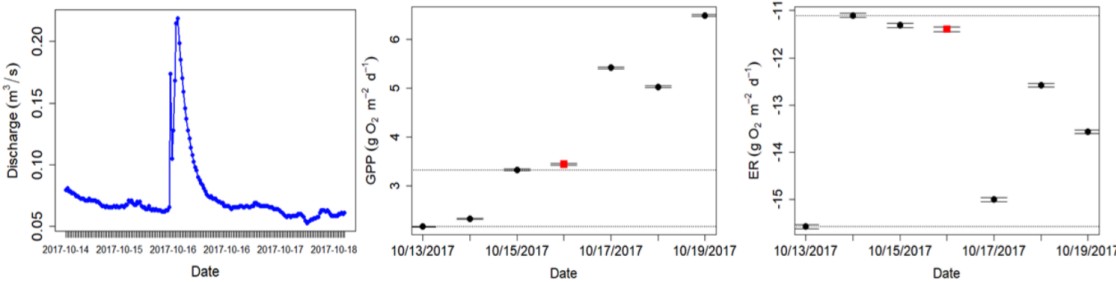

**Figure A18.** Hydrograph and metabolism (gross primary production; GPP and ecosystem respiration; ER) time series for the Stroubles Creek flow event on 2017-10-16.




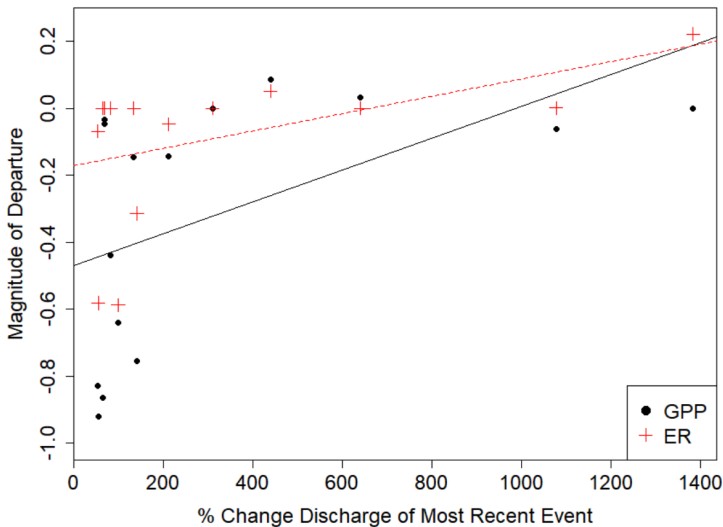

**Figure A19.** The magnitude of the previous high flow event (% change in cumulative daily discharge) had a positive relationship with $M_{GPP}$ and $M_{ER}$. GPP is represented by filled black circles; ER by red crosses. The black, solid regression line reflections the relationship between magnitude of the last event and $M_{GPP}$, whereas the dashed, red regression line represents the relationship between the magnitude of the last event and $M_{ER}$.

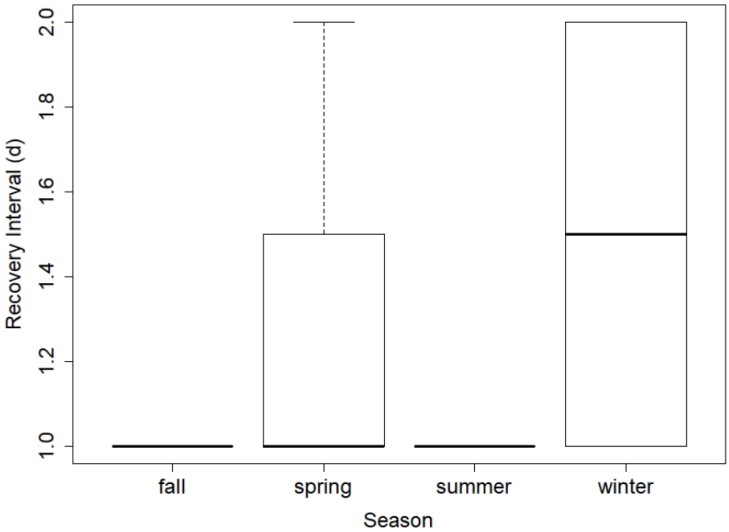

**Figure A20.** Recovery intervals of gross primary production ($RI_{GPP}$; days) grouped by season.





**Table A1.** Literature review of published reduction and recovery intervals (RI) of stream gross primary production (GPP) and ecosystem respiration (ER) after high flow events. If not enough information was given to calculate reduction or RI, we listed as "n/a". *=approximated days of recovery from figure in publication. **=approximation given in publication.

| Source | Reduction in GPP (%) | Reduction in ER (%) | $RI_{GPP}$ (days) | $RI_{ER}$ (days) |
|---|---|---|---|---|
| Uehlinger and Naegeli 1998 | 0.53 | 0.24 | n/a | n/a |
| Uehlinger 2000 | 0.53 | 0.37 | n/a | n/a |
| Uehlinger 2000 | 0.37 | 0.14 | n/a | n/a |
| Uehlinger 2006 | 0.49 | 0.19 | n/a | n/a |
| Roberts et al. 2007 | 0.90** | n/a | 5 | 5 |
| Roberts et al. 2007 | n/a | n/a | 4* | 4 |
| Roley et al. 2014 | -1.1 | -1.1 | 3.8 | 2.8 |
| Roley et al. 2014 | -0.1 | -1.5 | 5.2 | 1.8 |
| Roley et al. 2014 | 0.5 | -0.8 | 16.9 | 1.4 |
| Roley et al. 2014 | -0.1 | -1.2 | 7.6 | 4.0 |
| Smith and Kaushal 2015 | 0.50** | n/a | 14-21 | n/a |
| Reisinger et al. 2017 | 0.92 | 0.86 | 18.2 | 15.7 |
| Reisinger et al. 2017 | 0.84 | 0.72 | 7.2 | 10.3 |
| Reisinger et al. 2017 | 0.99 | 0.88 | 5.4 | 6.9 |
| Reisinger et al. 2017 | 0.99 | 0.81 | 10.1 | 14.1 |
| Reisinger et al. 2017 | 0.53 | 0.89 | 7.1 | 11.2 |
| Reisinger et al. 2017 | 0.94 | 0.79 | 7.6 | 13.1 |
| Reisinger et al. 2017 | 0.71 | 0.11 | 4.3 | no recovery |
| Reisinger et al. 2017 | 0.88 | 0.70 | 6.9 | 11.2 |
| Reisinger et al. 2017 | 0.97 | 0.84 | 9.0 | 8.8 |
| Reisinger et al. 2017 | 0.83 | 0.20 | 13.8 | 9.9 |
| Reisinger et al. 2017 | 0.17 | 0.50 | 11.3 | 11.7 |
| Qasem et al. 2019 | 0.06 | -0.49 | 3.3 | 1.7 |
| Qasem et al. 2019 | -0.25 | -0.68 | 6.7 | 3.7 |
| Qasem et al. 2019 | 0.01 | -0.80 | 4.5 | 2.0 |
| Qasem et al. 2019 | 0.25 | -0.10 | 2.0 | 5.4 |
| Qasem et al. 2019 | 0.11 | -1.43 | 2.6 | 9.5 |
| Qasem et al. 2019 | -1.2 | -1.02 | 2.3 | 1.6 |
| Qasem et al. 2019 | -1.2 | -1.02 | 0.9 | 2.6 |
| this study (mean) | -0.3 | -0.1 | 2.2 | 0.6 |