# Peer review of "Resistance and resilience of stream metabolism to high flow disturbances"

_Biogeosciences, 2020_

## Referee Comment (RC1) · Anonymous Referee #1 · 8 Nov 2020

bg-2020-304 Resistance and resilience of stream metabolism to high flow disturbances

**General comments:**

The authors analyse the effects of 15 isolated storm events on stream metabolism, focusing on subsidy-stress hypotheses and drivers of response and recovery. The authors make use of a five- year high-temporal resolution dataset to address their four hypotheses. While the general ideas and approach are interesting and worthy of study, the current manuscript requires major revisions before publication. Some of the suggested revisions are substantial changes to analysis and fall more into the "reject and resubmit" category, but the authors of course can provide adequate justification for not conducting these changes.

**Specific comments:**

*Major:*

1.  After reviewing the approach and the data, and because it underlies all results in the paper, I think the authors need to present a stronger rationale (which currently does not exist)—or change the approach— for how they arrived at their use of arbitrary discharge thresholds (e.g., 50%, 10%) in discriminating the "isolated events". An option could be a simple sensitivity analysis. Additional rationale should support the use of cumulative daily discharge, as opposed to other commonly used metrics in hydrology for event detection. I further think using cumulative discharge may be obscuring some results and indeed missing many events. I first started down this path of inquiry because only 15

isolated events over five years seemed to be a small sample size. Would the same thresholds result in different events if applied to depth or even comparing maximum daily discharge as opposed to cumulative daily discharge? This, I realize, may be a bit of a task because it requires an entirely new analysis, but I think that the authors need to consider this route and defend their assertions more fully. If more events could be used based on a simple adjustment like threshold choice, there could be a much more robust sample size to draw inference from, and would make this a much stronger paper.

2.  Similarly, I think this paper could be much stronger by including as many events as possible, regardless of whether they are "isolated". As discussed in the Introduction, the pulsing paradigm/pulsing equilibrium would seem to include the pulses of hydrologic fluxes, whereas the presented approach discounts them. I understand that perhaps the authors were particularly interested in capturing "resilience" metrics, which may require a period of calm after the storm, so to speak. But, one can imagine a much richer analysis if, for example, the authors calculated some kind of "resistance" metric for as many events as possible, but parsing which ones were preceded by large events. And, for

"resilience", the authors could still calculate the time to return to pre-(initial)event conditions, but just parse which of these "initial" events had subsequent events. Without much effort, the authors could even estimate the subsequent rate of events and its influence on "resilience". One can imagine a figure of, for example, $\Delta$GPP vs. $\Delta$Q where points are colored by their recovery time and sized by their subsequent rate of events. I

mention these suggestions because the current methods seemed disingenuous in taking an arbitrarily "neat" approach to this potentially very fruitful test of the pulsing paradigm.

Another important point in this regard is how to take into account *when* a rain event occurs during the day. For example, if a rain event occurs at 23h, it seems like your approach considers its effect for the previous day, when it is probably more appropriate to consider its effect starting for the following day. (This is understood by the authors in their approach in section 2.5.1, but their simple correlation approach does not effectively get at this idea). The current approach also likely discounts many possible events for this reason. I am aware that this may be a big ask of the authors, and, if different routes are taken, I still suggest that they provide stronger rationale for the (apparently) arbitrary decision for identifying events.

3.  In the same vein, the authors should provide some kind of justification or sensitivity analysis for both of their critical choices in calculation of their resistance/resilience metrics. The first is the choice of "…three days prior to define a range of antecedent metabolism for each isolated flow event." (Lines 140–141). The second is the choice of defining "X prior [as] the maximum or minimum value of GPP or ER from the antecedent range…" (Lines 147–148). Why not the median or mean, which would represent more of the "equilibrium" of the previous period? And why not use the most similar previous day in terms of driving forces–in particular, light availability (this seems especially relevant for the recovery interval!). I'm sure the authors considered such options in their initial work, but they need to do more work to convince the audience of their presented approach—or take a different one if the evidence from sensitivity analyses suggests that they should. Both of these choices are major factors in the subsequent analyses because they define the metrics used, and because these choices do not appear to have literature support/precedent, they need to have clear rationale.

4. Lines 154–156: "To quantify the resilience of GPP and ER, we estimated recovery intervals (RI) by counting the number of days until metabolic rates returned to within the range of pre-event values, signifying a return to antecedent dynamic equilibrium (Figure

3)." This is a good illustration of a potential issue/untapped possibility with the current approach. If you look at the data for the event shown in Figure 3, depth increases in that event by approximately 0.12 m, which decreases light availability by approximately 13%

(according to exponential attenuation). This is nearly exactly the difference between GPP

on 7 February and 9 February, both of which had nearly identical incoming light signals (making them comparable).

5. Lines 157–160: "To ensure additional flow events did not obscure the recovery interval of GPP or ER, we stopped counting RI the day before the next event (i.e., if another flow event happened four days later, we stopped counting RI at 3 days), and have noted this in our results as days+ and used different symbols in data figures." Why? As far as I can tell, the authors throw these data points out in their analysis, and only reference them in

Table 3 (which already uses asterisks to note the issue). Is this to note that the system was on its way to "recovery"? Maybe it would be better to just show a recovery rate, instead of a time, which could result in more data points being included. So, instead of the time it takes to get back to some baseline (which I argue above is a bit arbitrary), you can calculate the rate of increase in GPP over a period (which could be equal to the baseline period that you settle on). Let's say an event occurs and on that day GPP was 5 g $O_2$ m$^{-2}$

d$^{-1}$; the subsequent days maybe it's 8, then 10 g $O_2$ m$^{-2}$ d$^{-1}$. The rate of increase could then be 2.5 g $O_2$ m$^{-2}$ d$^{-2}$ ((10 − 5) / 2). Then, even if a subsequent event occurs, you can still compare the rate of increase before that event. A rate also seems like it could be more comparable/scalable across systems in contrast with a number of days. I don't presume to have the best idea here, but I think an approach like the one outlined above could increase inclusion of useful data points, and thereby lead to more useful inferences.

6. Lines 165–166: "We assessed three categories of potential predictors of metabolic resistance and resilience: antecedent conditions, characteristics of the isolated flow event, and characteristics of the most recent prior flow event." Antecedent conditions and characteristics of the recent prior flow event (especially the latter) are unrelated to any stated hypothesis and appear to come out of nowhere. There needs to be clear rationale in the Introduction that leads the reader to understand why you are doing this.

7. Generally speaking, I had difficulty with the entire Results section, which I think needs a complete rewrite. Some specific details are presented below, but I glossed over several in the interest of time. This section needs to link to stated hypotheses (in the order that they are stated in the Introduction) and test them directly without including spurious tests and weak assertions.

8. Figure 5 as presented is not informative. What do the authors want the reader to understand from this figure? Is the $R^2$ based on a linear regression for all of the points or just the black circles? What is the slope of the regression and the p-value? How does the slope compare to the 1:1 lines? The second panel (right, ER vs GPP recovery interval) is not related to any stated hypothesis. The text discussing this Figure does not support the points on the figure, particularly for the high stated value of ER stimulation $= 0.22$ (Lines 189–190: "…The magnitude of departure for ER ($M_{ER}$) ranged from -0.59 to 0.22, with a median of 0."). Looking at this figure also raises red flags about how the authors defined stimulation/repression. How do the extremely small changes in magnitude shown here compare to the uncertainty in GPP/ER, which are never discussed or propagated through any of these analyses? For example, is a 1% increase (i.e., $M = 0.01$) detectable if uncertainty is considered? The authors should improve this figure substantially, or remove it/leave it as a table. One possible idea is to color or size points based on the event size. Moreover, based on this figure, I am not sure I believe the results on Line 193–195 (italics mine): "Although GPP exhibited stronger responses across isolated flow events than ER, $M_{GPP}$ and $M_{ER}$ were positively correlated ($R^2 = 0.39$, $p = 0.007$, Figure 5) *and not significantly different ($p = 0.06$, $\alpha = 0.05$).*" Just an eyeball test makes this seem unreasonable. ER magnitudes on average are about 0

9. Figure 6 is not easily understood and appears to simply repeat the information on Table 4 in a cluttered way. What key piece of information is the reader supposed to understand from this? The results of the controls on process response in this section 3.3 is quite
difficult to connect with any prior hypotheses and leaves the reader uninformed. There
are two figures and a table with only six sentences to describe them in this section. One
of the stated hypotheses (H2) is never even formally tested here, and only the resistance
metric is tested for H3 (somewhat, in Figure 7), which included both resistance and
resilience metrics.

10. As far as the Discussion and Conclusion, I have many comments, but the issues all stem
from previous issues relating to hypotheses, methods, and results. If the authors apply any
of my suggested revisions to their approach, they will inevitably have to rewrite these
sections. So, I have not provided many specific comments out of the interest of time, but
a few key ones are here. I again suggest to organize the Discussion (and entire document)
in order of the hypotheses as they are presented and as makes logical sense. As written,
the Discussion jumps around in its assertions and ideas. Finally, much of the content in
these sections is hypothetical and rhetorical, with little critical analysis of the results
actually presented in the manuscript and how they relate to the broader literature.

*Minor:*

1. Ideas of pulsing steady state could be clarified a bit with regard to the study design and
terminology throughout. In the Introduction, the authors note "Frequent disturbances
generate oscillations that form a pulsing steady state (sensu Odum et al. 1995) that
includes ambient variability in processes (Resh et al., 1988; Stanley et al., 2010)." (Lines
21–23). So, flow disturbance regime defines the pulsing steady state of lotic systems.
But, the authors then use–incorrectly I think–the periods outside of flow disturbance to
define a "pulsing steady state" (or at times, "pulsing equilibrium", like in Figure 1, and
"dynamic equilibrium", like in Figure 3, and "antecedent equilibrium" on Line 187), to
which they then compare to periods *with* flow disturbance. The approach is clear, but
there is some circular reasoning with respect to the definition of pulsing steady state. I
recommend perhaps using different terminology for these two concepts. One idea could
be to use something like "ambient equilibrium" for metabolism under baseflow
conditions, and "pulsing equilibrium" to refer to the larger scale, (inter)annual behaviour
of as originally conceived by Odum. I think these small changes would improve the
clarity of the study design and arguments within.

2. Similarly, I do not think that "resilience" is appropriately used throughout the manuscript,
first defined by the authors on Lines 59–60: "We can also quantify post-disturbance
ecosystem responses by estimating resilience: the time it takes for a process returns to
equilibrium following a disturbance (Carpenter et al., 1992)." We have first of all the
issue of "return[ing] to equilibrium", which is not so clear based on the previous
definition of a pulsed equilibrium that includes disturbance. In a system organized by
regular disturbance regimes, the idea of resilience to that same disturbance regime is a bit
convoluted. In contrast, the idea of a "recovery interval" to previous ambient conditions
is clear and appropriate. Resilience in this context might make more sense if there were
alternative metabolic equilibria that the stream could occupy, where each of these
equilibria were tolerant to different levels disturbance. Ultimately, this is a choice of
language and does not affect the analyses presented and if the authors opt to keep their
current choice, I suggest spending some more time to expand these ideas/defend their use
out in the Introduction and Discussion.
3. Line 73: "(H0) some flow events will not push GPP and ER outside of their pulsing
equilibrium." Should this by "(H4)"? Or is this some kind of null hypothesis? Consider
renumbering, or placing this at the beginning of the sentence–seems strange to go from
1–3 then back to 0.
4. Lines 70–71: "…(H2) there will be a stimulation of GPP and ER at intermediate flow
disturbances due to an influx of limiting carbon and nutrients…". Is this stream known to
be limited by carbon and nutrients? What is the timeframe for stimulation? It seems like
the influx of carbon and nutrients would pass through the system quite quickly in this
small stream, and would not be easily acquired/processed by organisms. In larger systems
with long recession curves, I think this perspective can make sense, but this hypothesis
does not seem well supported in the Introduction as currently written.
5. Lines 71–72: "…(H3) metabolic resistance and resilience will change with the size of the
event, with larger flow disturbances inducing more stress due to enhanced scour…" The
point about scour here seems important. Scour is a function of shear stress, which itself is
a linear function of depth. The authors focus on discharge as their subsidy/stress driver,
but I wonder if water depth would be more appropriate? Because depth only increases to
the square-root of discharge (for a large range of depth-discharge in their Supplemental data), a quadrupling of discharge only results in a doubling of benthic shear stress. I don't expect for the authors to redo any analyses with this perspective, but I do think this kind of information would be useful to include especially in the Discussion so that future works would consider this as well. It also could be used as a future framework to further test the idea of subsidy/stress balance. Depth is a first-order control on both light availability and shear stress at the benthos, making it a more appropriate indicator of stress than discharge.

6. The light data (first referred to on lines 92–93) appear to be in units of µA according to the supplementary material ("ODonnellHotchkiss_SuppData_ReadMe.pdf", under point

"1"). I am not familiar with this unit (is it micro-amperes?) for sunlight, and I think this needs some clarification. The light data in the data file itself appear to range between 0

and 1, but the *streammetabolizer* model take data in PAR (units $\mu\text{mol m}^{-2}\text{ s}^{-1}$), which can be upwards of 1000 by noontime. I'm sure this is not a major issue, but I do not think the results will be replicable as currently presented—those units, if fed into

*streammetabolizer*, will lead to very strange outputs I think. The sensor used (according to O'Donnell and Hotchkiss 2019) is a Campbell CS300, which should output data in typical units like $\text{W m}^{-2}$.

7. Line 140: "To acknowledge the pulsing, day-to-day variability…" I don't think "pulsing"

is appropriate or needed here.

8. Line 152: "…suppression…" please check for the consistent use of suppression and repression (and others) throughout.

9. In section 3.2 "Metabolic resistance and resilience", it would be very helpful to explicitly organize/label these paragraphs according to your numbered hypotheses from the

Introduction. For example, Lines 187–192: There is no directly stated connection between any of the statements presented here and the actual hypotheses.

10. Lines 194–196 bring up another issue with the idea of "magnitude" (italics mine): "*M*

*GPP was less than M ER for nearly all flow events*, except for one in which M GPP and

M ER were both zero and two where M GPP and M ER were both small (Figure 5,

Figure 195 A19)." The general idea of magnitude is that is not directional. I would argue that the magnitude of GPP response was *greater* than that of ER, and that they both had similar directional change (decrease in process magnitude). Consider different language throughout.

11. Lines 198–199: "Similarly, the only other event that stimulated GPP (M GPP = 0.03) had no ER response, suggesting many flow disturbances may decouple GPP and ER." This seems like an unsupported assertion (which should be in the Discussion, if anywhere) based on one event with an extremely small signal.

12. Table 3: n/a is not clearly defined.

13. Lines 208–209: "Although GPP and ER are linked processes, the variables that were moderate or strong predictors of resistance or resilience $(r > 0.5)$." Why is 0.5 the threshold for being a strong predictor? That's only 25% of the variance explained.

14. Lines 210–211: "Because the median RI ER was zero, bivariate correlations could not be used to determine potential predictors of ER resilience." Another reason to consider rate instead of day count.

15. Lines 214–215: "Overall, there were multiple environmental controls on metabolic resistance or resilience that were strongly correlated with either GPP or ER, but no significant drivers of both GPP and ER resistance and resilience." This is not supported by the figure or the table.

16. Line 219: "Notably, ER was more resistant than GPP (Figure 1)." Figure 1 is a conceptual figure and does not support this statement.

17. Line 239–240: "In assessing metabolic responses and recovery from smaller flow events relative to the dynamic equilibrium of metabolism at baseflow, we found some of the shortest metabolic recovery intervals recorded in the literature (Figure 8; Table A1)." Do these other studies use the exact same methodology as you? How are they comparable? Are they similar sized streams? You should compare and contrast more here.

18. Line 259–260: "Contrary to our predictions, the size of the most recent antecedent flow disturbance had a positive relationship with M GPP and M ER (Figure A19)." Where is this prediction?

**19. Technical corrections:**

1. Equation 1 (Line 110) seems boiler-plate and unnecessary.

2. There are extra parentheses in Figure 2c description for "$((m^3 \ d^{-1}))$"

3. Figure 3 should describe what the error bars are on the GPP estimates.

4.  Lines 163–164: "Quantifying how different antecedent conditions induce variable
responses from GPP and ER is critical to furthering our understanding of stream
ecosystem responses to flow disturbances." This belongs in the Introduction, not the
Methods.

5.  Lines 167–168: "Antecedent medians for turbidity were estimated from seven days prior
due to missing sensor data." This is not clear, please explain what this means. There was
always missing data for turbidity within the three days prior to an event? I can't imagine
turbidity changes very much at baseflow.

6.  Lines 190–191: "Three of 15 flow events stimulated ER, 5 repressed ER, and ER did not
deviate from the antecedent equilibrium for 7 events (i.e., M ER was 0)." It's more
common to use numerals for numbers greater than 10, and to spell the numbers out for
numbers less than 10.

---

## Referee Comment (RC2) · Anonymous Referee #2 · 9 Nov 2020

General comments: The manuscript bg-2020-304 "Resistance and resilience of stream metabolism to high flow disturbances" by O'Donnel & Hotchkiss analyzes in a third-order stream the response of Gross Primary Production (GPP) and Ecosystem Respiration (ER) altered by disturbances such as isolated high flow events. The study is relevant as it is based on a long-term monitoring (5 years) of GPP and ER, which is critical to decipher seasonal and multiyear variability of stream ecology in the context of climate change. Overall, I found the approach of the study interesting but the authors should explore their dataset further, therefore I suggest major revisions.

Major comments: I was surprised that the authors did not discuss about in-stream net

ecosystem production (NEP). NEP is critical to decipher stream ecology as it does indicate whether an ecosystem is fixing more C than is respiring. The authors showed that ER has higher resistance and resilience in comparison to GPP, thereby should shifted NEP towards heterotrophy (decrease of the GPP:ER ratio). I believe it would be very interesting for the reader to understand/know how NEP is affected by high flow events. I suggest adding figures and discussion about NEP.

The dataset used by the authors is extended in time but the paper lacks of seasonal variability analysis. How GPP, ER and NEP, resistance and resilience are affected by seasons and by year-to-year variability. Indeed, temperature effect on stream metabolism is usually significant. The authors needs justify that the variability induced by the temperature does not overcome the variability induced by flow events. In the revised paper, I suggest the authors adding a figure such as GPP vs ER with points colored according to seasons or river flow.

In low order streams, GPP and ER are affected by groundwater inputs, as groundwater inputs are usually significant in such streams. Groundwater exhibit usually low oxygen concentration, which may be problematic when GPP and ER are based on oxygen monitoring. Inputs of low-O2 groundwater in stream can overestimate ER and underestimate GPP. However, the equation 1 does not take into account groundwater inputs. Why? Oxygen measurements during high flow, especially in low-order streams, can give erroneous values, so are the authors sure to measure appropriate values during the high flow events.

There is some variability in day-to-day metabolism rates; therefore, I do not understand why the authors took the maximum or minimum value of GPP (or ER) from the antecedent range to estimate the resistance. I believe that the median or the mean would be more appropriate. In addition, why the authors used 3 days as the antecedent range. Is it arbitrary?

I have concerns on how the isolated flow events are selected. Indeed, in the figure 2A,

I observe that only few high flow events (15 events in 5 years) are actually selected by the authors. In the figure 2C, the authors did not provide statistical analysis on the difference of cumulative daily discharge between all days and isolated flow events. Is it statistically different? Visually, it seems not, considering the high range of cumulative daily discharge during "all days". If it is not statistically different, it means that the disturbance is the same in both groups. Is there a way for the authors to arbitrary select a greater number of high flow events? As examples, the authors could use maximum daily discharge vs cumulative daily discharge or the change in discharge from pre- to peak-storm flow. By the way, I do not understand why the authors wants to select isolated flow events rather than all high flow events. I believe that estimating resistance and resilience in each high flow events would be much more robust. In addition, the paper aims to study ecosystem response to high flow events, but the paper do not contain figures showing the relation of river flow versus stream metabolism. What is the relationship between river flow and GPP, ER, NEP, resistance and resilience? Resistance represents the change in GPP (or ER) during a change in river flow, so maybe it would be interesting to show $\Delta$GPP (or ER) with $\Delta$Q?

I the discussion section, I do not feel that the authors fully responds to their four hypotheses. How can the authors responds to H2 where they actually do not show carbon or nutrients measurements? H0 is strongly dependent on how you arbitrary selected the flow events. To my point of view, with their study design (unless the authors have measurements of carbon and nutrients) the authors can discuss only about H1 and H3. In addition, I also suggest rewriting the Discussion section in a more logical sense following the order of their hypotheses.

Minor Comments: L.1: Please, add somewhere in the abstract the ranges of ER, GPP, NEP, resistance and resilience. L.10-11: You defined the metabolic resistance as the magnitude of departure from the dynamic equilibrium during antecedent lower flows, so why using the words "ER magnitude of departure" to refer to resistance. Better used the word resistance and resilience throughout the text once you have defined those words.

Please add also in the abstract that more ER or GPP is resistant less the magnitude of departure is large. L. 69: It is strange to start with H1 and finish with H0 L.81: Usually precipitation is in mm L.90: How did you calibrate the different sensors, and how often did you check the calibration? L.93: can you add the weather station on the figure A1. The figure A1 needs a scale, a geographic footprint. L105: Please, specify that you works with gas exchange coefficient not gas exchange velocity L.110: How did you measure the PAR? How did you calculate the average depth? L.119: What are the values of K? L.133: Please define Qi L.167:169: For the different variables other than GPP and ER you used the medians from three days prior the flow event for correlations, but for resistance you used the maximum or minimum GPP or ER before the flow event. I believe it would be robust to use the same methods. L.180: Is the cumulative daily discharge statistically different between isolated events and other days? L.182-185: As mentioned in the major comments please showed how GPP, ER and NEP are affected by seasons and river flow. L219 Where can I see that ER was more resistant than GPP. It is on a daily basis? Yearly basis? Multi-year basis? Please give some details, some stats should be applied. Figure 1 do not show your results. L.228: Same comments 228-230: Can you show some results confirming what you stipulate? In the table 4 turbidity seems weakly correlated with resilience of ER and GPP. 254: The authors have a dataset representing 5 years of monitoring so why they cannot answer to this question, at least partly? Figure 5: I am not convinced by this figure. Figure 6: Same data as in the table 4, perhaps not relevant. Figure 8: Is there a better way to present these results? Table 4: Please indicate the p-values, Indeed, two parameters can have a correlation coefficient greater than 0.5 but they are still not correlated together if the p-value is greater than 0.05. Figure A4 to A18: In each isolated flow events: GPP, ER and Discharge can be combined in one figure with 3 axis Figure A20: To my point of view a figure such as this one showing the seasonal variability of the different parameters (GPP, ER, NEP, résistance and resilience) is important and must appear in the main text.

---

## Author Comment (AC1) · 17 Dec 2020

With the exception of the "GENERAL RESPONSE", all of our response text sections begin with "RESPONSE" immediately following reviewer comments.

GENERAL RESPONSE:

This text is included in both responses to reviewers, with specific responses to reviewers below.

Here we respond more generally to questions about why/how we selected isolated flow events and the resulting number of events suitable for our analyses (n = 15 events over

5 years). We emphasize that we focused on quality over quantity when selecting for and analyzing stream metabolism before, during, and after high flow events. Our methods were chosen to address a lingering knowledge gap in our understanding of ecosystem processes: how biological processes (gross primary production and ecosystem respiration, GPP and ER) respond to and recover from discrete higher flow disturbances during storms, how those two processes compare to one another, and which environmental drivers may best explain these dynamics. Potential metabolic responses include subsidy (increasing rates due to higher substrate concentrations), stress (decreasing rates due to physical or chemical disturbances), or no change, which allows for our work to build on concepts fundamental to biogeochemistry and ecology. An additional knowledge gap is how different processes (i.e., GPP, ER) may respond differently to high flows. How we chose to quantify changes in metabolism during higher flow acknowledges the "pulsing steady state" of ecosystems in a novel way. In our revised manuscript, we will better introduce and identify how and where these different concepts apply to, inform, and are answered by our research.

The goal of this work was to assess how metabolism responded to and recovered from higher flow events that were also isolated flow events. Indeed, this decreased the number of suitable events for analysis. But our choice of methods allowed us to focus on response/recovery to discrete disturbances and avoid biased comparisons of pre/during/post multiple high flow (but not isolated) events that encompass time periods that are long enough (e.g., weeks) where pre/post comparisons are less meaningful. Perhaps we could have selected a more pristine stream with less flashy hydrology at the start of this project, but another motivation of our work is to better understand processes in less pristine ecosystems (historically understudied because they are more challenging sites to obtain high-quality metabolism estimates from, another factor that decreased the number of events with appropriate data for our analysis). Despite having "only 15 events", most past analyses included a similar or fewer number of events (e.g, n=10 in Reisinger et al. 2017) over a shorter time period. Our work fills in substantial knowledge gaps: we analyzed across seasons (not only summer months or a short

sensor deployment period) and high flow magnitudes (not only base flow or the highest flow disturbances).

After all appropriate QA/QC measures, we had 1375 days of metabolism estimates over 5 years (which were reported in full in O'Donnell & Hotchkiss 2019 Water Resources Research). To calculate resistance and recovery, we needed consecutive days of high-quality metabolism estimates, which further limited the number of high flow events appropriate for our analyses. For example, in 2016 there were 52 (out of 352) days with quality-checked sensor data that had a 50% flow change relative to the day prior. After looking at these 52 storms and selecting those that had 3 days before and 3 days after without any other flow events, we had 12 that were isolated. After quality-checking our metabolism estimates for all of those days, we had 4 high flow events from 2016 that passed all quality-checking steps required for this analysis.

/ end of GENERAL RESPONSE

REVIEWER 1:

General comments:

The authors analyse the effects of 15 isolated storm events on stream metabolism, focusing on subsidy-stress hypotheses and drivers of response and recovery. The authors make use of a five-year high-temporal resolution dataset to address their four hypotheses. While the general ideas and approach are interesting and worthy of study, the current manuscript requires major revisions before publication. Some of the suggested revisions are substantial changes to analysis and fall more into the "reject and resubmit" category, but the authors of course can provide adequate justification for not conducting these changes.

RESPONSE: We appreciate this reviewer's feedback acknowledging the novelty of our work, encouraging us to clarify our research objectives, and highlighting areas where we can better justify the methods used to address current knowledge gaps related

to how ecosystem metabolism responds to and recovers from high flow events. It appears that placing our work in the context of subsidy-stress, pulsing baselines, and ecosystem resistance/resilience "muddied" our communication of research objectives, which we will clarify and focus in a revised manuscript. We highlight how we will step through the relevance of multiple ecological concepts to this research in our general response above. We respond to specific suggestions in more detail below.

Specific comments:

Major:

1. After reviewing the approach and the data, and because it underlies all results in the paper, I think the authors need to present a stronger rationale (which currently does not exist)—or change the approach— for how they arrived at their use of arbitrary discharge thresholds (e.g., 50%, 10%) in discriminating the "isolated events". An option could be a simple sensitivity analysis. Additional rationale should support the use of cumulative daily discharge, as opposed to other commonly used metrics in hydrology for event detection. I further think using cumulative discharge may be obscuring some results and indeed missing many events. I first started down this path of inquiry because only 15 isolated events over five years seemed to be a small sample size. Would the same thresholds result in different events if applied to depth or even comparing maximum daily discharge as opposed to cumulative daily discharge? This, I realize, may be a bit of a task because it requires an entirely new analysis, but I think that the authors need to consider this route and defend their assertions more fully. If more events could be used based on a simple adjustment like threshold choice, there could be a much more robust sample size to draw inference from, and would make this a much stronger paper.

RESPONSE: We responded to concerns about "only 15 isolated events" in our general response at the start of this document, but add a bit more detail here. Again, what we sacrificed in quantity we gained in quality. A full dataset for a single day required

quality-checked sensor data as well as metabolism estimates that passed all QA/QC steps, which means some isolated events were excluded from our analysis. We excluded values of physicochemical parameters that were below the 1% or above the 99% quantile and removed physicochemical measurements we knew were inaccurate due to sensors being out of the water during low flow or not working properly (e.g., turbidity at zero). This is covered in this text in the results and in greater detail in O'Donnell and Hotchkiss 2019. In our revised manuscript we will clarify how and why we selected isolated storms. The 50% change in flow for our high flow events ensured those events were indeed outside of a pulsing baseline flow. We defined a flow event as >10% change in Q when comparing the high flow changes to prior metabolic rates, as smaller changes in Q may still influence metabolism. In prior analyses leading up to those presented in our manuscript, we did test different thresholds of flow change or different discharge metrics, and settled on our current method because after exploring the trade-off between different delta Q thresholds, number of quality-checked events, and differences between ambient stream flow and higher flow events.

2. Similarly, I think this paper could be much stronger by including as many events as possible, regardless of whether they are "isolated". As discussed in the Introduction, the pulsing fluxes, whereas the presented approach discounts them. I understand that perhaps the authors were particularly interested in capturing "resilience" metrics, which may require a period of calm after the storm, so to speak. But, one can imagine a much richer analysis if, for example, the authors calculated some kind of "resistance" metric for as many events as possible, but parsing which ones were preceded by large events. And, for "resilience", the authors could still calculate the time to return to pre-(initial)event conditions, but just parse which of these "initial" events had subsequent events. Without much effort, the authors could even estimate the subsequent rate of events and its influence on "resilience". One can imagine a figure of, for example, $\Delta$GPP vs. $\Delta$Q where points are colored by their recovery time and sized by their subsequent rate of events. I mention these suggestions because the current methods seemed disingenuous in taking an arbitrarily "neat" approach to this potentially very

fruitful test of the pulsing paradigm. Another important point in this regard is how to take into account when a rain event occurs during the day. For example, if a rain event occurs at 23h, it seems like your approach considers its effect for the previous day, when it is probably more appropriate to consider its effect starting for the following day. (This is understood by the authors in their approach in section 2.5.1, but their simple correlation approach does not effectively get at this idea). The current approach also likely discounts many possible events for this reason. I am aware that this may be a big ask of the authors, and, if different routes are taken, I still suggest that they provide stronger rationale for the (apparently) arbitrary decision for identifying events.

RESPONSE: We like the idea of the reviewer's suggested plot of "$\Delta$GPP vs. $\Delta$Q where points are colored by their recovery time and sized by their subsequent rate of events", and will see if such a plot might help better communicate our results. We disagree that our decisions were "arbitrary" and we do not think it is appropriate to change our research objectives for this work because we could analyze the data for a different purpose or with a different set of QA/QC standards to gain more events. As the reviewer acknowledged, our focus was to understand metabolic resistance to and recovery from isolated high flow events. Furthermore, we were interested in assessing these dynamics in a hydrologically flashy stream draining a heavily modified landscape. We agree these other topics are very interesting and look forward to seeing more research addressing this type of proposed work, but they are not the aim of our paper. We will ensure our objectives are focused, well-described, and clearly justified in our revised manuscript. Again, a higher-level response to this and #1 above is in our overall summary at the top of this document.

3. In the same vein, the authors should provide some kind of justification or sensitivity analysis for both of their critical choices in calculation of their resistance/resilience metrics. The first is the choice of "...three days prior to define a range of antecedent metabolism for each isolated flow event." (Lines 140–141).

RESPONSE: This is a good question and one that we will further describe in our revised manuscript. Our choice of three days was the result of balancing best practices from published papers on similar topics (e.g,. 4 days prior stable baseflow, Reisinger et al. 2017) while still analyzing as many events with appropriately QA/QC'ed data as possible.

The second is the choice of defining "X prior [as] the maximum or minimum value of GPP or ER from the antecedent range..." (Lines 147–148). Why not the median or mean, which would represent more of the "equilibrium" of the previous period?

RESPONSE: We wanted to capture the pulse of days prior (Figure 1). The pulse isn't captured in a mean or median, and analyzing the metabolism data in this way can result in estimating a departure from a mean or median that is erroneously considered to be different from baseflow tendencies when in reality it is within the ambient pulse of the system. We will revisit where we highlighted this choice in our methods (we adopted this more conservative method for measuring magnitude of departure and recovery that is more appropriate for variable ecosystems) to ensure this decision is well-described in our revised manuscript.

And why not use the most similar previous day in terms of driving forces–in particular, light availability (this seems especially relevant for the recovery interval!). I'm sure the authors considered such options in their initial work, but they need to do more work to convince the audience of their presented approach—or take a different one if the evidence from sensitivity analyses suggests that they should. Both of these choices are major factors in the subsequent analyses because they define the metrics used, and because these choices do not appear to have literature support/precedent, they need to have clear rationale.

RESPONSE: While an interesting idea, this approach requires too many assumptions about our ability to predict GPP and ER based on light or temperature or other environmental data alone. While our knowledge of metabolism in streams and rivers is growing rapidly, we are not at a point where we can justify how we would select a "most

similar previous day" for comparisons between the range of baseline metabolic rates and responses/recovery from high flow events with the environmental data we have for this site.

4. Lines 154–156: "To quantify the resilience of GPP and ER, we estimated recovery intervals (RI) by counting the number of days until metabolic rates returned to within the range of pre-event values, signifying a return to antecedent dynamic equilibrium (Figure 3)." This is a good illustration of a potential issue/untapped possibility with the current approach. If you look at the data for the event shown in Figure 3, depth increases in that event by approximately 0.12 m, which decreases light availability by approximately 13% (according to exponential attenuation). This is nearly exactly the difference between GPP on 7 February and 9 February, both of which had nearly identical incoming light signals (making them comparable).

RESPONSE: This is an interesting observation that we will explore and include in a revised manuscript if deemed an appropriate way to discuss results within the context of our research objectives.

5. Lines 157–160: "To ensure additional flow events did not obscure the recovery interval of GPP or ER, we stopped counting RI the day before the next event (i.e., if another flow event happened four days later, we stopped counting RI at 3 days), and have noted this in our results as days+ and used different symbols in data figures." Why? As far as I can tell, the authors throw these data points out in their analysis, and only reference them in Table 3 (which already uses asterisks to note the issue). Is this to note that the system was on its way to "recovery"? Maybe it would be better to just show a recovery rate, instead of a time, which could result in more data points being included. So, instead of the time it takes to get back to some baseline (which I argue above is a bit arbitrary), you can calculate the rate of increase in GPP over a period (which could be equal to the baseline period that you settle on). Let's say an event occurs and on that day GPP was 5 g O2 m-2 d-1 ; the subsequent days maybe it's 8, then 10 g O2 m d . The rate of increase could then be 2.5 g O2 m-2 d-1 ( (10-5) / 2).

Then, even if a subsequent event occurs, you can still compare the rate of increase before that event. A rate also seems like it could be more comparable/scalable across systems in contrast with a number of days. I don't presume to have the best idea here, but I think an approach like the one outlined above could increase inclusion of useful data points, and thereby lead to more useful inferences.

RESPONSE: This is a great point, and an idea we explored early on but did not include in our final analysis. However, seeing that there is interest in recovery rates in addition to recovery days, we will update our revised manuscript to include this metric. We will still include the number of days to recovery, when measurable (yes, the days+ is to note it was on track to recovery, but did not return to the range of baseline metabolic rates before the next high flow event). We will keep RI estimates to allow us to compare our results to other studies (e.g., Figure 8) for broader discussion.

6. Lines 165–166: "We assessed three categories of potential predictors of metabolic resistance and resilience: antecedent conditions, characteristics of the isolated flow event, and characteristics of the most recent prior flow event." Antecedent conditions and characteristics of the recent prior flow event (especially the latter) are unrelated to any stated hypothesis and appear to come out of nowhere. There needs to be clear rationale in the Introduction that leads the reader to understand why you are doing this.

RESPONSE: We will update the objectives and hypotheses outlined in the introduction to clarify our interest in estimating metabolic responses to flow change and potential drivers of those responses.

7. Generally speaking, I had difficulty with the entire Results section, which I think needs a complete rewrite. Some specific details are presented below, but I glossed over several in the interest of time. This section needs to link to stated hypotheses (in the order that they are stated in the Introduction) and test them directly without including spurious tests and weak assertions.

RESPONSE: Agreed. Thank you for the detailed comments below and the reminder

to ensure the introduction and discussion are better aligned. We will revisit and revise with these comments in mind.

8. Figure 5 as presented is not informative. What do the authors want the reader to understand from this figure?

RESPONSE: We respectfully disagree that showing these results graphically is not informative, which is why it's cited numerous times in the manuscript. We want the readers to understand the relationship between recovery and magnitude of departure for GPP and ER, which this figure visualizes.

Is the R based on a linear regression for all of the points or just the black circles? What is the slope of the regression and the p-value? How does the slope compare to the 1:1 lines?

RESPONSE: R is based on all the points. We will clarify this in our analysis section and the figure legend. P-values are included in the text (lines 194 and 206). We will add them to the figure legend as well.

The second panel (right, ER vs GPP recovery interval) is not related to any stated hypothesis.

RESPONSE: While we do not agree that every plot or table is only justified if related to a specific hypothesis, we will clarify that in trying to understand how metabolism changes with higher flow (and why), we are interested in the differences between GPP and ER as well. This is included in Figure 1, but can certainly be highlighted better in the text as well in a way that justifies the inclusion of the second panel.

The text discussing this Figure does not support the points on the figure, particularly for the high stated value of ER stimulation = 0.22 (Lines 189–190: "...The magnitude of departure for ER ($M_{ER}$) ranged from -0.59 to 0.22, with a median of 0.").

RESPONSE: We will check this mismatch between the plot and our text - it appears part of the y axis may have been cut. The revised manuscript will correct accordingly.

Looking at this figure also raises red flags about how the authors defined stimulation/repression. How do the extremely small changes in magnitude shown here compare to the uncertainty in GPP/ER, which are never discussed or propagated through any of these analyses? For example, is a 1% increase (i.e., M = 0.01) detectable if uncertainty is considered?

RESPONSE: We note that our metrics were indeed detectible relative to metabolism estimates (with low uncertainty, as shown in Fig 3, A4-A18, and supplementary data files). Consequently, we disagree that this approach "raises red flags".

The authors should improve this figure substantially, or remove it/leave it as a table. One possible idea is to color or size points based on the event size. Moreover, based on this figure, I am not sure I believe the results on Line 193–195 (italics mine): "Although GPP exhibited stronger responses across isolated flow events than ER, M GPP and M ER were positively correlated (R 2 = 0.39, p = 0.007, Figure 5) and not significantly different (p = 0.06, $\alpha$ = 0.05)." Just an eyeball test makes this seem unreasonable. ER magnitudes on average are about 0.

RESPONSE: We will consider these additional comments and questions about this figure as we decide whether to convert to a table or update it for the revised manuscript.

9. Figure 6 is not easily understood and appears to simply repeat the information on Table 4 in a cluttered way. What key piece of information is the reader supposed to understand from this? The results of the controls on process response in this section 3.3 is quite difficult to connect with any prior hypotheses and leaves the reader uninformed. There are two figures and a table with only six sentences to describe them in this section. One of the stated hypotheses (H2) is never even formally tested here, and only the resistance metric is tested for H3 (somewhat, in Figure 7), which included both resistance and resilience metrics.

RESPONSE: We will move either Table 4 or Figure 6 to the appendix in our revised manuscript.

10. As far as the Discussion and Conclusion, I have many comments, but the issues all stem from previous issues relating to hypotheses, methods, and results. If the authors apply any of my suggested revisions to their approach, they will inevitably have to rewrite these sections. So, I have not provided many specific comments out of the interest of time, but a few key ones are here. I again suggest to organize the Discussion (and entire document) in order of the hypotheses as they are presented and as makes logical sense. As written, the Discussion jumps around in its assertions and ideas. Finally, much of the content in these sections is hypothetical and rhetorical, with little critical analysis of the results actually presented in the manuscript and how they relate to the broader literature.

RESPONSE: As stated above in response to similar comments, we will keep this in mind when we revise the manuscript.

Minor:

1. Ideas of pulsing steady state could be clarified a bit with regard to the study design and terminology throughout. In the Introduction, the authors note "Frequent disturbances generate oscillations that form a pulsing steady state (sensu Odum et al. 1995) that includes ambient variability in processes (Resh et al., 1988; Stanley et al., 2010)." (Lines 21–23). So, flow disturbance regime defines the pulsing steady state of lotic systems. But, the authors then use–incorrectly I think–the periods outside of flow disturbance to define a "pulsing steady state" (or at times, "pulsing equilibrium", like in Figure 1, and "dynamic equilibrium", like in Figure 3, and "antecedent equilibrium" on Line 187), to which they then compare to periods with flow disturbance. The approach is clear, but there is some circular reasoning with respect to the definition of pulsing steady state. I recommend perhaps using different terminology for these two concepts. One idea could be to use something like "ambient equilibrium" for metabolism under baseflow conditions, and "pulsing equilibrium" to refer to the larger scale, (inter)annual behaviour of as originally conceived by Odum. I think these small changes would improve the clarity of the study design and arguments within.

RESPONSE: We agree these small changes would improve clarify and will revise accordingly when we update our manuscript.

2. Similarly, I do not think that "resilience" is appropriately used throughout the manuscript, first defined by the authors on Lines 59–60: "We can also quantify post-disturbance ecosystem responses by estimating resilience: the time it takes for a process returns to equilibrium following a disturbance (Carpenter et al., 1992)." We have first of all the issue of "return[ing] to equilibrium", which is not so clear based on the previous definition of a pulsed equilibrium that includes disturbance. In a system organized by regular disturbance regimes, the idea of resilience to that same disturbance regime is a bit convoluted. In contrast, the idea of a "recovery interval" to previous ambient conditions is clear and appropriate. Resilience in this context might make more sense if there were alternative metabolic equilibria that the stream could occupy, where each of these equilibria were tolerant to different levels disturbance. Ultimately, this is a choice of language and does not affect the analyses presented and if the authors opt to keep their current choice, I suggest spending some more time to expand these ideas/defend their use out in the Introduction and Discussion.

RESPONSE: We agree, and will keep these comments about word choice in mind as we revise our manuscript.

3. Line 73: "(H0) some flow events will not push GPP and ER outside of their pulsing equilibrium." Should this by "(H4)"? Or is this some kind of null hypothesis? Consider renumbering, or placing this at the beginning of the sentence–seems strange to go from 1–3 then back to 0.

RESPONSE: We will re-order or change to H4 in our revised manuscript.

4. Lines 70–71: "...(H2) there will be a stimulation of GPP and ER at intermediate flow disturbances due to an influx of limiting carbon and nutrients...". Is this stream known to be limited by carbon and nutrients? What is the timeframe for stimulation? It seems like the influx of carbon and nutrients would pass through the system quite quickly in

this small stream, and would not be easily acquired/processed by organisms. In larger systems with long recession curves, I think this perspective can make sense, but this hypothesis does not seem well supported in the Introduction as currently written.

RESPONSE: We will update the introduction and site information to provide better context for this work. While the speed at which nutrients and carbon travels during storms will increase, the ability of microbes to respond to increases in carbon and nutrients is not limited to microbes in larger rivers (e.g., Demars 2019). And yes, the stream is carbon limited. We have unpublished data on carbon and nutrient limitation at our study site that we can mention as part of the site information in our revised manuscript.

5. Lines 71–72: "...(H3) metabolic resistance and resilience will change with the size of the event, with larger flow disturbances inducing more stress due to enhanced scour..." The point about scour here seems important. Scour is a function of shear stress, which itself is a linear function of depth. The authors focus on discharge as their subsidy/stress driver, but I wonder if water depth would be more appropriate? Because depth only increases to the square-root of discharge (for a large range of depth-discharge in their Supplemental data), a quadrupling of discharge only results in a doubling of benthic shear stress. I don't expect for the authors to redo any analyses with this perspective, but I do think this kind of information would be useful to include especially in the Discussion so that future works would consider this as well. It also could be used as a future framework to further test the idea of subsidy/stress balance. Depth is a first-order control on both light availability and shear stress at the benthos, making it a more appropriate indicator of stress than discharge.

RESPONSE: Thanks for this comment. We agree that is another useful way to consider these processes, and will include it in our revised discussion.

6. The light data (first referred to on lines 92–93) appear to be in units of $\mu$A according to the supplementary material ("ODonnellHotchkiss_SuppData_ReadMe.pdf", under

point "1"). I am not familiar with this unit (is it micro-amperes?) for sunlight, and I think this needs some clarification. The light data in the data file itself appear to range between 0 and 1, but the streammetabolizer model take data in PAR (units $\mu$mol m s ), which can be upwards of 1000 by noontime. I'm sure this is not a major issue, but I do not think the results will be replicable as currently presented—those units, if fed into streammetabolizer, will lead to very strange outputs I think. The sensor used (according to O'Donnell and Hotchkiss 2019) is a Campbell CS300, which should output data in typical units like W m .

RESPONSE: The units for light do not matter because how we use light in the metabolism model (Eq 1) is a ratio of light at time i divided over the sum of light over the entire day: PARi / sumPAR. We just realized we need to correct this in our revised manuscript, as it's noted as PARt, not PARi, in equation 1. That said, we will check units and make sure they are the same throughout data files and the revised manuscript.

7. Line 140: "To acknowledge the pulsing, day-to-day variability..." I don't think "pulsing" is appropriate or needed here.

RESPONSE: We will remove pulsing and state "To acknowledge the day-to-day variability...."

8. Line 152: "...suppression..." please check for the consistent use of suppression and repression (and others) throughout.

RESPONSE: We will check for consistent use of suppression and repression throughout.

9. In section 3.2 "Metabolic resistance and resilience", it would be very helpful to explicitly organize/label these paragraphs according to your numbered hypotheses from the Introduction. For example, Lines 187–192: There is no directly stated connection between any of the statements presented here and the actual hypotheses.

RESPONSE: We will update the hypothesis section of the introduction to include discussion that is not just focused on S-S hypotheses.

10. Lines 194–196 bring up another issue with the idea of "magnitude" (italics mine): "M GPP was less than M ER for nearly all flow events, except for one in which M GPP and M ER were both zero and two where M GPP and M ER were both small (Figure 5, Figure 195 A19)." The general idea of magnitude is that is not directional. I would argue that the magnitude of GPP response was greater than that of ER, and that they both had similar directional change (decrease in process magnitude). Consider different language throughout.

RESPONSE: Thanks for this perspective. We will use appropriate terms related to magnitude and direction in our revised manuscript.

11. Lines 198–199: "Similarly, the only other event that stimulated GPP (M GPP = 0.03) had no ER response, suggesting many flow disturbances may decouple GPP and ER." This seems like an unsupported assertion (which should be in the Discussion, if anywhere) based on one event with an extremely small signal.

RESPONSE: We will refine the language in the results to better reflect the magnitude of responses/signals. We will move the "suggesting many flow disturbances may decouple GPP and ER" to the discussion with additional support for this statement from other results.

12. Table 3: n/a is not clearly defined.

RESPONSE: Thank you for catching this. We will define n/a in the updated manuscript. It is when ER or GPP did not deviate from the antecedent range, and therefore had no recovery interval.

13. Lines 208–209: "Although GPP and ER are linked processes, the variables that were moderate or strong predictors of resistance or resilience (r > 0.5)." Why is 0.5 the threshold for being a strong predictor? That's only 25% of the variance explained.

RESPONSE: We will revisit all descriptive wording used and make sure it matches the

thresholds we set in our statistical analyses. We will cite references supporting our decision for these thresholds in the revised manuscript.

14. Lines 210–211: "Because the median RI ER was zero, bivariate correlations could not be used to determine potential predictors of ER resilience." Another reason to consider rate instead of day count.

RESPONSE: As described above, our revised manuscript will include both.

15. Lines 214–215: "Overall, there were multiple environmental controls on metabolic resistance or resilience that were strongly correlated with either GPP or ER, but no significant drivers of both GPP and ER resistance and resilience." This is not supported by the figure or the table.

RESPONSE: We will update to refer to Table 4, which supports this statement.

16. Line 219: "Notably, ER was more resistant than GPP (Figure 1)." Figure 1 is a conceptual figure and does not support this statement.

RESPONSE: We will remove the citation to Figure 1 and replace with a citation referring readers to Figures 5 and 7.

17. Line 239–240: "In assessing metabolic responses and recovery from smaller flow events relative to the dynamic equilibrium of metabolism at baseflow, we found some of the shortest metabolic recovery intervals recorded in the literature (Figure 8; Table A1)." Do these other studies use the exact same methodology as you? How are they comparable? Are they similar sized streams? You should compare and contrast more here.

RESPONSE: We will revise to include types of sites and landscapes in our discussion. The methods we used to generate this graph (from reported metabolism rates in papers) were standardized. We selected calculations/metrics that could make use of the most publications that included metabolism during high and low flows. Metabolism modeling methods were similar among projects.

18. Line 259–260: "Contrary to our predictions, the size of the most recent antecedent flow disturbance had a positive relationship with M GPP and M ER (Figure A19)." Where is this prediction?

RESPONSE: We will articulate this prediction more clearly in the introduction when we revise to highlight that the aim of this paper is not only to address subsidy stress hypotheses and differences between GPP and ER, but also to understand which environmental variables are related to how metabolism responds to and recovers from higher flow events.

Technical Corrections:

1. Equation 1 (Line 110) seems boiler-plate and unnecessary.

RESPONSE: While it is indeed a commonly used (but often slightly modified) equation, we respectfully disagree with the reviewer's opinion that is unnecessary. We strive to ensure that our work is understandable and repeatable without requiring readers to visit many other papers to understand our methods.

2. There are extra parentheses in Figure 2c description for "((m d ))"

RESPONSE: Thank you for catching this. We will remove the extra parentheses.

3. Figure 3 should describe what the error bars are on the GPP estimates.

RESPONSE: We will update our figure legend to define error bars for GPP.

4. Lines 163–164: "Quantifying how different antecedent conditions induce variable responses from GPP and ER is critical to furthering our understanding of stream ecosystem responses to flow disturbances." This belongs in the Introduction, not the Methods.

RESPONSE: We will move this method justification to the end of the introduction where we talk about overall project objectives that are not limited to subsidy-stress analyses.

5. Lines 167–168: "Antecedent medians for turbidity were estimated from seven days

prior due to missing sensor data." This is not clear, please explain what this means. There was always missing data for turbidity within the three days prior to an event? I can't imagine turbidity changes very much at baseflow.

RESPONSE: We had to remove poor-quality data from the turbidity dataset and chose to set methods that would accommodate the most storms possible for our analysis. We compared the outcome of changing the days prior for events with turbidity data available for both 3- and 7-day analyses and found no difference in the results. We will include that information in our revised manuscript to clarify this difference.

6. Lines 190–191: "Three of 15 flow events stimulated ER, 5 repressed ER, and ER did not deviate from the antecedent equilibrium for 7 events (i.e., M ER was 0)." It's more common to use numerals for numbers greater than 10, and to spell the numbers out for numbers less than 10.

RESPONSE: We will write out lower numbers per journal guidelines.

---

## Author Comment (AC2) · 17 Dec 2020

With the exception of the "GENERAL RESPONSE", all of our response text sections begin with "RESPONSE" immediately following reviewer comments.

GENERAL RESPONSE:

This text is included in both responses to reviewers, with specific responses to reviewers below.

Here we respond more generally to questions about why/how we selected isolated flow events and the resulting number of events suitable for our analyses (n = 15 events over

5 years). We emphasize that we focused on quality over quantity when selecting for and analyzing stream metabolism before, during, and after high flow events. Our methods were chosen to address a lingering knowledge gap in our understanding of ecosystem processes: how biological processes (gross primary production and ecosystem respiration, GPP and ER) respond to and recover from discrete higher flow disturbances during storms, how those two processes compare to one another, and which environmental drivers may best explain these dynamics. Potential metabolic responses include subsidy (increasing rates due to higher substrate concentrations), stress (decreasing rates due to physical or chemical disturbances), or no change, which allows for our work to build on concepts fundamental to biogeochemistry and ecology. An additional knowledge gap is how different processes (i.e., GPP, ER) may respond differently to high flows. How we chose to quantify changes in metabolism during higher flow acknowledges the "pulsing steady state" of ecosystems in a novel way. In our revised manuscript, we will better introduce and identify how and where these different concepts apply to, inform, and are answered by our research.

The goal of this work was to assess how metabolism responded to and recovered from higher flow events that were also isolated flow events. Indeed, this decreased the number of suitable events for analysis. But our choice of methods allowed us to focus on response/recovery to discrete disturbances and avoid biased comparisons of pre/during/post multiple high flow (but not isolated) events that encompass time periods that are long enough (e.g., weeks) where pre/post comparisons are less meaningful. Perhaps we could have selected a more pristine stream with less flashy hydrology at the start of this project, but another motivation of our work is to better understand processes in less pristine ecosystems (historically understudied because they are more challenging sites to obtain high-quality metabolism estimates from, another factor that decreased the number of events with appropriate data for our analysis). Despite having "only 15 events", most past analyses included a similar or fewer number of events (e.g, n=10 in Reisinger et al. 2017) over a shorter time period. Our work fills in substantial knowledge gaps: we analyzed across seasons (not only summer months or a short

sensor deployment period) and high flow magnitudes (not only base flow or the highest flow disturbances).

After all appropriate QA/QC measures, we had 1375 days of metabolism estimates over 5 years (which were reported in full in O'Donnell & Hotchkiss 2019 Water Resources Research). To calculate resistance and recovery, we needed consecutive days of high-quality metabolism estimates, which further limited the number of high flow events appropriate for our analyses. For example, in 2016 there were 52 (out of 352) days with quality-checked sensor data that had a 50% flow change relative to the day prior. After looking at these 52 storms and selecting those that had 3 days before and 3 days after without any other flow events, we had 12 that were isolated. After quality-checking our metabolism estimates for all of those days, we had 4 high flow events from 2016 that passed all quality-checking steps required for this analysis.

/ end of GENERAL RESPONSE

REVIEWER 2:

General comments: The manuscript bg-2020-304 "Resistance and resilience of stream metabolism to high flow disturbances" by O'Donnel & Hotchkiss analyzes in a third-order stream the response of Gross Primary Production (GPP) and Ecosystem Respiration (ER) altered by disturbances such as isolated high flow events. The study is relevant as it is based on a long-term monitoring (5 years) of GPP and ER, which is critical to decipher seasonal and multiyear variability of stream ecology in the context of climate change. Overall, I found the approach of the study interesting but the authors should explore their dataset further, therefore I suggest major revisions.

RESPONSE: Thank you for your positive impression of our work. We've responded to your suggestions for further explorations below.

Major comments: I was surprised that the authors did not discuss about in-stream net ecosystem production (NEP). NEP is critical to decipher stream ecology as it does

indicate whether an ecosystem is fixing more C than is respiring. The authors showed that ER has higher resistance and resilience in comparison to GPP, thereby should shifted NEP towards heterotrophy (decrease of the GPP:ER ratio). I believe it would be very interesting for the reader to understand/know how NEP is affected by high flow events. I suggest adding figures and discussion about NEP.

RESPONSE: While our aim was to focus on the response and recovery of specific processes (GPP versus ER), we agree that changes in the balance between GPP and ER (i.e., NEP) are an interesting angle for discussion. Most, if not all, of the discussion about changes to NEP will be similar to ER, so we will also be careful to avoid repetition in places where we add discussion related to NEP. We will also refer to our recent paper discussion overall patterns in discharge, ER, GPP, and NEP where this was one of our primary objectives (O'Donnell & Hotchkiss 2019).

The dataset used by the authors is extended in time but the paper lacks of seasonal variability analysis. How GPP, ER and NEP, resistance and resilience are affected by seasons and by year-to-year variability. Indeed, temperature effect on stream metabolism is usually significant. The authors needs justify that the variability induced by the temperature does not overcome the variability induced by flow events. In the revised paper, I suggest the authors adding a figure such as GPP vs ER with points colored according to seasons or river flow.

RESPONSE: This is an interesting question that we hoped to pursue, but one that we found to be beyond the scope of something we could discuss with certainty because of the few isolated storm events with appropriate data and model output for our full analyses (n=15), thus limiting our ability to make concrete conclusions related to seasonality. Despite that, we were able to discuss patterns of GPP and ER more broadly (not for specific flow events) in O'Donnell and Hotchkiss 2019, and will better integrate results from that analysis throughout the paper.

In low order streams, GPP and ER are affected by groundwater inputs, as groundwater

inputs are usually significant in such streams. Groundwater exhibit usually low oxygen concentration, which may be problematic when GPP and ER are based on oxygen monitoring. Inputs of low-O2 groundwater in stream can overestimate ER and under-estimate GPP. However, the equation 1 does not take into account groundwater inputs. Why? Oxygen measurements during high flow, especially in low-order streams, can give erroneous values, so are the authors sure to measure appropriate values during the high flow events.

RESPONSE: We understand the challenges associated with groundwater inputs and metabolism modeling assumptions (e.g., Hall and Hotchkiss 2017). We will revise the methods to state that we did not see evidence for groundwater inputs in our study reach using conservative tracer additions (not part of this study, but part of other work at this site).

There is some variability in day-to-day metabolism rates; therefore, I do not under-stand why the authors took the maximum or minimum value of GPP (or ER) from the antecedent range to estimate the resistance. I believe that the median or the mean would be more appropriate. In addition, why the authors used 3 days as the antecedent range. Is it arbitrary?

RESPONSE: Thank you for the opportunity to clarify this. We will update our meth-ods to ensure readers understand that analyzing the 3 day antecedent range was not arbitrary, it was the most appropriate trade-off between keeping our analysis to iso-lated flow events (the aim of our study) but including more than one or two days to estimate the dynamic range of prior conditions and metabolic rates (e.g., "pulsing equi-librium", Figures 1,3). If we used >3 days, we would have fewer flow events to analyze. If we used <3 days, we do not have as much information about prior conditions and metabolic rates with which to track metabolic responses to changing flow. We chose the range because this captured the full range of average metabolism estimates in ways that summarizing pre-storm rates to means or medians would exclude.

I have concerns on how the isolated flow events are selected. Indeed, in the figure 2A, I observe that only few high flow events (15 events in 5 years) are actually selected by the authors. In the figure 2C, the authors did not provide statistical analysis on the difference of cumulative daily discharge between all days and isolated flow events. Is it statistically different? Visually, it seems not, considering the high range of cumulative daily discharge during "all days". If it is not statistically different, it means that the disturbance is the same in both groups. Is there a way for the authors to arbitrary select a greater number of high flow events? As examples, the authors could use maximum daily discharge vs cumulative daily discharge or the change in discharge from pre- to peak-storm flow. By the way, I do not understand why the authors wants to select isolated flow events rather than all high flow events. I believe that estimating resistance and resilience in each high flow events would be much more robust. In addition, the paper aims to study ecosystem response to high flow events, but the paper do not contain figures showing the relation of river flow versus stream metabolism. What is the relationship between river flow and GPP, ER, NEP, resistance and resilience? Resistance represents the change in GPP (or ER) during a change in river flow, so maybe it would be interesting to show $\triangle$GPP (or ER) with $\triangle$Q?

RESPONSE: Please see the general response at the top of this document for more information about how we prioritized quality over quantity in selecting isolated flow events. We will assess differences among isolated flow events and the flow of prior days used for comparison. Figure 2C shows that the isolated flow events we were able to use for this analysis were typical of the range of flow changes we see at this site, and should thus give appropriate insights into the range of potential changes to GPP and ER due to flow disturbances. As mentioned above, we will refer readers to O'Donnell and Hotchkiss 2019 for the paper that discusses all of the flow and metabolism data (the 2019 paper did not analyze how metabolic rates changed during and after isolated flow disturbances, which is the objective of this paper).

I the discussion section, I do not feel that the authors fully responds to their four hy-

potheses. How can the authors responds to H2 where they actually do not show carbon or nutrients measurements? H0 is strongly dependent on how you arbitrary selected the flow events. To my point of view, with their study design (unless the authors have measurements of carbon and nutrients) the authors can discuss only about H1 and H3. In addition, I also suggest rewriting the Discussion section in a more logical sense following the order of their hypotheses.

RESPONSE: We agree, and as mentioned in our response to Reviewer 1, will revise our manuscript to better align the introduction and discussion topics. While we do not have high-frequency concentrations of carbon and nutrients to compare with patterns in metabolism RE: H2, we note that if metabolic rates did follow a hump-shaped curve as predicted by subsidy-stress, the changes in rates are more appropriate to test metabolic responses than changes in concentrations, as the concentrations themselves have already been altered by any carbon and nutrient uptake that occurred before the sampling point (i.e., concentrations may reflect potential for process, but we cannot know what is missing from carbon and nutrient pools if it's already been removed from the water by biota).

Minor Comments:

L.1: Please, add somewhere in the abstract the ranges of ER, GPP, NEP, resistance and resilience.

RESPONSE: Good idea. We will look for opportunities to optimize the inclusion of summary results within the character limits for the abstract.

L.10-11: You defined the metabolic resistance as the magnitude of departure from the dynamic equilibrium during antecedent lower flows, so why using the words "ER magnitude of departure" to refer to resistance. Better used the word resistance and resilience throughout the text once you have defined those words.

RESPONSE: We wanted to be true to the method used when referring to specific

analyses and results, but will review opportunities to streamline terminology throughout the manuscript to clarify when we are talking about specific metrics and when we are referring to broader topics.

Please add also in the abstract that more ER or GPP is resistant less the magnitude of departure is large.

RESPONSE: We will update the abstract to highlight these results.

L. 69: It is strange to start with H1 and finish with H0

RESPONSE: We will re-order to start with H0, no change, in our revised manuscript.

L.81: Usually precipitation is in mm

RESPONSE: We will change to mm in our revised manuscript unless the journal prefers different units.

L.90: How did you calibrate the different sensors, and how often did you check the calibration?

RESPONSE: Sensors were calibrated every 2-4 weeks (line 96) according to best practice recommendations from the manufacturer or, in the case of the PME dissolved oxygen sensor, with Winkler titration checks of our 100% and 0% calibration solutions. We will add this information about sensor calibration in our revised manuscript.

L.93: can you add the weather station on the figure A1. The figure A1 needs a scale, a geographic footprint.

RESPONSE: Great suggestions. Will will note the location of the weather station in Figure A1 and add a scale bar. We are not sure what the reviewer means by geographic footprint, but perhaps that will be resolved with the scale bar.

L105: Please, specify that you works with gas exchange coefficient not gas exchange velocity.

RESPONSE: We will specify this in our revised manuscript.

L.110: How did you measure the PAR? How did you calculate the average depth?

RESPONSE: Both of these questions are addressed in lines 92-96.

L.119: What are the values of K?

RESPONSE: All K estimates are in our Supplementary Data file of daily metabolism estimates. K for the high flow days analyzed are in Table 2. We will summarize the mean, median, and range of K estimates for Stroubles Creek in the main text of the revised manuscript.

L.133: Please define Qi

RESPONSE: Thank you for catching this. We will clarify in our revised manuscript. $Q_i$ is the discharge of the day of interest and $Q_{i-1}$ is the discharge of the day prior.

L.167:169: For the different variables other than GPP and ER you used the medians from three days prior the flow event for correlations, but for resistance you used the maximum or minimum GPP or ER before the flow event. I believe it would be robust to use the same methods.

RESPONSE: Great point. We used GPP and ER range to account for the variability in metabolism (and to be consistent with how we assessed metabolism for other analyses), but will test for any differences in using medians in a revised analysis for manuscript revisions.

L.180: Is the cumulative daily discharge statistically different between isolated events and other days?

RESPONSE: We interpret this question as asking whether discharge on high flow days for an isolated event was different than lower flow days before/after. We did not test for differences, but can include that analysis in our revised manuscript and will refer to Figure 2 for the visualization of differences.

L.182-185: As mentioned in the major comments please showed how GPP, ER and NEP are affected by seasons and river flow.

RESPONSE: In our revised manuscript, we will remind our readers to refer to O'Donnell & Hotchkiss 2019 for the relationships between Q, GPP, ER, and NEP and for different seasons. We will also better integrate those findings into our revised discussion.

L219 Where can I see that ER was more resistant than GPP. It is on a daily basis? Yearly basis? Multi-year basis? Please give some details, some stats should be applied. Figure 1 do not show your results.

RESPONSE: We will change this citation to direct readers to Figure 5 in our revised manuscript.

L.228: Same comments 228-230: Can you show some results confirming what you stipulate? In the table 4 turbidity seems weakly correlated with resilience of ER and GPP.

RESPONSE: It was weakly correlated across the 15 events appropriate for our analyses. We address this more fully in O'Donnell & Hotchkiss 2019 with daily metabolism and turbidity data and will expand our discussion based on the 2019 paper in our revised manuscript.

254: The authors have a dataset representing 5 years of monitoring so why they cannot answer to this question, at least partly?

RESPONSE: This was beyond the scope of this project and more appropriate for sites with more high flow events that conform to sensor and metabolism QA/QC, hence the call for 'future analyses'.

Figure 5: I am not convinced by this figure.

RESPONSE: It was unclear from the reviewer comment why this figure is not convincing, but Reviewer 1 was also unenthusiastic about this presentation! As described

in our response to Reviewer 1, we believe this is an important way to visualize and compare results between GPP and ER, but will explore additional ways to show these results before committing to the figure in a revised manuscript.

Figure 6: Same data as in the table 4, perhaps not relevant.

RESPONSE: We will move either Table 4 or Figure 6 to the appendix in our revised manuscript.

Figure 8: Is there a better way to present these results?

RESPONSE: We are not sure what about this graph did not work for the reviewer, so we do not know how best to respond to this comment. Because we were limited in the data collected and reported by other studies, wanted to be as inclusive as possible, and thought placing our work in the context of other flow-metabolism studies was important, this was the best format to calculate and highlight as many metabolic responses and recovery intervals across studies as possible.

Table 4: Please indicate the p-values, Indeed, two parameters can have a correlation coefficient greater than 0.5 but they are still not correlated together if the p-value is greater than 0.05.

RESPONSE: We will include p-values in our revised manuscript.

Figure A4 to A18: In each isolated flow events: GPP, ER and Discharge can be combined in one figure with 3 axis

RESPONSE: They can, but we prefer to avoid multiple axes on plots whenever possible to avoid potential misinterpretations of data.

Figure A20: To my point of view a figure such as this one showing the seasonal variability of the different parameters (GPP, ER, NEP, reÌ₃sistance and resilience) is important and must appear in the main text.

RESPONSE: Please see our comments above RE: challenges with assessing differences among seasons (we do not have the data power to do this well). We will make sure we more clearly refer readers to O'Donnell & Hotchkiss 2019, where daily metabolism data (not a subset of data focused on analyzing high flows) are graphed with different symbols for each season.

―――――――――――――――――――

---

## Author Response (AR1)

*This is an updated response to reviewers after receiving a decision of "Reconsider after Major Revisions" and comments from the Associate Editor.*

*With the exception of the "GENERAL RESPONSE" directly below, all of our response text sections begin with "RESPONSE" immediately following reviewer comments. All Page-Line numbers in our responses correspond to new line numbers in the revised manuscript, not the original manuscript or the draft with track changes.*

*Thank you for the opportunity to respond to reviewers and revise our work!*

**GENERAL RESPONSE:**

Here we respond more generally to questions about why/how we selected isolated flow events and the resulting number of events suitable for our analyses (n = 15 events over 5 years). We emphasize that we focused on quality over quantity when selecting for and analyzing stream metabolism before, during, and after high flow events. Our methods were chosen to address a lingering knowledge gap in our understanding of ecosystem processes: how biological processes (gross primary production and ecosystem respiration, GPP and ER) respond to and recover from discrete higher flow disturbances during storms, how those two processes compare to one another, and which environmental drivers may best explain these dynamics. Potential metabolic responses include subsidy (increasing rates due to higher substrate concentrations), stress (decreasing rates due to physical or chemical disturbances), or no change, which allows for our work to build on concepts fundamental to biogeochemistry and ecology. An additional knowledge gap is how different processes (i.e., GPP, ER) may respond differently to high flows. How we chose to quantify changes in metabolism during higher flow acknowledges the "pulsing steady state" of ecosystems in a novel way. In our revised manuscript, we will better introduce and identify how and where these different concepts apply to, inform, and are answered by our research.

The goal of this work was to assess how metabolism responded to and recovered from higher flow events that were also isolated flow events. Indeed, this decreased the number of suitable events for analysis. But our choice of methods allowed us to focus on response/recovery to discrete disturbances and avoid biased comparisons of pre/during/post multiple high flow (but not isolated) events that encompass time periods that are long enough (e.g., weeks) where pre/post comparisons are less meaningful. Perhaps we could have selected a more pristine stream with less flashy hydrology at the start of this project, but another motivation of our work is to better understand processes in less pristine ecosystems (historically understudied because they are more challenging sites to obtain high-quality metabolism estimates from, another factor that decreased the number of events with appropriate data for our analysis). Despite having "only 15 events", most past analyses included a similar or fewer number of events (e.g, n=10 in Reisinger et al. 2017) over a shorter time period. Our work fills in substantial knowledge gaps: we analyzed across seasons (not only summer months or a short sensor deployment period) and high flow magnitudes (not only base flow or the highest flow disturbances).

After all appropriate QA/QC measures, we had 1375 days of metabolism estimates over 5 years (which were reported in full in O'Donnell & Hotchkiss 2019 Water Resources Research). To calculate resistance and recovery, we needed consecutive days of high-quality metabolism estimates, which further limited the number of high flow events appropriate for our analyses. For example, in 2016 there were 52 (out of 352) days with quality-checked sensor data that had a 50% flow change relative to the day prior. After looking at these 52 storms and selecting those that had 3 days before and 3 days after without any other flow events, we had 12 that were isolated. After quality-checking our metabolism estimates for all of those days, we had 4 high flow events from 2016 that passed all quality-checking steps required for this analysis.

A revised version of this response is now distributed within our revised text, as requested by the Associate Editor (we know they focused on adding some of this information to our methods, but we found a few other places appropriate for some of this response as well!). LINES: 83-87, 160-173, 292-300, 365-374.

//

**Comments from Editor after reviewing our pre-revision response to reviewers**

**I found very clear your response of the comments by the two reviewers on the way you have processed your data ("response to general comments"). This argumentation must appear in the Mat&Met of your revised MS. As said before, one difficulty for this decision came from your responses to the other comments that were vague in general and not always acceptable as valuable scientific arguments. This was the case for many responses to comments by both reviewers, and as editor, I would need more guaranties on your capacity to deny or integrate into the revised MS all these comments. In other word, the fact that you will "keep in mind" the comments when rewriting the MS is not a guaranty that the scientific content of these comments will be adequately integrated. More precise explanations of the planned changes would have been necessary at that step. This is why I suggest that you submit a new detailed and precise response to the reviewers together with your revised MS, as it would facilitate and accelerate the second round of review.**

**When making your revision, I also recommend that you keep focussed on WHAT YOUR DATA ARE TELLING, using simple and already well-defined hydrological terms (rather than "concepts") like "base flow", "flood", "storms events", "transient" versus "steady state", eventually "pulsing", and so on… I feel this MS is not necessarily the appropriate place for defining new "revolutionary" concepts. More maturation would probably be necessary for that. For instance, "pulsing steady state" (in your response to general comments) sounds like an oxymoron to me. There is no room for oxymoron in science.**

RESPONSE: We appreciate your guidance and encouragement to revise this manuscript. We have included a revised version of our general response to reviewers in our Methods section, as you suggested. And we have updated our manuscript and responses to reviewers with specific

changes made through our revision, as requested. That said, we would like to clarify two things in response to the editor comments we received: (1) We were responding to reviewers given the following instructions: "*Please note that your revised manuscript should not be prepared at this stage.*" We were not in a position to make more precise responses to some reviewer comments because we had not revised the manuscript yet. We apologize if we misunderstood the author instructions for this different (to us) reviewing system. We have now specifically stated what we changed and why in an updated response to reviewers below. (2) We used these revisions as an opportunity to clarify terminology and avoid using slightly different or interchangeable terms that could be confusing, especially with respect to hydrology. We do note, however, that we did not attempt to define "*new "revolutionary" concepts*" in our paper. For example: we included "pulsing steady state" to acknowledge prior work on ecosystem dynamics by Odum et al. 1995 "Nature's pulsing paradigm" which used the same term (thus noted as "*sensu* Odum et al. 1995" in our manuscript). Because ecosystem ecology shares many techniques, research motivations, and concepts with biogeosciences, we believe it is appropriate and necessary to use and acknowledge concepts from ecology where they apply. Revisions were made to ensure readers can understand the terms and citations in our revised manuscript, including taking the suggestion of one reviewer to discuss variability at lower flow as "dynamic ambient equilibrium".

//

**REVIEWER 1:**

**General comments:**

**The authors analyse the effects of 15 isolated storm events on stream metabolism, focusing on subsidy-stress hypotheses and drivers of response and recovery. The authors make use of a five-year high-temporal resolution dataset to address their four hypotheses. While the general ideas and approach are interesting and worthy of study, the current manuscript requires major revisions before publication. Some of the suggested revisions are substantial changes to analysis and fall more into the "reject and resubmit" category, but the authors of course can provide adequate justification for not conducting these changes.**

RESPONSE: We appreciate this reviewer's feedback acknowledging the novelty of our work, encouraging us to clarify our research objectives, and highlighting areas where we can better justify the methods used to address current knowledge gaps related to how ecosystem metabolism responds to and recovers from high flow events. We respond to specific suggestions below.

**Specific comments:**

**Major:**

**1. After reviewing the approach and the data, and because it underlies all results in the paper, I think the authors need to present a stronger rationale (which currently does not exist)—or change the approach— for how they arrived at their use of arbitrary discharge thresholds (e.g., 50%, 10%) in discriminating the "isolated events". An option could be a simple sensitivity analysis. Additional rationale should support the use of cumulative daily discharge, as opposed to other commonly used metrics in hydrology for event detection. I further think using cumulative discharge may be obscuring some results and indeed missing many events. I first started down this path of inquiry because only 15 isolated events over five years seemed to be a small sample size. Would the same thresholds result in different events if applied to depth or even comparing maximum daily discharge as opposed to cumulative daily discharge? This, I realize, may be a bit of a task because it requires an entirely new analysis, but I think that the authors need to consider this route and defend their assertions more fully. If more events could be used based on a simple adjustment like threshold choice, there could be a much more robust sample size to draw inference from, and would make this a much stronger paper.**

RESPONSE: We responded to concerns about "only 15 isolated events" in our general response at the start of this document, which has also been modified for addition to the end of the methods section in our revised manuscript, but add a bit more detail here. Again, what we sacrificed in quantity we gained in quality. A full dataset for a single day required quality-checked sensor data as well as metabolism estimates that passed all QA/QC steps, which means some isolated events were excluded from our analysis. We excluded values of physicochemical parameters that were below the 1% or above the 99% quantile and removed physicochemical measurements we knew were inaccurate due to sensors being out of the water during low flow or not working properly (e.g., turbidity at zero). This is covered in this text in the results and in greater detail in O'Donnell and Hotchkiss 2019 which we reference in our methods.

In our revised manuscript we clarify how and why we selected isolated storms. The following text is now in our revised methods (LINES 160-164): "The designation of 50% change in flow for high flow events ensured analyzed events were outside of the range of baseline flows. We defined a flow event as >10% change in Q when comparing the high flow changes to prior metabolic rates, as smaller changes in Q may still influence metabolism. In testing different thresholds of flow change and different discharge metrics, we settled on our current method to optimize the delta Q thresholds resulting in the highest number of quality-checked events, while ensuring differences between classifications of ambient stream flow and higher flow events."

**2. Similarly, I think this paper could be much stronger by including as many events as possible, regardless of whether they are "isolated". As discussed in the Introduction, the pulsing fluxes, whereas the presented approach discounts them. I understand that perhaps the authors were particularly interested in capturing "resilience" metrics, which may require a period of calm after the storm, so to speak. But, one can imagine a much richer analysis if, for example, the authors calculated some kind of "resistance" metric for as many events as possible, but parsing which ones were preceded by large events.**

**And, for "resilience", the authors could still calculate the time to return to pre-(initial)event conditions, but just parse which of these "initial" events had subsequent events. Without much effort, the authors could even estimate the subsequent rate of events and its influence on "resilience". One can imagine a figure of, for example, ΔGPP vs. ΔQ where points are colored by their recovery time and sized by their subsequent rate of events. I mention these suggestions because the current methods seemed disingenuous in taking an arbitrarily "neat" approach to this potentially very fruitful test of the pulsing paradigm. Another important point in this regard is how to take into account when a rain event occurs during the day. For example, if a rain event occurs at 23h, it seems like your approach considers its effect for the previous day, when it is probably more appropriate to consider its effect starting for the following day. (This is understood by the authors in their approach in section 2.5.1, but their simple correlation approach does not effectively get at this idea). The current approach also likely discounts many possible events for this reason. I am aware that this may be a big ask of the authors, and, if different routes are taken, I still suggest that they provide stronger rationale for the (apparently) arbitrary decision for identifying events.**

RESPONSE: We respectfully disagree that our decisions were "arbitrary". As the reviewer acknowledged, our focus was to understand metabolic resistance to and recovery from isolated high flow events. Furthermore, we were interested in assessing these dynamics in a hydrologically flashy stream draining a heavily modified landscape. We agree other topics are also interesting and look forward to seeing more research addressing these topics, but they are not the aim of this paper. We have clarified our research objectives in our revised manuscript. Again, a higher-level response to this and #1 above is in our overall summary at the top of this document.

**3. In the same vein, the authors should provide some kind of justification or sensitivity analysis for both of their critical choices in calculation of their resistance/resilience metrics. The first is the choice of "...three days prior to define a range of antecedent metabolism for each isolated flow event." (Lines 140–141).**

RESPONSE: This is a great point and one we have answered in our revised manuscript. "Because flow was so variable, we chose three days to balance best practices from past work on metabolic responses to storms (e.g., four days of prior stable baseflow, Reisinger et al., 2017) while ensuring we could analyze as many events with appropriately quality-checked data as possible." (LINES 202-204)

**The second is the choice of defining "X prior [as] the maximum or minimum value of GPP or ER from the antecedent range..." (Lines 147–148). Why not the median or mean, which would represent more of the "equilibrium" of the previous period?**

RESPONSE: We wanted to capture the pulse of days prior (Figure 1). The pulse isn't captured in a mean or median, and analyzing the metabolism data in this way can result in estimating a departure from a mean or median that is erroneously considered to be different from baseflow

tendencies when in reality it is within the ambient pulse of the system. We revisited where we highlighted this choice in our methods (we adopted this more conservative method for measuring magnitude of departure and recovery that is more appropriate for variable ecosystems) to ensure this decision is well-described in our revised manuscript.

New text: "We use the maximum or minimum values instead of the median or mean because this approach allowed us to better capture the full range of average metabolism estimates in ways that summarizing pre-storm rates to means or medians would exclude." (LINES 185-187)

**And why not use the most similar previous day in terms of driving forces–in particular, light availability (this seems especially relevant for the recovery interval!). I'm sure the authors considered such options in their initial work, but they need to do more work to convince the audience of their presented approach—or take a different one if the evidence from sensitivity analyses suggests that they should. Both of these choices are major factors in the subsequent analyses because they define the metrics used, and because these choices do not appear to have literature support/precedent, they need to have clear rationale.**

RESPONSE: While an interesting idea, this approach requires too many assumptions about our ability to predict GPP and ER based on light or temperature or other environmental data alone. While our knowledge of metabolism in streams and rivers is growing rapidly, we are not at a point where we can justify how we would select a "most similar previous day" for comparisons between the range of baseline metabolic rates and responses/recovery from high flow events with the environmental data we have for this site.

**4. Lines 154–156: "To quantify the resilience of GPP and ER, we estimated recovery intervals (RI) by counting the number of days until metabolic rates returned to within the range of pre-event values, signifying a return to antecedent dynamic equilibrium (Figure 3)." This is a good illustration of a potential issue/untapped possibility with the current approach. If you look at the data for the event shown in Figure 3, depth increases in that event by approximately 0.12 m, which decreases light availability by approximately 13% (according to exponential attenuation). This is nearly exactly the difference between GPP on 7 February and 9 February, both of which had nearly identical incoming light signals (making them comparable).**

RESPONSE: This is an observation that we explored, but as in our response to point 3 above, did not feel it was appropriate to compare to a similar instead of a previous day to test how metabolism responds to high flow (from prior conditions) and recovers from high flow (compared to prior conditions).

**5. Lines 157–160: "To ensure additional flow events did not obscure the recovery interval of GPP or ER, we stopped counting RI the day before the next event (i.e., if another flow event happened four days later, we stopped counting RI at 3 days), and have noted this in our results as days+ and used different symbols in data figures." Why? As far as I can**

tell, the authors throw these data points out in their analysis, and only reference them in Table 3 (which already uses asterisks to note the issue). Is this to note that the system was on its way to "recovery"? Maybe it would be better to just show a recovery rate, instead of a time, which could result in more data points being included. So, instead of the time it takes to get back to some baseline (which I argue above is a bit arbitrary), you can calculate the rate of increase in GPP over a period (which could be equal to the baseline period that you settle on). Let's say an event occurs and on that day GPP was 5 g O2 m-2 d-1 ; the subsequent days maybe it's 8, then 10 g O2 m d . The rate of increase could then be 2.5 g O2 m-2 d-1 ( (10-5) / 2). Then, even if a subsequent event occurs, you can still compare the rate of increase before that event. A rate also seems like it could be more comparable/scalable across systems in contrast with a number of days. I don't presume to have the best idea here, but I think an approach like the one outlined above could increase inclusion of useful data points, and thereby lead to more useful inferences.**

RESPONSE: This another metric we had explored early on but did not include in our final analysis because it communicated a similar story as RI and was not a metric we were able to compare with other publications that reported changes in metabolism after storms. We include the number of days to recovery, when measurable (yes, the days+ is to note it was on track to recovery, but did not return to the range of baseline metabolic rates before the next high flow event). We kept RI estimates to allow us to compare our results to other studies (e.g., Figure 8) for broader discussion, and because we were not satisfied with the storm-specific standards that would need to be used for dates that did or didn't recover (e.g., how to compare recovery rate if the trajectory finished or did not for different storms, for recovery periods of different length, etc.).

**6. Lines 165–166: "We assessed three categories of potential predictors of metabolic resistance and resilience: antecedent conditions, characteristics of the isolated flow event, and characteristics of the most recent prior flow event." Antecedent conditions and characteristics of the recent prior flow event (especially the latter) are unrelated to any stated hypothesis and appear to come out of nowhere. There needs to be clear rationale in the Introduction that leads the reader to understand why you are doing this.**

RESPONSE: We revised the objectives and hypotheses outlined in the introduction to clarify our interest in estimating metabolic responses to flow change and potential drivers of those responses.

**7. Generally speaking, I had difficulty with the entire Results section, which I think needs a complete rewrite. Some specific details are presented below, but I glossed over several in the interest of time. This section needs to link to stated hypotheses (in the order that they are stated in the Introduction) and test them directly without including spurious tests and weak assertions.**

RESPONSE: Agreed. Thank you for the detailed comments below and the reminder to ensure the introduction and discussion are better aligned. We have revised with these comments in mind.

**8. Figure 5 as presented is not informative. What do the authors want the reader to understand from this figure?**

RESPONSE: We respectfully disagree that showing these results graphically is not informative. We have removed 5B since many of the GPP or ER pairs did not have both metrics that fully recovered, which means we didn't have many data pairs we could plot. Updated Figure 5 plot is pasted below and in the revised manuscript.

[Figure]

Figure 5. Resistance (i.e., magnitude of departure) of gross primary production (GPP) versus ecosystem respiration (ER) in Stroubles Creek, VA. Dashed line is the 1:1 line; the solid line is the linear model fit through all data (p-value = 0.007, $R^2$ = 0.39).

**Is the R based on a linear regression for all of the points or just the black circles? What is the slope of the regression and the p-value? How does the slope compare to the 1:1 lines?**

RESPONSE: R is based on all the points. p-values are included in the results and were added to the figure legend as well (see above).

**The second panel (right, ER vs GPP recovery interval) is not related to any stated hypothesis.**

RESPONSE: We removed this part of Figure 5 from our revised manuscript.

**The text discussing this Figure does not support the points on the figure, particularly for the high stated value of ER stimulation = 0.22 (Lines 189–190: "...The magnitude of departure for ER (M ER ) ranged from -0.59 to 0.22, with a median of 0.").**

RESPONSE: All points were included and the results presented are correct (we checked!), labels on the y axis were off, which has been modified in the updated Figure 5.

**Looking at this figure also raises red flags about how the authors defined stimulation/repression. How do the extremely small changes in magnitude shown here compare to the uncertainty in GPP/ER, which are never discussed or propagated through any of these analyses? For example, is a 1% increase (i.e., M = 0.01) detectable if uncertainty is considered?**

RESPONSE: We note that our metrics were indeed detectable relative to metabolism estimates (with low uncertainty, as shown in Fig 3, A4-A18, and supplementary data files). Consequently, we disagree that this approach "raises red flags".

**The authors should improve this figure substantially, or remove it/leave it as a table. One possible idea is to color or size points based on the event size. Moreover, based on this figure, I am not sure I believe the results on Line 193–195 (italics mine): "Although GPP exhibited stronger responses across isolated flow events than ER, M GPP and M ER were positively correlated (R 2 = 0.39, p = 0.007, Figure 5) and not significantly different (p = 0.06, α = 0.05)." Just an eyeball test makes this seem unreasonable. ER magnitudes on average are about 0.**

RESPONSE: This has been clarified after updating and simplifying Figure 5 (shown above) as well as including summary statistics for $M_{GPP}$ and $ME_R$ in section 3.2: "The magnitude of departure for GPP ($M_{GPP}$) ranged from -0.92 to 0.09, with a median of -0.14 (Table 3; Figure 5). .... The magnitude of departure for ER ($M_{ER}$) ranged from -0.59 to 0.22, with a median of 0.00 (Table 3; Figure 5). "

**9. Figure 6 is not easily understood and appears to simply repeat the information on Table 4 in a cluttered way. What key piece of information is the reader supposed to understand from this? The results of the controls on process response in this section 3.3 is quite difficult to connect with any prior hypotheses and leaves the reader uninformed. There are two figures and a table with only six sentences to describe them in this section. One of the stated hypotheses (H2) is never even formally tested here, and only the resistance metric is tested for H3 (somewhat, in Figure 7), which included both resistance and resilience metrics.**

RESPONSE: We agree that Table 4 and Figure 6 somewhat show the same information, and consequently, we moved Figure 6 to the appendix in our revised manuscript, where it is now Figure A21.

**10. As far as the Discussion and Conclusion, I have many comments, but the issues all stem from previous issues relating to hypotheses, methods, and results. If the authors apply any of my suggested revisions to their approach, they will inevitably have to**

**rewrite these sections. So, I have not provided many specific comments out of the interest of time, but a few key ones are here. I again suggest to organize the Discussion (and entire document) in order of the hypotheses as they are presented and as makes logical sense. As written, the Discussion jumps around in its assertions and ideas. Finally, much of the content in these sections is hypothetical and rhetorical, with little critical analysis of the results actually presented in the manuscript and how they relate to the broader literature.**

RESPONSE: As stated above in response to similar comments, we revised the Discussion and Conclusion with these comments in mind, adding further discussion of our own work and past work on these topics, while taking care to avoid discussing too far beyond the scope of our objectives. We ordered the results and discussion sections according to specific hypotheses and predictions covering: (1) metabolic resistance and resilience and (2) controls on resistance and resilience.

**Minor:**

**1. Ideas of pulsing steady state could be clarified a bit with regard to the study design and terminology throughout. In the Introduction, the authors note "Frequent disturbances generate oscillations that form a pulsing steady state (sensu Odum et al. 1995) that includes ambient variability in processes (Resh et al., 1988; Stanley et al., 2010)." (Lines 21–23). So, flow disturbance regime defines the pulsing steady state of lotic systems. But, the authors then use–incorrectly I think–the periods outside of flow disturbance to define a "pulsing steady state" (or at times, "pulsing equilibrium", like in Figure 1, and "dynamic equilibrium", like in Figure 3, and "antecedent equilibrium" on Line 187), to which they then compare to periods with flow disturbance. The approach is clear, but there is some circular reasoning with respect to the definition of pulsing steady state. I recommend perhaps using different terminology for these two concepts. One idea could be to use something like "ambient equilibrium" for metabolism under baseflow conditions, and "pulsing equilibrium" to refer to the larger scale, (inter)annual behaviour of as originally conceived by Odum. I think these small changes would improve the clarity of the study design and arguments within.**

RESPONSE: Thank you for this suggestion. We have updated the term for the day-to-day variability in metabolism during lower flow periods to "ambient equilibrium" or "dynamic ambient equilibrium" (to avoid reader misconceptions about equilibrium being constant) in Figure 1 and throughout the text of the manuscript. We have included the updated figure 1 and legend below. Additional changes can be reviewed via track changes.

[Figure]

"Figure 1. Potential metabolic responses along a subsidy-stress gradient of stream flow (adopted from Odum et al. 1979). Flow is on the x-axis. The y-axis represents ecosystem metabolism (i.e., gross primary production and ecosystem respiration; GPP and ER), scaled to the same "normal" starting values for comparison, and is broken into four categories as proposed by Odum et al. (1979): (1) subsidy (when flow replenishes carbon and nutrients and metabolism increases), (2) normal (periods of dynamic equilibrium under ambient flow), (3) stress (when ecosystem processes are suppressed by disturbance), and (4) replacement (when there is a severe reduction in metabolism and communities are scoured or replaced). H1-H4 labels correspond to different hypotheses about how GPP and ER may respond differently to flow (H1) and how metabolism might change with flow (H2-H4), and are described further in the main text of the introduction. The inset graph next to the 'normal' bracket depicts how ambient process rates are best represented by a dynamic ambient equilibrium rather than a fixed point of stability (*sensu* Odum et al. 1995)."

**2. Similarly, I do not think that "resilience" is appropriately used throughout the manuscript, first defined by the authors on Lines 59–60: "We can also quantify post-disturbance ecosystem responses by estimating resilience: the time it takes for a process returns to equilibrium following a disturbance (Carpenter et al., 1992)." We have first of all the issue of "return[ing] to equilibrium", which is not so clear based on the previous definition of a pulsed equilibrium that includes disturbance. In a system organized by regular disturbance regimes, the idea of resilience to that same disturbance regime is a bit convoluted. In contrast, the idea of a "recovery interval" to previous ambient conditions is clear and appropriate. Resilience in this context might make more sense if there were alternative metabolic equilibria that the stream could occupy, where each of these equilibria were tolerant to different levels disturbance. Ultimately, this is a choice of language and does not affect the analyses presented and if the authors opt to keep their current choice, I suggest spending some more time to expand these ideas/defend their use out in the Introduction and Discussion.**

RESPONSE: We agree that recovery interval is appropriate for the bulk of our discussion, but disagree that this is not related to resilience as we define it in the introduction. We hope that refining and simplifying the terminology has helped alleviate this specific concern (including the updated figure and text associated with Figure 1 in response to the prior comment); we made those edits with this comment in mind.

**3. Line 73: "(H0) some flow events will not push GPP and ER outside of their pulsing equilibrium." Should this by "(H4)"? Or is this some kind of null hypothesis? Consider renumbering, or placing this at the beginning of the sentence–seems strange to go from 1–3 then back to 0.**

RESPONSE: This was somewhat of a null hypothesis. We changed to H4 in our revised manuscript given other reviewer comments and, as mentioned already above, expanded the final paragraph in the introduction to stress that we did not do this work to only test four alternate hypotheses on a single topic. New LINES for that final paragraph are 73-87.

**4. Lines 70–71: "...(H2) there will be a stimulation of GPP and ER at intermediate flow disturbances due to an influx of limiting carbon and nutrients...". Is this stream known to be limited by carbon and nutrients? What is the timeframe for stimulation? It seems like the influx of carbon and nutrients would pass through the system quite quickly in this small stream, and would not be easily acquired/processed by organisms. In larger systems with long recession curves, I think this perspective can make sense, but this hypothesis does not seem well supported in the Introduction as currently written.**

RESPONSE: We updated the introduction and site information to provide better context for this work. While the speed at which nutrients and carbon travels during storms will increase, the ability of microbes to respond to increases in carbon and nutrients is not limited to larger rivers (e.g., Demars 2019). And yes, the stream is carbon-limited. We have unpublished data on carbon and nutrient limitation at our study site that we now mention as part of the site information in our revised manuscript. New LINES: 96-98 - "Stroubles  Creek  has  been designated  an  impaired  waterway due to high sediment loading and has NO3 concentrations that typically exceed 1 mg/L (O'Donnell and Hotchkiss, 2019); biological oxygen demand in Stroubles Creek appears to limited by organic carbon availability moreso than inorganic nutrients (O'Donnell and Hotchkiss, unpublished data)."

**5. Lines 71–72: "...(H3) metabolic resistance and resilience will change with the size of the event, with larger flow disturbances inducing more stress due to enhanced scour..." The point about scour here seems important. Scour is a function of shear stress, which itself is a linear function of depth. The authors focus on discharge as their subsidy/stress driver, but I wonder if water depth would be more appropriate? Because depth only increases to the square-root of discharge (for a large range of depth-discharge in their Supplemental data), a quadrupling of discharge only results in a doubling of benthic shear stress. I don't expect for the authors to redo any analyses with this perspective,**

**but I do think this kind of information would be useful to include especially in the Discussion so that future works would consider this as well. It also could be used as a future framework to further test the idea of subsidy/stress balance. Depth is a first-order control on both light availability and shear stress at the benthos, making it a more appropriate indicator of stress than discharge.**

RESPONSE: Thanks for this comment. We agree that is another useful way to consider these processes, and while we have not changed our analyses or motivating hypotheses, we have included the following in our revised discussion after original text stating"At some threshold of higher flows that disturb more protected areas within and below streambeds, we expect ER will decline as flow-induced stress exceeds flow-induced carbon and nutrient subsidies." New Text: "Analyses of the interactions between flow-induced changes in shear stress, water depth, and light availability may provide additional insights to tests of predicted subsidy-stress dynamics related to stream metabolism." LINES: 319-320

**6. The light data (first referred to on lines 92–93) appear to be in units of μA according to the supplementary material ("ODonnellHotchkiss_SuppData_ReadMe.pdf", under point "1"). I am not familiar with this unit (is it micro-amperes?) for sunlight, and I think this needs some clarification. The light data in the data file itself appear to range between 0 and 1, but the streammetabolizer model take data in PAR (units μmol m s ), which can be upwards of 1000 by noontime. I'm sure this is not a major issue, but I do not think the results will be replicable as currently presented—those units, if fed into streammetabolizer, will lead to very strange outputs I think. The sensor used (according to O'Donnell and Hotchkiss 2019) is a Campbell CS300, which should output data in typical units like W m .**

RESPONSE: The units for light do not matter because how we use light in the metabolism model (Eq 1) is a ratio of light at time i divided over the sum of light over the entire day: PARi / sumPAR. We realized we needed to correct this in our revised manuscript, as it's noted as PARt, not PARi-dt (Equation 1), and included a statement on how the light data are used in the model: "We note that GPP is multiplied by the proportion of light (PAR) at the previous measurement over total daily light, so the light term in the metabolism model function is unitless." LINES 128-129

**7. Line 140: "To acknowledge the pulsing, day-to-day variability..." I don't think "pulsing" is appropriate or needed here.**

RESPONSE: Agreed. We removed "pulsing" and now state "To acknowledge the ambient day-to-day variability…." LINE 175

**8. Line 152: "...suppression..." please check for the consistent use of suppression and repression (and others) throughout.**

RESPONSE: We were not sure what the reviewer was recommending here, but we checked for consistent use of suppression, declined, or reduced throughout and removed all uses of repression.

**9. In section 3.2 "Metabolic resistance and resilience", it would be very helpful to explicitly organize/label these paragraphs according to your numbered hypotheses from the Introduction. For example, Lines 187–192: There is no directly stated connection between any of the statements presented here and the actual hypotheses.**

RESPONSE: As mentioned above, we updated the hypothesis section of the introduction to include discussion that is not only focused on S-S hypotheses to clarify all of the goals of this work (it was not limited to one set of hypotheses). We now specifically state objectives and predictions we had about metabolic resistance and recovery (LINES 72-87) and match the order of results and discussion to the introduction and methods.

**10. Lines 194–196 bring up another issue with the idea of "magnitude" (italics mine): "M GPP was less than M ER for nearly all flow events, except for one in which M GPP and M ER were both zero and two where M GPP and M ER were both small (Figure 5, Figure 195 A19)." The general idea of magnitude is that is not directional. I would argue that the magnitude of GPP response was greater than that of ER, and that they both had similar directional change (decrease in process magnitude). Consider different language throughout.**

RESPONSE: Thanks for sharing this perspective. We agree that we can consider the absolute value of the magnitude of change in addition to whether that change was positive or negative. However, we have chosen to keep magnitude calculations as they were because whether that change was positive or negative (e.g., a stimulation or suppression of ER), is also important information from the perspective of stream processes. We believe the updated figure 5 does a better job of making these comparisons.

**11. Lines 198–199: "Similarly, the only other event that stimulated GPP (M GPP = 0.03) had no ER response, suggesting many flow disturbances may decouple GPP and ER." This seems like an unsupported assertion (which should be in the Discussion, if anywhere) based on one event with an extremely small signal.**

RESPONSE: We refined the language in the results to better reflect the magnitude of responses/signals. We moved the "suggesting many flow disturbances may decouple GPP and ER" to the discussion to provide additional support for this statement from other results.

**12. Table 3: n/a is not clearly defined.**

RESPONSE: Thank you for catching this. We now define n/a in the updated legend for Table 3. It is when ER or GPP did not deviate from the antecedent range, and therefore had no recovery interval.

**13. Lines 208–209: "Although GPP and ER are linked processes, the variables that were moderate or strong predictors of resistance or resilience (r > 0.5)." Why is 0.5 the threshold for being a strong predictor? That's only 25% of the variance explained.**

RESPONSE: In our opinion, explaining 25% of the variance of ecosystem processes is notable - ecosystems are variable and explaining a high proportion of variance is rare. But we think we understand where the reviewer was coming from and revised with this comment in mind - we reviewed all descriptive wording used and edited our text to make sure it matches the thresholds we set in our statistical analyses. We also added p values to Table 4 at the request of reviewer 2. We cite references supporting our decision for these thresholds in the manuscript: "We interpreted correlation strengths as: negligible (r = 0.0-0.3), low (0.3-0.5), moderate (0.5-0.7), or high (0.7-1.0) (Hinkle et al., 2003)." LINES 218-219

**14. Lines 210–211: "Because the median RI ER was zero, bivariate correlations could not be used to determine potential predictors of ER resilience." Another reason to consider rate instead of day count.**

RESPONSE: As described above, our revised manuscript includes both.

**15. Lines 214–215: "Overall, there were multiple environmental controls on metabolic resistance or resilience that were strongly correlated with either GPP or ER, but no significant drivers of both GPP and ER resistance and resilience." This is not supported by the figure or the table.**

RESPONSE: We updated to reference Table 4, which supports this statement.

**16. Line 219: "Notably, ER was more resistant than GPP (Figure 1)." Figure 1 is a conceptual figure and does not support this statement.**

RESPONSE: We removed the citation to Figure 1 and replaced with a citation referring readers to Figures 5 and 7.

**17. Line 239–240: "In assessing metabolic responses and recovery from smaller flow events relative to the dynamic equilibrium of metabolism at baseflow, we found some of the shortest metabolic recovery intervals recorded in the literature (Figure 8; Table A1)." Do these other studies use the exact same methodology as you? How are they comparable? Are they similar sized streams? You should compare and contrast more here.**

RESPONSE: We revised to include types of sites and landscapes in our discussion. The methods we used to generate this graph (from reported metabolism rates in papers) were standardized. We selected calculations/metrics that could make use of the most publications

that included metabolism during high and low flows. Metabolism modeling methods were similar among projects.

New text includes: "Of the other stream metabolism studies that provided results suitable to include in our comparison of \% reduction in GPP or ER and metabolic recovery intervals ($RI_{GPP}$, $RI_{ER}$; Figure 7), two were from streams draining more heavily urbanized watersheds (Reisinger et al. 2017, Qasem et al. 2019), and one was from a stream draining an agriculturally-dominated landscape (Roley et al. 2014). It appears streams draining more urbanized landscapes have larger reductions in metabolism and longer recovery intervals after higher flow disturbances; additional analyses at sites covering a range of land cover types and flow regimes will provide exciting opportunities to see if the trends in Figure 7 are more broadly applicable." LINES 307-313

**18. Line 259–260: "Contrary to our predictions, the size of the most recent antecedent flow disturbance had a positive relationship with M GPP and M ER (Figure A19)." Where is this prediction?**

RESPONSE: We now articulate this prediction more clearly in the introduction: "In addition to testing the subsidy stress hypotheses and differences in how GPP and ER may respond to and recover from higher flow events (Figure 1), we also analyzed the relationships between environmental variables and metabolic responses, including those prior to flow disturbances that may influence how stream microbial communities respond to flow changes. We predicted recent disturbances might make microbes more vulnerable and less resistant to the next high flow disturbance." LINES 80-81

We also more clearly articulate what this prediction was in the discussion section. New text: "Contrary to our prediction that past scouring might reduce future resistance to disturbances…" LINES 328-329

**Technical Corrections:**

**1. Equation 1 (Line 110) seems boiler-plate and unnecessary.**

RESPONSE: While it is indeed a commonly used (but often slightly modified) equation, we respectfully disagree with the reviewer's opinion that is unnecessary. We strive to ensure that our work is understandable and repeatable without requiring readers to visit many other papers to understand our methods.

**2. There are extra parentheses in Figure 2c description for "((m d ))"**

RESPONSE: Thank you for catching this. We removed the extra parentheses.

**3. Figure 3 should describe what the error bars are on the GPP estimates.**

RESPONSE: Error bars did not correspond to standard error from the metabolism model (error was very small and not visible, as you can see in the appendix figures). We used this as an opportunity to clean up and update this plot accordingly. Updated figure 3 and legend is below.

[Figure]

"Figure 3. Example calculations of metabolic resistance (M) and resilience (RI). Daily gross primary production (GPP) was estimated for the three days before, one day during (grey square), and two days following an isolated flow event that occurred on 2017-02-09. Dashed horizontal lines represent the maximum and minimum GPP estimates from three days prior to the flow event. In this case, GPP declined, and the magnitude of departure (M with arrow) is the difference between minimum GPP estimate from the antecedent range (bottom dashed line) and GPP during the event. After this flow event, GPP recovered to its antecedent range on day two."

**4. Lines 163–164: "Quantifying how different antecedent conditions induce variable responses from GPP and ER is critical to furthering our understanding of stream ecosystem responses to flow disturbances." This belongs in the Introduction, not the Methods.**

RESPONSE: We agree that this was not appropriate for the Methods, but found a more intuitive place for this sentence in the discussion and moved it there. LINES 327-328

**5. Lines 167–168: "Antecedent medians for turbidity were estimated from seven days prior due to missing sensor data." This is not clear, please explain what this means. There was always missing data for turbidity within the three days prior to an event? I can't imagine turbidity changes very much at baseflow.**

RESPONSE: We now include the following text in our revised manuscript to clarify this difference: "We had to remove poor-quality data from the turbidity dataset and chose to set methods that would accommodate inclusion of the most storms for our analysis. We compared the outcome of changing the number of days prior for events with turbidity data available for both 3- and 7-day analyses and found no difference in the results." LINES 209-211

**6. Lines 190–191: "Three of 15 flow events stimulated ER, 5 repressed ER, and ER did not deviate from the antecedent equilibrium for 7 events (i.e., M ER was 0)." It's more common to use numerals for numbers greater than 10, and to spell the numbers out for numbers less than 10.**

RESPONSE: Agreed. We edited to write out lower numbers per this journal's guidelines here and elsewhere. Thanks.

//

**REVIEWER 2:**

**General comments: The manuscript bg-2020-304 "Resistance and resilience of stream metabolism to high flow disturbances" by O'Donnel & Hotchkiss analyzes in a third-order stream the response of Gross Primary Production (GPP) and Ecosystem Respiration (ER) altered by disturbances such as isolated high flow events. The study is relevant as it is based on a long-term monitoring (5 years) of GPP and ER, which is critical to decipher seasonal and multiyear variability of stream ecology in the context of climate change. Overall, I found the approach of the study interesting but the authors should explore their dataset further, therefore I suggest major revisions.**

RESPONSE: Thank you for your positive impression of our work. We've responded to your suggestions for further explorations below.

**Major comments: I was surprised that the authors did not discuss about in-stream net ecosystem production (NEP). NEP is critical to decipher stream ecology as it does indicate whether an ecosystem is fixing more C than is respiring. The authors showed that ER has higher resistance and resilience in comparison to GPP, thereby should shifted NEP towards heterotrophy (decrease of the GPP:ER ratio). I believe it would be very interesting for the reader to understand/know how NEP is affected by high flow events. I suggest adding figures and discussion about NEP.**

RESPONSE: While our aim was to focus on the response and recovery of specific processes (GPP versus ER), we agree that changes in the balance between GPP and ER (i.e., NEP) are an important outcome of different metabolic responses. Most, if not all, of the discussion about changes to NEP are similar to ER, so were careful to avoid repetition in places where we add discussion related to NEP and refer to our recent paper discussion overall patterns in discharge, ER, GPP, and NEP where this was one of our primary objectives (O'Donnell & Hotchkiss 2019). An example of text added in response to this comment: "While we do not discuss net ecosystem production results in the context of this work because the patterns mirror those for ER (O'Donnell and Hotchkiss, 2019), we note that during the time periods of different GPP and ER responses and recovery to higher flow, Stroubles Creek was even more heterotrophic due to the higher resistance and resilience of ER relative to GPP." LINES 269-272.

**The dataset used by the authors is extended in time but the paper lacks of seasonal variability analysis. How GPP, ER and NEP, resistance and resilience are af- fected by seasons and by year-to-year variability. Indeed, temperature effect on stream metabolism is usually significant. The authors needs justify that the variability induced by the temperature does not overcome the variability induced by flow events. In the revised paper, I suggest the authors adding a figure such as GPP vs ER with points colored according to seasons or river flow.**

RESPONSE: This is certainly an interesting question that we hoped to pursue, but one that we found to be beyond the scope of something we could discuss with certainty because of the few isolated storm events with appropriate data and model output for our full analyses (n=15), thus limiting our ability to make concrete conclusions related to seasonality. We were able to discuss patterns of GPP and ER more broadly (not for specific flow events) in O'Donnell and Hotchkiss 2019, and better integrate and discuss results from that analysis throughout the revised paper.

**In low order streams, GPP and ER are affected by groundwater inputs, as groundwater inputs are usually significant in such streams. Groundwater exhibit usually low oxygen concentration, which may be problematic when GPP and ER are based on oxygen monitoring. Inputs of low-O2 groundwater in stream can overestimate ER and under- estimate GPP. However, the equation 1 does not take into account groundwater inputs. Why? Oxygen measurements during high flow, especially in low-order streams, can give erroneous values, so are the authors sure to measure appropriate values during the high flow events.**

RESPONSE: We are very aware of the challenges associated with groundwater inputs and metabolism modeling assumptions. We updated the methods to state that we did not see evidence for groundwater inputs in our study reach using conservative tracer additions (not part of this study, but part of other work at this site). While we of course cannot confirm that groundwater dynamics don't change at the highest flows that prevent access to this monitoring site (and, realistically, make tracer experiments less feasible!), we have collected the supporting data to the best of our abilities to ensure high-quality metabolism estimates.

New text: "Conservative tracer additions (Hotchkiss and O'Donnell, unpublished data) suggested there are no substantial groundwater inputs to this study reach that would otherwise bias our estimates of GPP and ER (Hall and Hotchkiss 2017)." LINES 123-125.

**There is some variability in day-to-day metabolism rates; therefore, I do not understand why the authors took the maximum or minimum value of GPP (or ER) from the antecedent range to estimate the resistance. I believe that the median or the mean would be more appropriate. In addition, why the authors used 3 days as the antecedent range. Is it arbitrary?**

RESPONSE: We updated our methods to ensure readers understand that analyzing the 3 day antecedent range was not arbitrary, it was the most appropriate trade-off between keeping our analysis to isolated flow events (the aim of our study) but including more than one or two days to estimate the dynamic range of prior conditions and metabolic rates (e.g., "pulsing equilibrium", Figures 1,3). If we used >3 days, we would have fewer flow events to analyze. If we used <3 days, we do not have as much information about prior conditions and metabolic rates with which to track metabolic responses to changing flow.

New text includes: "The goal of this work was to assess how metabolism responded to and recovered from higher flow events that were also isolated flow events. We focused on quality over quantity when selecting for and analyzing stream metabolism results before, during, and after high flow events. After all appropriate quality checking measures, we had 1375 days of metabolism estimates over five years (as reported in O'Donnell & Hotchkiss, 2019). To calculate resistance and recovery, we needed consecutive days of high-quality metabolism estimates, which further limited the number of high flow events appropriate for our analyses. For example, in 2016: there were 52 (out of 352) days with quality-checked sensor data that had a 50% flow change relative to the day prior. After looking at these 52 storms and selecting those that had three days before and three days after without any other flow events, we had 12 that were isolated. After quality-checking our metabolism estimates for all of those days, we had four high flow events from 2016 that passed all quality-checking steps required for this analysis." LINES 165-173

In response to the reviewer question about using max/min or median/mean we added the following text: "We use the maximum or minimum values instead of the median or mean because this approach allowed us to better capture the full range of average metabolism estimates in ways that summarizing pre-storm rates to means or medians would exclude." LINES 185-187

**I have concerns on how the isolated flow events are selected. Indeed, in the figure 2A, I observe that only few high flow events (15 events in 5 years) are actually selected by the authors. In the figure 2C, the authors did not provide statistical analysis on the difference of cumulative daily discharge between all days and isolated flow events. Is it statistically different? Visually, it seems not, considering the high range of cumulative daily discharge during "all days". If it is not statistically different, it means that the disturbance is the same in both groups. Is there a way for the authors to arbitrary select a greater number of high flow events? As examples, the authors could use maximum daily discharge vs cumulative daily discharge or the change in discharge from pre- to peak-storm flow. By the way, I do not understand why the authors wants to select isolated flow events rather than all high flow events. I believe that estimating resistance and resilience in each high flow events would be much more robust. In addition, the paper aims to study ecosystem response to high flow events, but the paper do not contain figures showing the relation of river flow versus stream metabolism. What is the relationship between river flow and GPP, ER, NEP, resistance and resilience? Resistance**

**represents the change in GPP (or ER) during a change in river flow, so maybe it would be interesting to show ∆GPP (or ER) with ∆Q?**

RESPONSE: Please see the general response at the top of this document for more information about how we prioritize quality over quantity in selecting isolated flow events and added text to the revised manuscript to better describe these decisions. Figure 2C highlights how the isolated flow events we were able to use for this analysis were typical of the range of flow changes we see at this site, and should thus give appropriate insights into the range of potential changes to GPP and ER due to flow disturbances. As mentioned above, we refer readers to O'Donnell and Hotchkiss 2019 for the paper that discusses all of the flow and metabolism data but did not analyze how metabolic rates changed during and after isolated flow disturbances (the objective of this paper).

**I the discussion section, I do not feel that the authors fully responds to their four hypotheses. How can the authors responds to H2 where they actually do not show carbon or nutrients measurements? H0 is strongly dependent on how you arbitrary selected the flow events. To my point of view, with their study design (unless the authors have measurements of carbon and nutrients) the authors can discuss only about H1 and H3. In addition, I also suggest rewriting the Discussion section in a more logical sense following the order of their hypotheses.**

RESPONSE: We agree, and as mentioned in our response to Reviewer 1, have revised our manuscript to better align the introduction and discussion topics. While we do not have high-frequency concentrations of carbon and nutrients to compare with patterns in metabolism RE: H2, we note that if metabolic rates did follow a hump-shaped curve as predicted by subsidy-stress, the changes in rates are more appropriate to test metabolic responses than changes in concentrations, as the concentrations themselves have already been altered by any carbon and nutrient uptake that occurred before the sampling point (i.e., concentrations may reflect potential for process, but we cannot know what is missing from carbon and nutrient pools if it's already been removed from the water by biota).

**Minor Comments:**

**L.1: Please, add somewhere in the abstract the ranges of ER, GPP, NEP, resistance and resilience.**

RESPONSE: We added the following (while still trying to keep a balance between including key info and keeping the abstract short!): "Metabolism was variable from day to day, even during lower flows; median and ranges for GPP and ER over the full measurement period were 3.7 (0.0, 17.3) and -9.6 (-2.2, -20.5) g $O_2$ m$^{-2}$ d$^{-1}$." and "Median $M_{GPP}$ and $M_{ER}$ were -0.14 and 0.00, respectively." The resilience results were included as "GPP was less resilient and took longer to recover (0 to >9 days, mean = 2.2) than ER (0 to 2 days, mean = 0.6)."

**L.10-11: You defined the metabolic resistance as the magnitude of departure from the dynamic equilibrium during antecedent lower flows, so why using the words "ER magnitude of departure" to refer to resistance. Better used the word resistance and resilience throughout the text once you have defined those words.**

RESPONSE: We wanted to be true to the method used when referring to specific analyses and results, but took the opportunity to streamline terminology throughout the manuscript to clarify when we are talking about specific metrics and when we are referring to broader topics. Our revisions tried to strike the best balance between reviewer language preferences, which did not always appear to be aligned as communicated.

**Please add also in the abstract that more ER or GPP is resistant less the magnitude of departure is large.**

RESPONSE: We had trouble understanding this request as it was written, but updated the abstract to include the following: "We calculated metabolic resistance as the magnitude of departure ($M_{GPP}$, $M_{ER}$) from the range of metabolism during during antecedent lower flows (lower values of M represent higher resistance) and estimated resilience as the time until GPP and ER returned to the prior range of ambient equilibrium."

**L. 69: It is strange to start with H1 and finish with H0**

RESPONSE: We updated the entire last paragraph of the introduction, which includes the re-ordering of hypotheses per reviewer preference. LINES 72-87 for the full paragraph, which includes: " We had four hypotheses (Figure 1): (H1) ER will be more resistant than GPP to flow disturbances, given the protection of many heterotrophs within the streambed; (H2) there will be a stimulation of GPP and ER at intermediate flow disturbances due to an influx of limiting carbon and nutrients; (H3) metabolic resistance and resilience will change with the size of the event, with larger flow disturbances inducing more stress due to enhanced scour;and (H4) some flow events will not push GPP and ER outside of their ambient dynamic equilibrium."

**L.81: Usually precipitation is in mm**

RESPONSE: We changed to mm.

**L.90: How did you calibrate the different sensors, and how often did you check the calibration?**

RESPONSE: Good question. Sensors were calibrated every 2-4 weeks. We updated this section of the methods with more information: "Sensors were calibrated every 2-4 weeks according to best practice recommendations from the manufacturer (Hession et al. 2020) or, in the case of the PME DO sensor, with Winkler titration checks of our 100% and 0% calibration solutions (Hall and Hotchkiss 2017, O'Donnell and Hotchkiss 2019)." LINES 111-113

**L.93: can you add the weather station on the figure A1. The figure A1 needs a scale, a geographic footprint.**

RESPONSE: Great suggestions. We noted the location of the weather station in Figure A1 (just west of the upper watershed delineated above the monitoring site, but still in the stream valley) and added a scale bar. We are not sure what the reviewer means by geographic footprint, but perhaps that will be resolved with the scale bar.

**L105: Please, specify that you works with gas exchange coefficient not gas exchange velocity.**

RESPONSE: We updated text as suggested.

**L.110: How did you measure the PAR? How did you calculate the average depth?**

RESPONSE: Both of these questions are addressed in updated text:

"We obtained the data needed to model the relative change in light over 24-hours (Equation 1) from a nearby weather station (Figure A1), which also provided estimates of barometric pressure." LINES 106-107

"Velocity (v) and width (w) measurements were taken over multiple years to create site-specific relationships between stage, velocity, wetted width, and discharge (Q). A stage-discharge relationship was created in 2013 and updated in 2018 to allow for daily estimates of depth (z) from Q = vwz." LINES 109-111

**L.119: What are the values of K?**

RESPONSE: Daily K estimates are in our Supplementary Data file of daily metabolism estimates. K for the high flow days analyzed are in Table 2.

**L.133: Please define Qi**

RESPONSE: Thank you for catching this. We defined in our revised manuscript: "where $Q_i$ is the discharge of the day of interest and $Q_{i-1}$ is the discharge of the day prior." LINE 154

**L.167:169: For the different variables other than GPP and ER you used the medians from three days prior the flow event for correlations, but for resistance you used the maximum or minimum GPP or ER before the flow event. I believe it would be robust to use the same methods.**

RESPONSE: We used GPP and ER range to account for the variability in metabolism (and to be consistent with how we assessed metabolism for other analyses). We added the following text: "We use the maximum or minimum values instead of the median or mean because this

approach allowed us to better capture the full range of average metabolism estimates in ways that summarizing pre-storm rates to means or medians would exclude." LINES 185-187

**L.180: Is the cumulative daily discharge statistically different between isolated events and other days?**

RESPONSE: We interpret this question as asking whether discharge on high flow days for an isolated event was different than lower flow days before/after? Yes at the scale of a single flow event. No at the scale of comparing all of the flow relative to the 15 storms we analyzed.

**L.182-185: As mentioned in the major comments please showed how GPP, ER and NEP are affected by seasons and river flow.**

RESPONSE: Please see our response above to the earlier major comment.

**L219 Where can I see that ER was more resistant than GPP. It is on a daily basis? Yearly basis? Multi-year basis? Please give some details, some stats should be applied. Figure 1 do not show your results.**

RESPONSE: We changed this citation to include data figures that support this statement: "Notably, ER was more resistant than GPP (Figures 5,6)." LINE 264

**L.228: Same comments 228-230: Can you show some results confirming what you stipulate? In the table 4 turbidity seems weakly correlated with resilience of ER and GPP.**

RESPONSE: It was weakly correlated across the 15 events appropriate for our analyses. We address this more fully in O'Donnell & Hotchkiss 2019 with daily metabolism and turbidity data and, as mentioned above, expanded our discussion a bit (while trying not to replicate the discussion of the 2019 paper) in our revised manuscript.

**254: The authors have a dataset representing 5 years of monitoring so why they cannot answer to this question, at least partly?**

RESPONSE: This was beyond the scope of this project and more appropriate for sites with more high flow events that conform to sensor and metabolism QA/QC, hence the call for 'future analyses'.

**Figure 5: I am not convinced by this figure.**

RESPONSE: It was unclear from the reviewer comment why this figure is not convincing, but Reviewer 1 was also unenthusiastic about this presentation! So, as also described in our response to Reviewer 1, we cleaned up the Figure 5A panel (now much easier to see all points, linear model, etc.) and removed Figure 5B.

**Figure 6: Same data as in the table 4, perhaps not relevant.**

RESPONSE: Agreed. We moved Figure 6 to the appendix (Figure A21) in our revised manuscript.

**Figure 8: Is there a better way to present these results?**

RESPONSE: We are not sure what about this graph did not work for the reviewer, so we are not sure how best to respond to this comment. Because we were limited in the data collected and reported by other studies, wanted to be as inclusive as possible, and thought placing our work in the context of other flow-metabolism studies was important, this was the best format to calculate and highlight as many metabolic responses and recovery intervals across studies as possible.

**Table 4: Please indicate the p-values, Indeed, two parameters can have a correlation coefficient greater than 0.5 but they are still not correlated together if the p-value is greater than 0.05.**

RESPONSE: We included p-values in our revised Table 4. We also clarified the language we used associated with different ranges in r in the methods.

**Figure A4 to A18: In each isolated flow events: GPP, ER and Discharge can be combined in one figure with 3 axis**

RESPONSE: They can, but we prefer to avoid multiple axes on plots whenever possible to avoid potential misinterpretation of data.

**Figure A20: To my point of view a figure such as this one showing the seasonal variability of the different parameters (GPP, ER, NEP, résistance and resilience) is important and must appear in the main text.**

RESPONSE: Please see our comments above RE: challenges with assessing differences among seasons (we do not have the data power to do this well). We did make sure we more clearly refer readers to O'Donnell & Hotchkiss 2019, where daily metabolism data (not a subset of data focused on analyzing high flows) are graphed with different symbols for each season.

---

## Referee Report (RR1)

**Major**

Overall, I appreciate the effort the authors made to clarify nearly every technical point requested by reviewers. However, it does not appear to me that the authors made any of the more major suggested changes to their analyses. Instead, they opted to defend the previous presentation of results. If this is OK with the Editor, it's OK with me. Below I present the critical examples:

1. Instead of showing or demonstrating a sensitivity analysis of their choice for the 50% cumulative discharge threshold, they simply added lines 160–164, which more or less say, "take our word for it". Even a simple discussion of results from a sensitivity analysis would suffice, but as written, the reader must accept the fact that the 10% and 50% choices were best. What about others who want to replicate the study? Should they use these values because they are shown here? I do not ask to be annoying to the authors and I trust they did their due diligence, but I think that by being more critical/quantitative in their choice here, they are also helping other researchers in their work, as well.

2. Similarly, I still do not understand the choice and rationale (Lines 185–187) for using the maximum/minimum as the departure baseline: "We use the maximum or minimum values instead of the median or mean because this approach allowed us to better capture the full range of average metabolism estimates in ways that summarizing pre-storm rates to means or medians would exclude". I apologize if I am being dense, but this is hard to follow. How do maxima or minima better "capture the full range of *average* [conditions]" (bold and italics mine) better than means or medians? This illustrates to me that the authors did not take into consideration both reviewers' valid point here, but instead discussed the issue away with one sentence. Would it be so hard to compare the results using mean previous conditions? It looks like it would drastically affect the results, implying strong sensitivity to method (see Figures A5–18).

3. The authors responded to the original suggestion about considering the effects of extremely small changes in magnitude as they relate to measurement uncertainty as follows: "We note that our metrics were indeed detectable relative to metabolism estimates (with low uncertainty, as shown in Fig 3, A4-A18, and supplementary data files). Consequently, we disagree that this approach "raises red flags." I have to disagree here, especially being familiar with uncertainty measurements in GPP and ER. I have looked at the supplementary files and the uncertainty measurements that the authors apparently used: the standard error of the mean. This value will always be extremely tiny because the authors are dividing the standard deviation of estimates by the square root of 2000 (the number of MCMC runs). A better uncertainty measurement would be the standard deviation, the IQR, or the 95% credible intervals, as are most commonly presented for the output of these Bayesian models. It is not reasonable to use the standard error of the mean in this regard and I find this to be a major issue with the approach. For example, the event on 2013–05–23 has overlapping credible intervals for the maximum GPP value on 2013–05–21 (2.2–4.6) and the event (2.5–4.9) And the event on 2013–06–02 has overlapping credible intervals for the maximum GPP value on 2013–05–31 (1.9–4.8) and the event (2.2–4.1). Most events are like this. Can you really distinguish between these with confidence? This is yet another reason why I think the mean of the previous three days would be better. Not to mention, the mean would better capture *ambient **equilibrium** conditions.*

**Minor**

Line 96 : Please specify if $NO_3$ is as N or not.

---

## Author Response (AR2)

We respond to the three major points raised by reviewer 1 below after the original text from reviewer 1 (in bold). We have not included a response to the editor because they did not highlight specific topics to focus on or identify additional needs for revisions beyond those listed by reviewer 1.

**Major**

Overall, I appreciate the effort the authors made to clarify nearly every technical point requested by reviewers. However, it does not appear to me that the authors made any of the more major suggested changes to their analyses. Instead, they opted to defend the previous presentation of results. If this is OK with the Editor, it's OK with me. Below I present the critical examples:

1. Instead of showing or demonstrating a sensitivity analysis of their choice for the 50% cumulative discharge threshold, they simply added lines 160–164, which more or less say, "take our word for it". Even a simple discussion of results from a sensitivity analysis would suffice, but as written, the reader must accept the fact that the 10% and 50% choices were best. What about others who want to replicate the study? Should they use these values because they are shown here? I do not ask to be annoying to the authors and I trust they did their due diligence, but I think that by being more critical/quantitative in their choice here, they are also helping other researchers in their work, as well.

**RESPONSE:** We appreciate that sensitivity analyses are useful tools in method development and validation, but it is unclear what the reviewer is asking for in a sensitivity analysis or different type of storm threshold cutoff and what would resolve their concern in a revised manuscript. Our limited understanding of reviewer vision based on what they wrote requires us to (a) start over with all analyses but without guidelines for comparison when deciding which threshold choice or (b) quantitatively justify to the reviewer why we kept previous thresholds if we don't re-do all analyses to include smaller "storms" or a different definition of baseflow for prior day calculations (different than <10% change in Q on prior days). Because we do not have a clear direction from the reviewer or a methods precedent in the literature, we have respectfully chosen not to re-do our analyses in one or more ways that may or may not satisfy reviewer questions.

Our early analyses of different flow thresholds were not at the level of doing a full analysis to delineate storms and analyze metabolic resistance and recovery for different delineation thresholds, but we did test how different % discharge (Q) change thresholds altered the number of isolated storm events we would be able to analyze. That preliminary analysis increased our storm threshold to 50% Q change to include more isolated storm events and added the requirement of less than 10% Q change during what we categorized as baseflow pre-storm days to remove days with smaller storms prior to larger storms from our analysis. For the 15 storms analyzed, we captured a wide range flow changes: 53 - 1105% change in cumulative daily Q.

We also note that there is not a standardized method for characterizing metabolic responses to flow disturbances, and we chose distinct cutoffs based on our *site-specific* understanding of changes in hydrology during baseflow and higher flows. As one example of this, we present the relevant methods from Reisinger et al. (2017, Ecosphere), one of the few other manuscripts that

analyzed storm-induced changes in stream metabolism after Hurricane Sandy and a subset of smaller storms: "For the storm events using the four additional sites, we sorted through a daily discharge and metabolism record spanning April–November 2015 to identify high-flow events with enough time at baseflow between events to allow for metabolic recovery. We only included events with at least four days of baseflow prior to the flood with relatively stable GPP and ER, coupled with enough time following the flood for GPP and ER to recover to pre-flood rates. We selected nine stream–storm events to include in addition to the Sandy data."

Given what we present above, past responses to reviewers on this topic, and a brief communication with the editor, we did not go through a total re-analysis of all of our data without a specific justification, resource, or expected outcome for this suggestion. We sincerely hope that our revised manuscript will not be penalized by this choice. We were simply not in a position to start over without clear guidance or justification and could not see how this exercise would be aligned with our objectives for this work.

2. Similarly, I still do not understand the choice and rationale (Lines 185–187) for using the maximum/minimum as the departure baseline: "We use the maximum or minimum values instead of the median or mean because this approach allowed us to better capture the full range of average metabolism estimates in ways that summarizing pre-storm rates to means or medians would exclude". I apologize if I am being dense, but this is hard to follow. How do maxima or minima better "capture the full range of average [conditions]" (bold and italics mine) better than means or medians? This illustrates to me that the authors did not take into consideration both reviewers' valid point here, but instead discussed the issue away with one sentence. Would it be so hard to compare the results using mean previous conditions? It looks like it would drastically affect the results, implying strong sensitivity to method (see Figures A5–18).

**RESPONSE:** In attempt to clarify this approach and our motivation for this approach in a different way: we used the highest (maximum) and lowest (minimum) mean estimate of daily GPP and ER during the 3+ pre-storm baseflow days. We did this because day-to-day variation in metabolism is high, and we believed that taking the mean or median of these 3+ days of median GPP and ER was doing a disservice to the characteristics of our study site.

However, based on the above request from reviewer 1 and to keep with older methods used by other researchers, we reanalyzed metabolism changes during (M, magnitude of departure) and after (RI, recovery interval) storms compared to the mean GPP or ER during the previous baseflow period (i.e., mean GPP over 3 days or mean ER over 3 days). Below we show updated plots (Fig 3,5,6,7 and appendices), tables (table 3,4), and text.

We revised the methods text for "MaxMin" or "Mean" in section 2.5 (lines 270-319 in the revised manuscript):

**"2.5 Characterizing metabolic resistance and resilience**

"To acknowledge the ambient day-to-day variability of GPP and ER, we used metabolism estimates from three days prior each isolated flow event to calculate a mean value of antecedent metabolism. We quantified metabolic responses to flow disturbances by comparing the pre-event metabolic means with event and post-event metabolism rates. To assess resistance, we estimated the metabolic magnitude of departure (M) during events to quantify the resistance of GPP and ER to higher flow disturbances. We calculated M per isolated flow event by comparing the difference between GPP and ER to the nearest value of the antecedent range (Equation 3; Figure 3),

[Eqn 3 – code copied from LaTeX version for track change]

 $M = 1 - \{X_{event}\} \{X_{prior}\}$

where  $X_{event}$  is either GPP or ER (g O2 m-2 d-1) on the day of the isolated flow event.  $X_{prior}$  is the mean value of GPP or ER from the antecedent range, and whether M is positive or negative depends on if the isolated flow event resulted in a stimulated (increased) or suppressed (reduced) metabolic response. For instance, if GPP declined during a flow event, M was calculated as the difference between GPP for the isolated flow event and the mean GPP from the antecedent 3-day range (Figure 3). If GPP or ER on the event day did not fall above or below the antecedent mean, M was zero, thus indicating high resistance. A negative M represents a suppression, and a positive M a stimulation, of GPP or ER relative to the antecedent mean.

To quantify the resilience of GPP and ER, we estimated recovery intervals (RI) by counting the number of days until metabolic rates returned to or exceeded pre-event mean GPP or ER, signifying a return to antecedent conditions (Figure 3). If metabolism (mean and 2.5-97.5% credible intervals) during the isolated flow event did not fall outside of the antecedent mean, the RI was zero days (metabolism cannot recover if it never shifts outside ambient values). To ensure additional flow events did not obscure the recovery interval of GPP or ER, we stopped counting RI the day before the next event (i.e., if another flow event happened four days later, we stopped counting RI at three days). To test for statistically significant differences between ER and GPP recovery intervals (RIER and RIGPP) and ER and GPP magnitude of departure (MER and MGPP), we ran Welch's t-tests in R (R Core Team, 2018)."

We updated the following figures, tables, and text to include results based on analyzing M and RI compared to a 3-day mean instead of highest or lowest mean GPP or ER prior to each storm. We also used this as an opportunity to improve some of our data visualizations.

(1) Figure 3 now removes the window of baseflow variability from our calculations (original dashed lines) and shows the calculation of M and RI relative to median prior GPP. Updated/original figure are below for comparison.

Above: Revised Figure 3 (left panel) illustrating the method requested by reviewer using the median instead of the min/max daily median (original figure showing this method is the right panel).

(2) Figure 5 now includes M-GPP and M-ER calculated using the "mean" approach instead of the "MaxMin" approach. Updated/original figure are below for comparison.

Above: Revised Figure 5 (left panel) with updated approach to calculate M. Original figure with these results, including M and RI (removed as requested by reviewer) is shown in the righthand two panels.

(3) Figure 6 now includes % Q change versus updated estimates of M-GPP and M-ER calculated using the "mean" approach. Updated/original figure are below for comparison.

Above: Revised Figure 6 with updated M from revised method (left panel) in comparison to original figure showing these results (right panel).

(4) Figure 7 now plots our data as RI from the "mean" approach, not "MaxMin". Updated/original figure are below for comparison.